# Federated Expectation Maximization with heterogeneity mitigation and variance reduction

**Aymeric Dieuleveut**
Centre de Mathématiques Appliquées
Ecole Polytechnique, France
Institut Polytechnique de Paris
aymeric.dieuleveut@polytechnique.edu

**Gersende Fort**
Institut de Mathématiques de Toulouse
Université de Toulouse; CNRS
UPS, Toulouse, France
gersende.fort@math.univ-toulouse.fr

**Eric Moulines**
Centre de Mathématiques Appliquées
Ecole Polytechnique, France
CS Dpt, HSE University, Russian Federation
eric.moulines@polytechnique.edu

**Geneviève Robin**
Laboratoire de Mathématiques
et Modélisation d'Évry
Université d'Évry Val d'Essonne; CNRS
Évry-Courcouronnes, France
genevieve.robin@cnrs.fr

## Abstract

The Expectation Maximization (EM) algorithm is the default algorithm for inference in latent variable models. As in any other field of machine learning, applications of latent variable models to very large datasets makes the use of advanced parallel and distributed architectures mandatory. This paper introduces FedEM, which is the first extension of the EM algorithm to the federated learning context. FedEM is a new communication efficient method, which handles partial participation of local devices, and is robust to heterogeneous distributions of the datasets. To alleviate the communication bottleneck, FedEM compresses appropriately defined complete data sufficient statistics. We also develop and analyze an extension of FedEM to further incorporate a variance reduction scheme. In all cases, we derive finite-time complexity bounds for smooth non-convex problems. Numerical results are presented to support our theoretical findings, as well as an application to federated missing values imputation for biodiversity monitoring.

## 1 Introduction

The Expectation Maximization (EM) algorithm is the most popular approach for inference in latent variable models. The EM algorithm, a special instance of the Majorize/Minimize algorithm [24], was formalized by [8] and is without doubt one of the fundamental algorithms in machine learning. Applications include among many others finite mixture analysis, latent factor models inference, and missing data imputation; see [38, 29, 26, 13] and the references therein. As in any other field of machine learning, training latent variable models on very large datasets make the use of advanced parallel and distributed architectures mandatory. Federated Learning (FL) [22, 39], which exploits the computation power of a large number of edge devices to perform distributed machine learning, is a powerful framework to achieve this goal.

The conventional EM algorithm is not suitable for FL settings. We propose several new distributed versions of the EM algorithm supporting compressed communication. More precisely, our objective

35th Conference on Neural Information Processing Systems (NeurIPS 2021).

is to minimize a non-convex finite-sum smooth objective function

$$\mathrm{Argmin}_{\theta \in \Theta} F(\theta), \qquad F(\theta) := \frac{1}{n} \sum_{i=1}^{n} \mathcal{L}_i(\theta) + \mathsf{R}(\theta) , \qquad \Theta \subseteq \mathbb{R}^d , \qquad (1)$$

where $n$ is the number of workers/devices which are connected to a central server, and the worker $\#i$ only has access to its local data; finally $\mathsf{R}$ is a penalty term which may be introduced to promote sparsity, regularity, etc. In latent variable models, $\mathcal{L}_i(\theta) = -m^{-1} \sum_{j=1}^{m} \log p(y_{ij}; \theta)$, where $\{y_{ij}\}_{j=1}^{m}$ are the $m$ observations available for worker $\#i$, and $p(y; \theta)$ is the *incomplete* likelihood. $p(y; \theta)$ is defined by marginalizing the *complete-data* likelihood $p(y, z; \theta)$ defined as the joint probability density function of the observation $y$ and a non-observed latent variable $z \in \mathsf{Z}$, i.e. $p(y; \theta) = \int_{\mathsf{Z}} p(y, z; \theta) \mu(\mathrm{d}z)$ where $\mathsf{Z}$ is the *latent space* and $\mu$ is a measure on $\mathsf{Z}$. We focus in this paper on the case where $p(y, z; \theta)$ belongs to a curved exponential family, given by

$$p(y, z; \theta) := \rho(y, z) \exp \left\{ \langle s(y, z), \phi(\theta) \rangle - \psi(\theta) \right\} ; \qquad (2)$$

where $s(y, z) \in \mathbb{R}^q$ is the *complete-data sufficient statistics*, $\phi : \Theta \to \mathbb{R}^q$ and $\psi : \Theta \to \mathbb{R}$, $\rho : \mathsf{Y} \times \mathsf{Z} \to \mathbb{R}^+$ are vector/scalar functions.

In absence of communication constraints, the EM algorithm is a popular method to solve (1). It alternates between two steps: in the Expectation (E) step, using the current value of the iterate $\theta_{\mathrm{curr}}$, it computes a majorizing function $\theta \mapsto \mathsf{Q}(\theta, \theta_{\mathrm{curr}})$ given up to an additive constant by

$$\mathsf{Q}(\theta, \theta_{\mathrm{curr}}) := - \langle \bar{\mathsf{s}}(\theta_{\mathrm{curr}}), \phi(\theta) \rangle + \psi(\theta) + \mathsf{R}(\theta) \quad \text{where} \quad \bar{\mathsf{s}}(\theta) := \frac{1}{n} \sum_{i=1}^{n} \bar{\mathsf{s}}_i(\theta) ; \qquad (3)$$

and $\bar{\mathsf{s}}_i(\theta)$ is the $i$th device conditional expectation of the complete-data sufficient statistics:

$$\bar{\mathsf{s}}_i(\theta) := \frac{1}{m} \sum_{j=1}^{m} \bar{\mathsf{s}}_{ij}(\theta) , \quad \bar{\mathsf{s}}_{ij}(\theta) := \int_{\mathsf{Z}} s(y_{ij}, z) p(z|y_{ij}; \theta) \mu(\mathrm{d}z) , \qquad (4)$$

where $p(z|y_{ij}; \theta) := p(y_{ij}, z; \theta)/p(y_{ij}; \theta)$. As for the M step, an updated value of $\theta_{\mathrm{curr}}$ is computed as a minimizer of $\theta \mapsto \mathsf{Q}(\theta, \theta_{\mathrm{curr}})$. The majorizing function is then updated with the new $\theta_{\mathrm{curr}}$; this process is iterated until convergence. The EM algorithm is most useful when for any $\theta_{\mathrm{curr}} \in \Theta$, the function $\theta \mapsto \mathsf{Q}(\theta, \theta_{\mathrm{curr}})$ is a convex function of the parameter $\theta$ which is solvable in $\theta$ either explicitly or with little computational effort. A major advantage of the EM algorithm stems from its invariance under homeomorphisms, contrary to classical first-order methods: the EM updates are the same for any continuous invertible re-parametrization [23].

In the FL context, the vanilla EM algorithm is affected by three major problems: (1) the communication bottleneck, (2) data heterogeneity, and (3) partial participation (PP) of the workers.

When the number of workers is large, the cost of communication becomes overwhelming. A classical technique to alleviate this problem is to use *communication compression*. Most FL algorithms are first order methods and compression is typically applied to stochastic gradients. Yet, these methods are not appropriate to solve (1) since *(i)* they do not preserve the desirable homeomorphic invariance property, and *(ii)* the full EM iteration is not distributed since the M step is performed by the central server only. This calls for an extension of the EM algorithm to the FL setting.

Since workers are often user personal devices, the issue of data heterogeneity naturally arises. Our model in Equations (1), (3) and (4) allows the local loss functions to depend on the worker $i \in \{1, \ldots, n\}$ and the observations $y_{ij}$ to be independent but not necessarily identically distributed. In addition, our theoretical results deal with specific behaviors for each worker $i \in \{1, \ldots, n\}$, see e.g., A5, 7 and 8. In the FL-EM setting, heterogeneity manifests itself by the non-equality of the *local* conditional expectations of the complete-data sufficient statistics $\bar{\mathsf{s}}_i$'s; modifications to the algorithms must be performed to ensure convergence at the central server.

Finally, a subset of users are potentially inactive in each learning round, being unavailable or unwilling to participate. Thus, taking into account PP of the workers and its impact on the convergence of algorithms, is a major issue.

• `FedEM`. The main contribution of our paper is a new method called `FedEM`, supporting communication compression, partial participation and data heterogeneity. In this algorithm, the workers compute an estimate of the *local complete-data sufficient statistics* $\bar{\mathsf{s}}_i$ using a minibatch of data, apply an unbiased compression operator to a noise compensated version (using a technique inspired by [17, 15]) and send the result to the central server, which performs aggregation and the M-step (i.e. the parameter update).

• `VR-FedEM`**.** We improve `FedEM` by adding a variance reduction method inspired by the `SPIDER` framework [9] which has recently been extended to the EM framework [10]. For both `FedEM` and `VR-FedEM`, the central server updates the expectations of the global complete-data sufficient statistics through a Stochastic Approximation procedure [3, 4]. When compared to `FedEM`, `VR-FedEM` additionally performs variance reduction for each worker, progressively alleviating the variance brought by the random oracles which provide approximations of the local complete-data sufficient statistics.

• **Theoretical analysis.** EM in the curved exponential family setting converges to the roots of a function h (see e.g. Section 2). We introduce a unified theoretical framework which covers the convergence of `FedEM` and `VR-FedEM` algorithms in the non-convex case and establish convergence guarantees for finding an $\epsilon$-stationary point (see Theorem 1 and Theorem 3). In both cases, we provide the number $K_{\mathrm{opt}}(\epsilon)$ of optimization steps and the number $K_{\mathrm{CE}}(\epsilon)$ of computed conditional expectations to reach $\epsilon$-stationarity. These results show that in the Stochastic Approximation steps of `VR-FedEM` , the step sizes are independent of $m$, the number of observations per server. Furthermore, the computational cost in terms of $\mathcal{K}_{\mathrm{CE}}(\epsilon)$ improves on earlier results. In this respect, `VR-FedEM` has the same advantages as `SPIDER` [9] compared to `SVRG` [18] and `SAGA` [6], or as `SPIDER-EM` [10] compared to `sEM-vr` [5] and `FIEM` [20, 11]. Lastly, our bounds demonstrate the robustness of `FedEM` and `VR-FedEM` to data heterogeneity.

• Finally, seen as a root finding algorithm in a quantized FL setting, `VR-FedEM` can be compared to `VR-DIANA` [17]: we show that `VR-FedEM` does not require the step sizes to decrease with $m$ and provides state of the art iteration complexity to reach a precision $\epsilon$.

**Notations.** For vectors $a, b$ in $\mathbb{R}^q$, $\langle a, b \rangle$ is the Euclidean scalar product, and $\|\cdot\|$ denotes the associated norm. For $r \geq 1$, $\|a\|_r$ is the $\ell_r$-norm of a vector $a$. The Hadamard product $a \odot b$ denotes the entrywise product of the two vectors $a, b$. By convention, vectors are column-vectors. For a matrix $A$, $A^\top$ is its transpose and $\|A\|_F$ is its Frobenius norm; for two matrices $A, B$, $\langle A, B \rangle :=$ $\mathrm{Trace}(B^\top A)$. For a positive integer $n$, set $[n]^\star := \{1, \cdots, n\}$ and $[n] := \{0, \cdots, n\}$. The set of non-negative integers (resp. positive) is denoted by $\mathbb{N}$ (resp. $\mathbb{N}^\star$). The minimum (resp. maximum) of two real numbers $a, b$ is denoted by $a \wedge b$ (resp. $a \vee b$). We will use the Bachmann-Landau notation $a(x) = O(b(x))$ to characterize an upper bound of the growth rate of $a(x)$ as being $b(x)$.

## 2 `FedEM`**: Expectation Maximization algorithms for federated learning**

Recall the definition of the negative penalized (normalized) log-likelihood $F(\theta)$ from (1). Along the entire paper, we make the following assumptions A1 to A3,which define the model at hand.

**A1.** *The parameter set $\Theta \subseteq \mathbb{R}^d$ is a convex open set. The functions $\mathsf{R} : \Theta \to \mathbb{R}$, $\phi : \Theta \to \mathbb{R}^q$, $\psi : \Theta \to \mathbb{R}$, and $\rho(y_{ij}, \cdot) : \mathsf{Z} \to \mathbb{R}_+$, $s(y_{ij}, \cdot) : \mathsf{Z} \to \mathbb{R}^q$ for $i \in [n]^\star$ and $j \in [m]^\star$ are measurable functions. For any $\theta \in \Theta$ and $i \in [n]^\star$, the log-likelihood is finite: $-\infty < \mathcal{L}_i(\theta) < \infty$.*

**A2.** *For all $\theta \in \Theta$ and $i \in [n]^\star$, the conditional expectation $\bar{\mathsf{s}}_i(\theta)$ is well-defined.*

**A3.** *For any $s \in \mathbb{R}^q$, the map $s \mapsto \mathrm{Argmin}_{\theta \in \Theta} \{\psi(\theta) + \mathsf{R}(\theta) - \langle s, \phi(\theta) \rangle\}$ exists and is unique; the singleton is denoted by $\{\mathsf{T}(s)\}$.*

EM defines a sequence $\{\theta_k, k \geq 0\}$ that can be computed recursively as $\theta_{k+1} = \mathsf{T} \circ \bar{\mathsf{s}}(\theta_k)$, where the map $\mathsf{T}$ is defined in A3 and $\bar{\mathsf{s}}$ is defined in (3). On the other hand, the EM algorithm can be defined through a mapping in the complete-data sufficient statistics, referred to as the *expectation space*. In this setting, the EM iteration defines a $\mathbb{R}^q$-valued sequence $\{\widehat{S}_k, k \geq 0\}$ given by $\widehat{S}_{k+1} = \bar{\mathsf{s}} \circ \mathsf{T}(\widehat{S}_k)$. Thus, we observe that the EM algorithm admits two equivalent representations:

$$\text{(Parameter space)} \ \theta_{k+1} = \mathsf{T} \circ \bar{\mathsf{s}}(\theta_k); \quad \text{(Expectation space)} \ \widehat{S}_{k+1} = \bar{\mathsf{s}} \circ \mathsf{T}(\widehat{S}_k). \quad (5)$$

In this paper, we focus on the expectation space representation; see [23] for an interesting discussion on the connection of EM and mirror descent. It has been shown in [7] that if $s_\star$ is a fixed point to the EM algorithm in the expectation space, then $\theta_\star := \mathsf{T}(s_\star)$ is a fixed point of the EM algorithm in the parameter space, i.e., $\theta_\star = \mathsf{T} \circ \bar{\mathsf{s}}(\theta_\star)$; note that the converse is also true. Define the functions $\mathsf{h}_i$ and $\mathsf{h}$ from $\mathbb{R}^q$ to $\mathbb{R}^q$ by $\mathsf{h}(s) := \frac{1}{n} \sum_{i=1}^n \mathsf{h}_i(s)$ with $\mathsf{h}_i(s) := \bar{\mathsf{s}}_i \circ \mathsf{T}(s) - s$ .

$$\mathsf{h}(s) := \frac{1}{n} \sum_{i=1}^n \mathsf{h}_i(s) \, , \qquad \mathsf{h}_i(s) := \bar{\mathsf{s}}_i \circ \mathsf{T}(s) - s \, . \quad (6)$$

A key property is that the fixed points of EM in the expectation space are the roots of the *mean field* $s \mapsto \mathsf{h}(s)$ (see (3) for the definition of $\bar{\mathsf{s}}$). Therefore, convergence of EM-based algorithms is

evaluated in terms of $\epsilon$-**stationarity** (see [14, 10]): for all $\epsilon > 0$, there exists a (possibly random) termination time $K$ s.t.: $\mathbb{E}\left[\|h(\widehat{S}_K)\|^2\right] \leq \epsilon$ . Another key property of EM is that it is a monotonic algorithm: each iteration leads to a decrease of the negative penalized log-likelihood i.e. $F(\theta_{k+1}) \leq F(\theta_k)$ or, equivalently in the expectation space $F \circ \mathsf{T}(\widehat{S}_{k+1}) \leq F \circ \mathsf{T}(\widehat{S}_k)$ (for sequences $\{\theta_k, k \geq 0\}$ and $\{\widehat{S}_k, k \geq 0\}$ given by (5)). A4 assumes that the roots of the mean field $h$ are the roots of the gradient of $F \circ \mathsf{T}$ (see [7] for the same assumption when studying Stochastic EM). A5 assumes global Lipschitz properties of the functions $h_i$'s.

**A4.** *The function* $\mathsf{W} := F \circ \mathsf{T} : \mathbb{R}^q \to \mathbb{R}$ *is continuously differentiable on* $\mathbb{R}^q$ *and its gradient is globally Lipschitz with constant* $L_{\dot{\mathsf{W}}}$. *Furthermore, for any* $s \in \mathbb{R}^q$, $\nabla \mathsf{W}(s) = -B(s)h(s)$ *where* $B(s)$ *is a* $q \times q$ *positive definite matrix. In addition, there exist* $0 < v_{\min} \leq v_{\max}$ *such that for any* $s \in \mathbb{R}^q$, *the spectrum of* $B(s)$ *is in* $[v_{\min}, v_{\max}]$.

**A5.** *For any* $i \in [n]^\star$, *there exists* $L_i > 0$ *such that for any* $s, s' \in \mathbb{R}^q$, $\|h_i(s) - h_i(s')\| = \|(\bar{s}_i \circ \mathsf{T}(s) - s) - (\bar{s}_i \circ \mathsf{T}(s') - s')\| \leq L_i \|s - s'\|$ .

### A Federated EM algorithm.

Our first contribution, the novel algorithm FedEM is described by algorithm 1. The algorithm encompasses partial participation of the workers: at iteration #$(k + 1)$, only a subset $\mathcal{A}_{k+1}$ of active workers participate to the training, see line 3. The averaged fraction of participating workers is denoted $p$. Each of the active workers #$i$ computes an *unbiased* approximation $\mathsf{S}_{k+1,i}$ (line 6) of $\bar{s}_i \circ \mathsf{T}(\widehat{S}_k)$; conditionally to the past (see Appendix D.2 for a rigorous definition), these approximations are independent. The workers then transmit to the central server a compressed information about the new sufficient statistics. A naive solution would be to compress and transmit $\mathsf{S}_{k+1,i} - \widehat{S}_k$, but data heterogeneity between servers often prevents these local differences from vanishing at the optimum, leading to large compression errors and impairing convergence of the algorithm. Following [28], a memory $V_{k,i}$ (initialized to $h_i(\widehat{S}_0)$ at $k = 0$) is introduced; and the *differences* $\Delta_{k+1,i} := \mathsf{S}_{k+1,i} - \hat{S}_k - V_{k,i}$ are compressed for $i \in \mathcal{A}_{k+1}$ (line 7

---

**Algorithm 1:** FedEM with partial participation

**Data:** $k_{\max} \in \mathbb{N}^\star$; for $i \in [n]^\star$, $V_{0,i} \in \mathbb{R}^q$;
$\widehat{S}_0 \in \mathbb{R}^q$; a positive sequence $\{\gamma_{k+1}, k \in [k_{\max} - 1]\}$; $\alpha > 0$; a coefficient $p = \mathbb{E}_{\mathcal{A} \sim \mathbb{P}_{\mathrm{PP}}}[\mathrm{card}(\mathcal{A})]/n$.

**Result:** The FedEM-PP sequence:
$\{\widehat{S}_k, k \in [k_{\max}]\}$

1 Set $V_0 = n^{-1} \sum_{i=1}^n V_{0,i}$
2 **for** $k = 0, \ldots, k_{\max} - 1$ **do**
3   Sample $\mathcal{A}_{k+1} \sim \mathbb{P}_{\mathrm{PP}}$
4   **for** $i \in \mathcal{A}_{k+1}$ **do**
5     *(worker #$i$)*
6     Sample $\mathsf{S}_{k+1,i}$, an approximation of $\bar{s}_i \circ \mathsf{T}(\widehat{S}_k)$
7     Set $\Delta_{k+1,i} = \mathsf{S}_{k+1,i} - V_{k,i} - \widehat{S}_k$
8     Set $V_{k+1,i} = V_{k,i} + \alpha\, \mathrm{Quant}(\Delta_{k+1,i})$.
9     Send $\mathrm{Quant}(\Delta_{k+1,i})$ to the central server
10   **for** $i \notin \mathcal{A}_{k+1}$ **do**
11     *(worker #$i$)*
12     Set $V_{k+1,i} = V_{k,i}$ (no update)
13   *(the central server)*
14   Set
$H_{k+1} = V_k + (np)^{-1} \sum_{i \in \mathcal{A}_{k+1}} \mathrm{Quant}(\Delta_{k+1,i})$
15   Set $\widehat{S}_{k+1} = \widehat{S}_k + \gamma_{k+1} H_{k+1}$   Set
$V_{k+1} = V_k + \alpha n^{-1} \sum_{i \in \mathcal{A}_{k+1}} \mathrm{Quant}(\Delta_{k+1,i})$
  Send $\widehat{S}_{k+1}$ and $\mathsf{T}(\widehat{S}_{k+1})$ to the $n$ workers

---

and line 9). These memories are updated locally: $V_{k+1,i} = V_{k,i} + \alpha\, \mathrm{Quant}(\Delta_{k+1,i})$, at line 8, with $\alpha > 0$ (typically set to $1/(1 + \omega)$ where $\omega$ is defined in A6). On its side, the central server releases an aggregated estimate $\widehat{S}_{k+1}$ of the complete-data sufficient statistics by averaging the quantized difference $(np)^{-1} \sum_{i \in \mathcal{A}_{k+1}} \mathrm{Quant}(\Delta_{k+1,i})$ and by adding $V_k$ (line 14 and line 15). Then, it updates $V_{k+1} = V_k + \alpha n^{-1} \sum_{i=1}^n \mathrm{Quant}(\Delta_{k+1,i})$, see line 15. The final step consists in solving the M-step of the EM algorithm, i.e. in computing $\mathsf{T}(\widehat{S}_{k+1})$ (see A3).

We finally state our assumption on the compression process. We consider a large class of *unbiased* compression operators Quant satisfying a variance bound:

**A6.** *There exists* $\omega \geq 0$ *s.t. for any* $s \in \mathbb{R}^q$: $\mathbb{E}[\mathrm{Quant}(s)] = s$, *and* $\mathbb{E}[\|\mathrm{Quant}(s)\|^2] \leq (1 + \omega)\|s\|^2$.

Intuitively, the stronger the compression is, the larger $\omega$ will be. Remark that if no compression is used, or equivalently for all $s \in \mathbb{R}^q$, $\mathrm{Quant}(s) = s$, then A6 is satisfied with $\omega = 0$. An example

of quantization operator satisfying A6 is the random dithering that can be described as the random operator $\mathrm{Quant} : \mathbb{R}^q \to \mathbb{R}^q$, $\mathrm{Quant}(x) = (1/s_{\mathrm{quant}})\|x\|_r \, \mathrm{sign}(x) \odot \lfloor s_{\mathrm{quant}}(|x|/\|x\|_r) + \xi \rfloor$ where $r \geq 1$ is user-defined, $\xi$ is a uniform random variable on $[0,1]^q$ and $s_{\mathrm{quant}} \in \mathbb{N}^\star$ is the number of levels of roundings; see [17, 2]. This operator satisfies A6 with $\omega = s_{\mathrm{quant}}^{-1} O(q^{1/r} + q^{1/2})$; see [17, Example 1]. Another example, namely the block-$p$-quantization, is provided in the supplemental (see Appendix B). More generally, this assumption is valid for many compression operators, for example resulting in sparsification [see. e.g. 28].

The convergence analysis is under the following assumptions on the oracle $\mathsf{S}_{k+1,i}$: for any $i \in [n]^\star$, the approximations $\mathsf{S}_{k+1,i}$ are unbiased and their conditional variances are uniformly bounded in $k$. For each $k \in \mathbb{N}$, denote by $\mathcal{F}_k$ the $\sigma$-algebra generated by $\{\mathsf{S}_{\ell,i}, \mathcal{A}_\ell; i \in [n]^\star, \ell \in [k]\}$ and including the randomness inherited from the quantization operator $\mathrm{Quant}$ up to iteration $\#k$.

**A 7.** *For all $k \in \mathbb{N}$, conditional to $\mathcal{F}_k$, $\{\mathsf{S}_{k+1,i}\}_{i=1}^n$ are independent. Moreover, for any $i \in [n]^\star$, $\mathbb{E}\left[\mathsf{S}_{k+1,i}|\mathcal{F}_k\right] = \bar{\mathsf{s}}_i \circ \mathsf{T}(\widehat{S}_k)$ and there exists $\sigma_i^2 > 0$ such that for any $k \geq 0$ $\mathbb{E}\left[\left\|\mathsf{S}_{k+1,i} - \bar{\mathsf{s}}_i \circ \mathsf{T}(\widehat{S}_k)\right\|^2 \Big| \mathcal{F}_k\right] \leq \sigma_i^2$.*

A7 covers both the finite-sum setting described in the introduction, and the online setting. In the finite-sum setting, $\bar{\mathsf{s}}_i$ is of the form $m^{-1}\sum_{j=1}^m \bar{\mathsf{s}}_{ij}$. In that case, $\mathsf{S}_{k+1,i}$ can be the sum over a mini-batch $\mathcal{B}_{k+1,i}$ of size $\mathsf{b}$ sampled at random in $[m]^\star$, with or without replacement and independently of the history of the algorithm: we have $\mathsf{S}_{k+1,i} = \mathsf{b}^{-1}\sum_{j\in\mathcal{B}_{k+1,i}} \bar{\mathsf{s}}_{ij} \circ \mathsf{T}(\widehat{S}_k)$. In the online setting, the oracles $\mathsf{S}_{k+1,i}$ come from an online processing of streaming informations; in that case $\mathsf{S}_{k+1,i}$ can be computed from a minibatch of independent examples so that the conditional variance $\sigma_i^2$, which will be inversely proportional to the size of the minibatch, can be made arbitrarily small.

**Reduction of communication complexity for FL.** Reducing the communication cost between workers is a crucial aspect of the FL approach [19]. In gradient based optimization, four techniques have been used to reduce the amount of communication: (i) increasing the minibatch size and reducing the number of iterations, (ii) increasing the number of *local steps* between two communication rounds, (iii) using compression, (iv) sampling clients at each step. Here, we provide a tight analysis of strategies (i), (iii) and (iv) (sampling client is part of PP).

Regarding the interest of performing multiple iterations (ii), as analyzed for example in [21, 27] for the classical gradient settings, we note that: first, from a theoretical standpoint, tradeoffs between larger minibatch and more local iterations are unclear [37]. Secondly, *performing local iterations is not possible in the EM setting*: one iteration of EM is the combination of two steps E and M and the M step, which required the use of the map $T$, is only performed by the central server; this remark is a fundamental specificity of the EM framework (which is not shared by the gradient framework). In applications, we usually do not want $T$ to be available at each local node. However, our work allows to perform multiple local iterations of the E step before communicating with the central server. In algorithm 1, the local statistics $S_{k+1,i}$ are general enough to cover this case; see the comment above on A7.

Finally, as we do not perform local full EM iterations, we do not face the well-identified *client-drift* challenge (in the presence of heterogeneity). Yet, we stress that combining compression and heterogeneity results in other challenges: it is known in the Gradient Descent setting (see e.g. [28, 31]), that heterogeneity strongly hinders convergence in the presence of compression. To alleviate the impact of heterogeneity, we introduce the $V_{k,i}$'s memory-variables.

**Convergence analysis, full participation regime.** In this paragraph, we focus on the *full-participation regime* ($p = 1$): for all $k \in [k_{\max}]^\star$, $\mathcal{A}_k = [n]^*$. We now present in Theorem 1 our key result, from which complexity expressions are derived. The proof is postponed to Appendix C.

**Theorem 1.** *Assume A1 to A7 and set $L^2 := n^{-1}\sum_{i=1}^n L_i^2$, $\sigma^2 := n^{-1}\sum_{i=1}^n \sigma_i^2$. Let $\{\widehat{S}_k, k \in [k_{\max}]\}$ be given by algorithm 1, with $\omega > 0$, $\alpha := (1+\omega)^{-1}$ and $\gamma_k = \gamma \in (0, \gamma_{\max}]$ where*

$$\gamma_{\max} := \frac{v_{\min}}{2L_{\dot{W}}} \wedge \frac{\sqrt{n}}{2\sqrt{2}L(1+\omega)\sqrt{\omega}} . \tag{7}$$

*Denote by $K$ the uniform random variable on $[k_{\max}-1]$. Then, taking $V_{0,i} = \mathsf{h}_i(\widehat{S}_0)$ for all $i \in [n]^\star$:*

$$v_{\min}\left(1 - \gamma\frac{L_{\dot{W}}}{v_{\min}}\right)\mathbb{E}\left[\|\mathsf{h}(\widehat{S}_K)\|^2\right] \leq \frac{1}{\gamma k_{\max}}\left(\mathrm{W}(\widehat{S}_0) - \min\mathrm{W}\right) + \gamma L_{\dot{W}}\frac{1 + 5\omega}{n}\sigma^2 . \tag{8}$$

When there is no compression ($\omega = 0$ so that $\mathrm{Quant}(s) = s$), we prove that the introduction of the random variables $V_{k,i}$'s play no role whatever $\alpha > 0$ and the choice of the $V_{0,i}$'s, and we have for any $\gamma \in (0, 2v_{\min}/L_{\dot{\mathrm{W}}})$ (see (29) in the supplemental)

$$\left(1 - \gamma \frac{L_{\dot{\mathrm{W}}}}{2v_{\min}}\right)\mathbb{E}\left[\|h(\widehat{S}_K)\|^2\right] \leq \frac{1}{\gamma k_{\max}}\left(\mathrm{W}(\widehat{S}_0) - \min \mathrm{W}\right) + \gamma L_{\dot{\mathrm{W}}}\frac{\sigma^2}{n} . \tag{9}$$

Optimizing the learning rate $\gamma$, we derive the following corollary (see the proof in Appendix C).

**Corollary 2** (of Theorem 1). *Choose* $\gamma := \left(\frac{(\mathrm{W}(\widehat{S}_0) - \min \mathrm{W})n}{k_{\max}L_{\dot{\mathrm{W}}}(1+5\omega)\sigma^2}\right)^{1/2} \wedge \gamma_{\max}$. *We get*

$$\mathbb{E}\left[\|h(\widehat{S}_K)\|^2\right] \leq \frac{4}{v_{\min}}\left(\sqrt{\frac{(\mathrm{W}(\widehat{S}_0) - \min \mathrm{W})L_{\dot{\mathrm{W}}}(1+5\omega)\sigma^2}{nk_{\max}}} \vee \frac{(\mathrm{W}(\widehat{S}_0) - \min \mathrm{W})}{\gamma_{\max}k_{\max}}\right) .$$

Theorem 1 and Corollary 2 do not require any assumption regarding the distributional heterogeneity of workers. These results remain thus valid when workers have access to data resulting from different distributions — a widespread situation in FL frameworks. Crucially, without assumptions on the heterogeneity of workers, the convergence of a "naive" implementation of compressed distributed EM (i.e. an implementation without the variables $V_{k,i}$'s) would not converge.

Let us comment the complexity to reach an $\epsilon$-stationary point, and more precisely how the complexity evaluated in terms of the number of optimization steps depend on $\omega, n, \sigma^2$ and $\epsilon$. Since $\mathcal{K}_{\mathrm{Opt}}(\epsilon) = k_{\max}$, from Corollary 2 we have that: $\mathcal{K}_{\mathrm{opt}}(\epsilon) = O\left(\frac{(1+\omega)\sigma^2}{n\epsilon^2}\right) \vee O\left(\frac{1}{\gamma_{\max}\epsilon}\right)$ .

**Maximal learning rate and compression.** The comparison of Theorem 1 with the no compression case (see (9)) shows that compression impacts $\gamma_{\max}$ by a factor proportional to $\sqrt{n}/\omega^{3/2}$ as $\omega$ increases (similar constraints were observed in the risk optimization literature, e.g. in [17, 32]). This highlights two different regimes depending on the ratio $\sqrt{n}/\omega^{3/2}$: if the number of workers $n$ scales at least as $\omega^3$, the maximal learning rate is not impacted by compression; on the other hand, for smaller numbers of workers $n \ll \omega^3$, compression can degrade the maximal learning rate. We highlight this conclusion with a small example in the case of scalar quantization for which $\omega \sim \sqrt{q}/s_{\mathrm{quant}}$: for $q = 10^2$ and $s_{\mathrm{quant}} = 4$ (obtaining a compression rate of a factor 16), the maximal learning rate is almost unchanged if $n \geq 16$.

**Dependency on $\epsilon$.** The complexity $\mathcal{K}_{\mathrm{opt}}(\epsilon)$ is decomposed into two terms scaling respectively as $\sigma^2\epsilon^{-2}$ and $\gamma_{\max}^{-1}\epsilon^{-1}$, the first term being dominant when $\epsilon \to 0$. This observation highlights two different regimes: a *high noise regime* corresponding to $\gamma_{\max}(1+\omega)\sigma^2/(n\epsilon^{-1}) \geq 1$ where the complexity is of order $\sigma^2\epsilon^{-2}$, and a *low noise regime* where $\gamma_{\max}(1+\omega)\sigma^2/(n\epsilon^{-1}) \leq 1$ and the complexity is of order $\gamma_{\max}^{-1}\epsilon^{-1}$. An extreme example of the low noise case is $\sigma^2 = 0$, occurring for example in the finite-sum case (i.e., when $\bar{s}_i = m^{-1}\sum_{j=1}^m \bar{s}_{ij}$) with the oracle $S_{k+1,i} = \bar{s}_i \circ T(\widehat{S}_k)$.

**Impact of compression for $\epsilon$-stationarity.** As mentioned above, the compression simultaneously impacts the maximal learning rate (as in (7)) and the complexity $\mathcal{K}_{\mathrm{opt}}(\epsilon)$. Consequently, the impact of the compression depends on the balance between $\omega, n, \sigma^2$ and $\epsilon$, and we can distinguish four different "main" regimes. In the following tabular, for each of the four situations, we summarize the *increase in complexity* $\mathcal{K}_{\mathrm{opt}}(\epsilon)$ resulting from compression.

| | Complexity regime: (Dominating term in $\mathcal{K}_{\mathrm{opt}}(\epsilon)$) | $\frac{(1+\omega)\sigma^2}{n\epsilon^2}$ | $\frac{1}{\gamma_{\max}\epsilon}$ |
|---|---|---|---|
| $\gamma_{\max}$ regime: (Dominating term in (7)) | Example situation | High noise $\sigma^2$, small $\epsilon$ | Low $\sigma^2$ (e.g., large minibatch) larger $\epsilon$ |
| $\frac{v_{\min}}{2L_{\dot{\mathrm{W}}}}$ | large ratio $n/\omega^3$ | $\times\omega$ | $\times 1$ |
| $\frac{\sqrt{n}}{2\sqrt{2}L(1+\omega)\sqrt{\omega}}$ | low ratio $n/\omega^3$ | $\times\omega$ | $\times\omega^{3/2}/\sqrt{n}$ |

Depending on the situation, the complexity can be multiplied by a factor ranging from 1 to $\omega \vee (\omega^{3/2}/\sqrt{n})$ . Remark that the communication cost of each iteration is typically reduced by compression of a factor at least $\omega$. Moreover, the benefit of compression is most significant in the *low noise* regime and when the maximal learning rate is $v_{\min}/(2L_{\dot{\mathrm{W}}})$ (e.g., when $n$ large enough). We then improve the communication cost of each iteration without increasing the optimization complexity, effectively reducing the communication budget "for free".

Because of space constraints, the results in the PP regime are postponed to Appendix A.

# 3 `VR-FedEM`: Federated EM algorithm with variance reduction

A novel algorithm, called `VR-FedEM` and described by algorithm 2, is derived to additionally incorporate a variance reduction scheme in `FedEM`. It is described in the finite-sum setting when for all $i \in [n]^\star$, $\bar{s}_i := m^{-1} \sum_{j=1}^m \bar{s}_{ij}$: at each iteration $\#(t, k+1)$, the oracle on $\bar{s}_i \circ \mathsf{T}(\widehat{S}_{t,k})$ will use a minibatch $\mathcal{B}_{t,k+1,i}$ of examples sampled at random (with or without replacement) in $[m]^\star$.

The algorithm is decomposed into $k_{\text{out}}$ outer loops (indexed by $t$), each of them having $k_{\text{in}}$ inner loops (indexed by $k$). At iteration $\#(k+1)$ of the inner loops, each worker $\#i$ updates a local statistic $\mathsf{S}_{t,k+1,i}$ based on a minibatch $\mathcal{B}_{t,k+1,i}$ of its own examples $\{\bar{s}_{ij}, j \in \mathcal{B}_{t,k+1,i}\}$ (see Line 8): starting from $\widehat{S}_{t,0,i} := m^{-1} \sum_{j=1}^m \bar{s}_{ij} \circ \mathsf{T}(\widehat{S}_{t,-1})$, $\widehat{S}_{t,k+1,i}$ is defined in such a way that it approximates $m^{-1} \sum_{j=1}^m \bar{s}_{ij} \circ \mathsf{T}(\widehat{S}_{t,k})$ (see Corollary 18). Then, the worker $\#i$ sends to the central server a quantization of $\Delta_{t,k+1,i}$ (see Line 12) which can be seen as an approximation of $\alpha^{-1}\{\mathsf{h}_i(\widehat{S}_{t,k}) - \mathsf{h}_i(\widehat{S}_{t,k-1})\}$ upon noting that the variable $V_{t,k+1,i}$ defined by Line 10 approximates $\mathsf{h}_i(\widehat{S}_{t,k})$ (see Proposition 26). The central server learns the mean value $V_{t,k+1} = n^{-1} \sum_{i=1}^n V_{t,k+1,i}$ (see Line 15 and Lemma 21) and, by adding the quantized quantities, defines a field $H_{t,k+1}$ which approximates $n^{-1} \sum_{i=1}^n \mathsf{h}_i(\widehat{S}_{t,k})$ (see Proposition 24). Line 14 can be seen as a Stochastic Approximation update, with learning rate $\gamma_{t,k+1}$ and mean field $s \mapsto n^{-1} \sum_{i=1}^n \mathsf{h}_i(s)$ (see (6) for the definition of $\mathsf{h}_i$).

The variance reduction is encoded in the definition of $\mathsf{S}_{t,k+1,i}$, Line 8. We

---

**Algorithm 2:** `VR-FedEM`

**Data:** $k_{\text{out}}, k_{\text{in}}, \mathsf{b} \in \mathbb{N}^\star$; for $i \in [n]^\star$, $V_{1,0,i} \in \mathbb{R}^q$;
$\widehat{S}_{\text{init}} \in \mathbb{R}^q$; a positive sequence
$\{\gamma_{t,k+1}, t \in [k_{\text{out}}]^\star, k \in [k_{\text{in}} - 1]\}$; $\alpha > 0$

**Result:** sequence: $\{\widehat{S}_{t,k}, t \in [k_{\text{out}}]^\star, k \in [k_{\text{in}}]\}$

1   $\widehat{S}_{1,0} = \widehat{S}_{1,-1} = \widehat{S}_{\text{init}}$, $V_{1,0} = n^{-1} \sum_{i=1}^n V_{1,0,i}$
2   **for** $i = 1, \ldots, n$ **do**
3     $\lfloor \mathsf{S}_{1,0,i} = \frac{1}{m} \sum_{j=1}^m \bar{s}_{ij} \circ \mathsf{T}(\widehat{S}_{\text{init}})$
4   **for** $t = 1, \ldots, k_{\text{out}}$ **do**
5    **for** $k = 0, \ldots, k_{\text{in}} - 1$ **do**
6     **for** $i = 1, \ldots, n$ (worker $\#i$, locally) **do**
7      Sample at random a batch $\mathcal{B}_{t,k+1,i}$ of size $\mathsf{b}$ in $[m]^\star$
8      Set $\mathsf{S}_{t,k+1,i} = \mathsf{S}_{t,k,i} +$
      $\mathsf{b}^{-1} \sum_{j \in \mathcal{B}_{t,k+1,i}} \left( \bar{s}_{ij} \circ \mathsf{T}(\widehat{S}_{t,k}) - \bar{s}_{ij} \circ \mathsf{T}(\widehat{S}_{t,k-1}) \right)$
9      Set $\Delta_{t,k+1,i} = \mathsf{S}_{t,k+1,i} - \widehat{S}_{t,k} - V_{t,k,i}$
10      Set $V_{t,k+1,i} = V_{t,k,i} + \alpha \operatorname{Quant}(\Delta_{t,k+1,i})$.
11      Send $\operatorname{Quant}(\Delta_{t,k+1,i})$ to the central server
12     *(the central server)*
13     Set $H_{t,k+1} = V_{t,k} + n^{-1} \sum_{i=1}^n \operatorname{Quant}(\Delta_{t,k+1,i})$
14     Set $\widehat{S}_{t,k+1} = \widehat{S}_{t,k} + \gamma_{t,k+1} H_{t,k+1}$
15     Set $V_{t,k+1} = V_{t,k} + \alpha n^{-1} \sum_{i=1}^n \operatorname{Quant}(\Delta_{t,k+1,i})$
16     Send $\widehat{S}_{t,k+1}$ and $\mathsf{T}(\widehat{S}_{t,k+1})$ to the $n$ workers
17   $\widehat{S}_{t+1,0} = \widehat{S}_{t+1,-1} = \widehat{S}_{t,k_{\text{in}}}$
18   $V_{t+1,0} = V_{t,k_{\text{in}}}$
19   **for** $i = 1, \ldots, n$ **do**
20    $\mathsf{S}_{t+1,0,i} = \frac{1}{m} \sum_{j=1}^m \bar{s}_{ij} \circ \mathsf{T}(\widehat{S}_{t+1,0})$
21    $V_{t+1,0,i} = V_{t,k_{\text{in}},i}$

---

have $\mathsf{S}_{t,k+1,i} = \mathsf{b}^{-1} \sum_{j \in \mathcal{B}_{t,k+1,i}} \bar{s}_{ij} \circ \mathsf{T}(\widehat{S}_{t,k}) + \Upsilon_{t,k+1,i}$. The first term is the natural approximation of $\bar{s}_i \circ \mathsf{T}(\widehat{S}_{t,k})$ based on a minibatch $\mathcal{B}_{t,k+1,i}$. Conditionally to the past, $\Upsilon_{t,k+1,i}$ is correlated to the first term and biased, but its bias is canceled at the beginning of each outer loop (see Line 20 and Appendix E.3.2): $\Upsilon_{t,k+1,i}$ defines a *control variate*. Such a variance reduction technique was first proposed in the stochastic gradient setting [30, 9, 36] and then extended to the EM setting [10, 12]. At the end of each outer loop, the local approximations $\mathsf{S}_{t+1,0,i}$ are initialized to the full sum $m^{-1} \sum_{j=1}^m \bar{s}_{ij} \circ \mathsf{T}(\widehat{S}_{t,k_{\text{in}}})$ (see Line 20) thus canceling the bias of $\mathsf{S}_{.,i}$ (see Proposition 17).

When there is a single worker and no compression is used ($n = 1$, $\omega = 0$), `VR-FedEM` reduces to `SPIDER-EM`, which has been shown to be rate optimal for smooth, non-convex finite-sum optimization [10]. Theorem 3 studies the FL setting ($n \geq 1$ and $\omega \geq 0$): it establishes a finite time control of convergence in expectation for `VR-FedEM`. Assumptions A5 and A7 are replaced with A8.

**A8.** *For any $i \in [n]^\star$ and $j \in [m]^\star$, the conditional expectations $\bar{s}_{ij}(\theta)$ are well defined for any $\theta \in \Theta$, and there exists $L_{ij}$ such that for any $s, s' \in \mathbb{R}^q$, $\|(\bar{s}_{ij} \circ \mathsf{T}(s) - s) - (\bar{s}_{ij} \circ \mathsf{T}(s') - s')\| \leq L_{ij}\|s - s'\|$.*

**Theorem 3.** *Assume A1 to 3, A4, A6 and A8. Set $L^2 := n^{-1}m^{-1}\sum_{i=1}^{n}\sum_{j=1}^{m}L_{ij}^2$. Let $\{\widehat{S}_{t,k}, t \in [k_{\mathrm{out}}]^\star, k \in [k_{\mathrm{in}}-1]\}$ be given by algorithm 2 run with $\alpha := 1/(1+\omega)$, $V_{1,0,i} := \mathsf{h}_i(\widehat{S}_{1,0})$ for any $i \in [n]^\star$, $\mathsf{b} := \lceil \frac{k_{\mathrm{in}}}{(1+\omega)^2}\rceil$ and*

$$\gamma_{t,k} = \gamma := \frac{v_{\min}}{L_{\dot{\mathrm{W}}}}\left(1 + 4\sqrt{2}\frac{v_{\max}}{L_{\dot{\mathrm{W}}}}\frac{L}{\sqrt{n}}(1+\omega)\left(\omega + \frac{1+10\omega}{8}\right)^{1/2}\right)^{-1}. \tag{10}$$

*Let $(\tau, K)$ be the uniform random variable on $[k_{\mathrm{out}}]^\star \times [k_{\mathrm{in}}-1]$, independent of $\{\widehat{S}_{t,k}, t \in [k_{\mathrm{out}}]^\star, k \in [k_{\mathrm{in}}]\}$. Then, it holds*

$$\mathbb{E}\left[\|H_{\tau,K+1}\|^2\right] \le \frac{2\left(\mathbb{E}\left[\,\mathrm{W}(\widehat{S}_{1,0})\right] - \min \mathrm{W}\right)}{v_{\min}\gamma k_{\mathrm{in}}k_{\mathrm{out}}}, \tag{11}$$

$$\mathbb{E}\left[\|\mathsf{h}(\widehat{S}_{\tau,K})\|^2\right] \le 2\left(1 + \gamma^2\frac{L^2(1+\omega)^2}{n}\right)\mathbb{E}\left[\|H_{\tau,K+1}\|^2\right]. \tag{12}$$

The proof is postponed to Appendix E. This result is a consequence of the more general Proposition 25. We make the following comments:

1. Eq. (11) provides the convergence of $\mathbb{E}\left[\|H_{\tau,K+1}\|^2\right]$, and Eq. (12) ensures that the quantity of interest $\mathbb{E}[\|\mathsf{h}(\widehat{S}_{\tau,K})\|^2]$ is controlled by $\mathbb{E}[\|H_{\tau,K+1}\|^2]$. We observe that $2(1 + \gamma^2\frac{L^2(1+\omega)^2}{n})$ is uniformly bounded w.r.t. $\omega$ as, by (10), $\gamma^2 = O_{\omega\to\infty}(\omega^{-3})$.
2. Up to our knowledge, this is the first result on Federated EM, that leverages advanced variance reduction techniques, while being robust to distribution heterogeneity (the theorem is valid without any assumption on heterogeneity) and while reducing the communication cost.
3. Without compression ($\omega = 0$) and in the single-worker case ($n = 1$), Fort et al. [10] use $k_{\mathrm{in}} = \mathsf{b}$: we recover this result as a particular case. When $n > 1$ and $\omega > 0$, the recommended batch size $\mathsf{b}$ decreases as $1/(1+\omega)^2$.

**Convergence rate and optimization complexity.** Our step-size $\gamma$ is chosen constant and *independent* of $k_{\mathrm{in}}, k_{\mathrm{out}}$. Indeed, contrary to Theorem 1, there is no Bias-Variance trade-off (as typically observed with variance reduced methods), and the optimal choice of $\gamma$ is the largest one to ensure convergence. Consequently, since the number of optimization steps is $k_{\mathrm{out}}k_{\mathrm{in}}$, we have $\mathcal{K}_{\mathrm{opt}}(\epsilon) = O(\frac{1}{\gamma\epsilon})$.

**Impact of compression on the learning rate and $\epsilon$-stationarity.** The compression constant $\omega$ does not directly appear in (11), but impacts the value of $\gamma$. Two different regimes appear:

1. if $4\sqrt{2}\frac{v_{\max}}{L_{\dot{\mathrm{W}}}}\frac{L}{\sqrt{n}}(1+\omega)\left(\omega + \frac{1+10\omega}{8}\right)^{1/2} \ll 1$ (i.e. we focus on the large $\omega, n$ asymptotics when $\omega^3 \ll n$), then $\gamma \simeq \frac{v_{\min}}{L_{\dot{\mathrm{W}}}}$ has nearly the same value as without compression [10]. The complexity is then similar to the one of SPIDER-EM [10], with a smaller communication cost. The gain from compression is maximal in this regime.
2. if $4\sqrt{2}\frac{v_{\max}}{L_{\dot{\mathrm{W}}}}\frac{L}{\sqrt{n}}(1+\omega)\left(\omega + \frac{1+10\omega}{8}\right)^{1/2} \gg 1$ (i.e. we focus on the large $\omega, n$ asymptotics when $\omega^3 \gg n$), then $\gamma = O\left(\frac{v_{\min}\sqrt{n}}{v_{\max}L\omega^{3/2}}\right)$ is strictly smaller than without compression. The optimization complexity is then higher to the one of SPIDER-EM[1] (by a factor proportional to $\omega^{3/2}/\sqrt{n}$) with a smaller communication cost (typically at least $\omega$ times less bits exchanged per iteration). The overall trade-off thus depends on the comparison between $\omega$ and $n$.

We summarize these two regimes in this tabular, focusing on the large $n$, large $\omega$ asymptotic regimes. For the two regimes, we indicate the *increase in complexity* $\mathcal{K}_{\mathrm{opt}}(\epsilon)$ resulting from compression.

| $\gamma$ regime: (Dominating term in (10)) | Complexity : $1/(\gamma\epsilon)$ | |
|---|---|---|
| | Example situation | |
| $v_{\min}/L_{\dot{\mathrm{W}}}$ | large ratio $n/\omega^3$ | $\times 1$ |
| $v_{\min}\sqrt{n}/(v_{\max}L\omega^{3/2})$ | low ratio $n/\omega^3$ | $\times\omega^{3/2}/\sqrt{n}$ |

We provide a discussion on *computed conditional expectations* complexity $\mathcal{K}_{\mathrm{CE}}$ in Appendix E.2.

---

[1]As a corollary of [10, Theorem 2], the optimization complexity of SPIDER-EM is $k_{\mathrm{out}} + k_{\mathrm{in}}k_{\mathrm{out}}$ that is $\epsilon^{-1}$ in order to reach $\epsilon$-stationarity.

## 4   Numerical illustrations

In this section, we illustrate the performance of FedEM and VR-FedEM applied to inference in Gaussian Mixture Models (GMM), on a synthetic data set and on the MNIST data set. We also present an application to Federated missing data imputation with the analysis of the eBird data set [34, 1].

**Synthetic data.**   The synthetic data are from the following GMM model: for all $\ell \in [N]^\star$ and $g \in \{0, 1\}$, $\mathbb{P}(Z_\ell = g) = \pi_g$; and conditionally to $Z_\ell = g$, $Y_\ell \sim \mathcal{N}_2(\mu_g, \Sigma)$. The $2 \times 2$ covariance matrix $\Sigma$ is known, and the parameters to be fitted are the weights $(\pi_0, \pi_1)$ and the expectations $(\mu_0, \mu_1)$. The total number of examples is $N = 10^4$, the number of agents is $n = 10^2$, and the probability of participation of servers is $p = 0.75$. FedEM and VR-FedEM are run with $\gamma = 10^{-2}$, $\omega = 1$ and $\alpha = 10^{-2}$. For FedEM, we consider the finite-sum setting when $\bar{\mathsf{s}}_i = m^{-1} \sum_{j=1}^m \bar{\mathsf{s}}_{ij}$ with $m = 10^2$; the oracle $\mathsf{S}_{k+1,i}$ is obtained by a sum over a minibatch of $\mathsf{b} = 20$ examples. For VR-FedEM, we set $\mathsf{b} = 5$ and $k_{\mathrm{in}} = 20$. We run the two algorithms for 500 epochs (one epoch corresponds to $N$ conditional expectation evaluations $\bar{\mathsf{s}}_{ij}$). Figure 1 shows a trajectory of $\|H_k\|^2$ given by FedEM (and $\|H_{t,k}\|^2$ given by VR-FedEM), along with the theoretical value of the mean field $\|\mathsf{h}(\widehat{S}_k)\|^2$ for FedEM (and $\|\mathsf{h}(\widehat{S}_{t,k})\|^2$ for VR-FedEM). The results illustrate the variance reduction, and gives insight on the variability of the trajectories resulting from the two algorithms.

**MNIST Data set.**   We perform a similar experiment on the MNIST dataset to illustrate the behaviour of FedEM and VR-FedEM on a GMM inference problem with real data. The dataset consists of $N = 7 \times 10^4$ images of handwritten digits, each with 784 pixels. We pre-process the dataset by removing 67 uninformative pixels (which are always zero across all images) to obtain $d = 717$ pixels per image. Second, we apply principal component analysis to reduce the data dimension. We keep the $d_{\mathrm{PC}} = 20$ principal components of each observation. These $N$ preprocessed observations are distributed at random across $n = 10^2$ servers, each containing $m = 700$ observations. We estimate a $\mathbb{R}^{d_{\mathrm{PC}}}$-multivariate GMM model with $G = 10$ components. Details on the multivariate Gaussian mixture model are given in the supplementary material (see Appendix F). Here again, $\bar{\mathsf{s}}_i$ is a sum over the $m$ examples available at server $\#i$; the minibatches are independent and sampled at random in $[m]^\star$ with replacement; we choose $\mathsf{b} = 20$ and the step size is constant and set to $\gamma = 10^{-3}$. The same initial value $\widehat{S}_{\mathrm{init}}$ is used for all experiments: we set $\widehat{S}_{\mathrm{init}} := \bar{s}(\pi^0, \mu^0, \widehat{\Sigma}^0)$, where $\pi_g^0 = 1/G$ for all $g \in [G]^\star$, the expectations $\mu_g^0$ are sampled uniformly at random among the available examples, and $\widehat{\Sigma}^0$ is the empirical covariance matrix of the $N$ examples. Figure 3 shows the sequence of parameter estimates for the weights and the squared norm of the mean field $\|H_k\|^2$ for FedEM (resp. $\|H_{t,k}\|^2$ for VR-FedEM ) vs the number of epochs.

**Federated missing values imputation for citizen science.**   We develop FedMissEM, a special instance of FedEM designed to missing values imputation in the federated setting; we apply it to the analysis of part of the *eBird* data base [34, 1], a citizen science smartphone application for biodiversity monitoring. In *eBird*, citizens record wildlife observations, specifying the ecological site they visited, the date, the species and the number of observed specimens. Two major challenges occur: (i) ecological sites are visited irregularly, which leads to missing values and (ii) non-professional observers have heterogeneous wildlife counting schemes.

● *Model and the FedMissEM algorithm.* $I$ observers participate in the programme, there are $J$ ecological sites and $L$ time stamps. Each observer $\#i$ provides a $J \times L$ matrix $X^i$ and a subset of indices $\Omega^i \subseteq [J]^\star \times [L]^\star$. For $j \in [J]^\star$ and $\ell \in [L]^\star$, the variable $X_{j\ell}^i$ encodes the observation that would be collected by observer $\#i$ if the site $\#j$ were visited at time stamp $\#\ell$; since there are unvisited sites, we denote by $Y^i := \{X_{j\ell}^i, (j, \ell) \in \Omega^i\}$ the set of observed values and $Z^i := \{X_{j\ell}^i, (j, \ell) \notin \Omega^i\}$ the set of unobserved values. The statistical model is parameterized by a matrix $\theta \in \mathbb{R}^{J \times L}$, where $\theta_{j\ell}$ is a scalar parameter characterizing the distribution of species individuals at site $j$ and time stamp $\ell$. For instance, $\theta_{j\ell}$ is the log-intensity of a Poisson distribution when the observations are count data or the log-odd of a binomial model when the observations are presence-absence data. This model could be extended to the case observers $\#i$ and $\#i'$ count different number of specimens on average at the same location and time stamp, because they do not have access to the same material or do not have the same level of expertise: heterogeneity between observers could be modeled by using different parameters for each individual $\#i$ say $\theta^i \in \mathbb{R}^{J \times L}$. Here, we consider the case when $\theta_{j\ell}^i = \theta_{j\ell}$ for all $(j, \ell) \in [J]^\star \times [L]^\star$ and $i \in [I]^\star$. We further assume that the entries $\{X_{j\ell}^i, i \in [I]^\star, j \in [J]^\star, \ell \in [L]^\star\}$ are independent with a distribution from an exponential

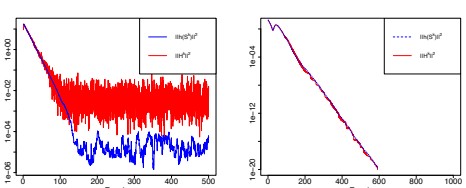
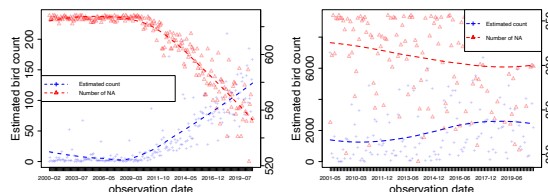

Figure 1: Trajectory of `FedEM` vs the number of epochs (left; blue line: $\|h(\widehat{S}^k)\|^2$; red line: $\|H_k\|^2$) and of `VR-FedEM` (right; dashed blue line: $\|h(\widehat{S}^k)\|^2$; solid red line: $\|H_{t,k}\|^2$).

Figure 2: Estimated temporal trends for Common Buzzard (Left) and Mallard (right). Blue crosses: estimated monthly counts; Red triangles: number of missing values. Dotted lines: LOESS regressions for the estimated counts (blue) and the number of missing values (red).

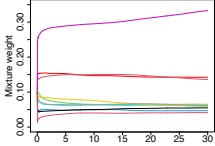
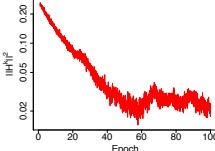
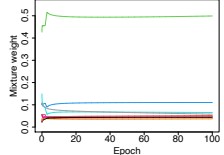
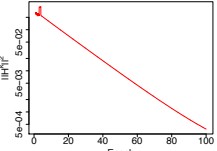

Figure 3: [Left to right] For `FedEM` : Evolution of the estimates of the weights $\pi_\ell$ for $\ell \in [G]^\star$ vs the number of epochs (first plot) and Evolution of the squared norm of the mean field $\|H_k\|^2$ vs the number of epochs (second plot). Then, the same things for `VR-FedEM` (third and fourth plots).

family with respect to some reference measure $\nu$ on $\mathbb{R}$ of the form: $x \mapsto \rho(x) \exp\{x\theta_{j\ell} - \psi(\theta_{j\ell})\}$. Algorithm 7 in Appendix F.2 provides details on the model, and the pseudo-code for `FedMissEM`.

• *Application to eBird data analysis.* We apply `FedMissEM` to the analysis of part of the *eBird* data base [34, 1] of field observations reported in France by $I = 2,465$ observers, across $J = 9,721$ sites and at $L = 525$ monthly time points. We analyze successively two data sets corresponding to observations of two relatively common species: the Common Buzzard and the Mallard. These subsamples correspond respectively to $N = 5,980$ and $N = 12,185$ field observations. The $I$ field observers are randomly assigned into $n = 10$ groups (the observations of the field observers from the group $c \in [n]^\star$ are allocated to the server $\#c$). For $c \in [n]^\star$, server $c$ contains $N_c$ observations; in our two examples, $N_c$ ranges between 400 and 1,500. We run `FedMissEM` for 150 epochs; with $\gamma = 10^{-4}$, $\alpha = 10^{-3}$, $b = 10^2$, a rank $r = 2$ and $\lambda = 0$; for the distribution of the variables $X_{j\ell}^i$, we use a Gaussian distribution with unknown expectation $\theta_{j\ell}$ and variance 1. We recover aggregated temporal trends at the national French level for these two bird species by summing the estimated counts across ecological sites, for each time stamp; the trends are displayed in Figure 2, along with a locally estimated scatterplot smoothing (LOESS).

## 5 Conclusions

We introduced `FedEM` which is, to the best of our knowledge, the first algorithm implementing EM in a FL setting, and handles compression of exchanged information, data heterogeneity and partial participation. We further extended it to incorporate a variance reduction scheme, yielding `VR-FedEM`. We derived complexity bounds which highlight the efficiency of the two algorithms, and illustrated our claims with numerical simulations, as well as an application to biodiversity monitoring data. In a simultaneously published work, Marfoq et al. [25] consider a different Federated EM algorithm, in order to address the personalization challenge by considering a mixture model. Under the assumption that each local data distribution is a mixture of unknown underlying distributions, their algorithm computes a model corresponding to each distribution. On the other hand, we focus on the curved exponential family, with variance reduction, partial participation and compression and on limiting the impact of heterogeneity, but do not address personalization.

**Acknowledgments** The work of A. Dieuleveut and E. Moulines is partially supported by ANR-19-CHIA-0002-01 /chaire SCAI, and Hi!Paris. The work of G. Fort is partially supported by the Fondation Simone et Cino del Duca under the project OpSiMorE.

**Broader Impact of this work** This work is mostly theoretical, and we believe it does not currently present any direct societal consequence. However, the methods described in this paper can be used to train machine learning models which could themselves have societal consequences. For instance, the deployment of machine learning models can suffer from gender and racial bias, or amplify existing inequalities.

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
