# Supplementary materials for "Federated Expectation Maximization with heterogeneity mitigation and variance reduction"

This supplementary material is organized as follows. Appendix A extends the results obtained in Theorem 1 to the Partial Participation regime. Appendix B contains additional details on compression mechanisms satisfying A6, including an example of admissible quantization operator. Appendix C contains the pseudo-code for algorithm FedEM in the full participation regime case, and the proof of Theorem 1 – including necessary technical lemmas. Appendix D contains details concerning the extension to partial participation of the workers and the proof of Theorem 4. Appendix E is devoted to the proof of Theorem 3 concerning the convergence of VR-FedEM and necessary technical results; it also contains a discussion on the complexity of VR-FedEM in terms of conditional expectations evaluations. Finally, Appendix F contains additional details about the latent variable models used in the numerical section, as well as the pseudo code for FedMissEM.
Note that, in order to make our numerical results reproducible, code is also provided as supplementary material.

**Notations**   For two vectors $a, b \in \mathbb{R}^q$, $\langle a, b \rangle$ is the Euclidean standard scalar product, and $\| \cdot \|$ denotes the associated norm. For $r \geq 1$, $\|a\|_r$ is the $\ell_r$-norm of a vector $a$. The Hadamard product $a \odot b$ denotes the entrywise product of the two vectors $a, b$. By convention, vectors are column-vectors. For a matrix $A$, $A^\top$ denotes its transpose and $\|A\|_F$ is its Frobenius norm. For a positive integer $n$, set $[n]^\star := \{1, \cdots, n\}$ and $[n] := \{0, \cdots, n\}$. The set of non-negative integers (resp. positive) is denoted by $\mathbb{N}$ (resp. $\mathbb{N}^\star$). The minimum (resp. maximum) of two real numbers $a, b$ is denoted by $a \wedge b$ (resp. $a \vee b$). We will use the Bachmann-Landau notation $a(x) = O(b(x))$ to characterize an upper bound of the growth rate of $a(x)$ as being $b(x)$.
We denote by $\mathcal{K}_p(\mu, \Sigma)$ the Gaussian distribution in $\mathbb{R}^p$, with expectation $\mu$ and covariance matrix $\Sigma$.

## A   Results for FedEM with partial participation and compression.

In this paragraph, we extend the results of Theorem 1 to the *Partial Participation* (PP) regime, in which only a fraction of the workers participate to the training at each step of the learning process. This is a key feature in the FL framework, as individuals may not always be available or willing to participate [27]. To analyze the convergence in this situation, we make the following assumption.

**A9.** *For all* $k \in [k_{\max} - 1]$, $\mathcal{A}_{k+1} := \{i \in [n]^\star \text{ s.t. } B_{k+1,i} = 1\}$ *where the random variables* $B_{k+1,i}$ *for* $i \in [n]^\star$ *and* $k \in [k_{\max} - 1]$ *are independent Bernoulli random variables with success probability* $p \in (0, 1)$.

This assumption is standard in the FL literature [33, 35, 31], and can easily be extended to worker dependent probabilities of participation [16].

**Usage of the control variates** $(V_{k,i})_{i \in [n]^*}$ **with PP.** We have $V_k = n^{-1} \sum_{i=1}^n V_{k,i}$ for all $k \geq 0$ (see Proposition 12) even when the workers are not all active at iteration $\#k$. A noteworthy point is that, upon receiving $\text{Quant}(\Delta_{k+1,i})$ for all $i \in \mathcal{A}_{k+1}$, the central server computes

$$H_{k+1} = V_k + (np)^{-1} \sum_{i \in \mathcal{A}_{k+1}} \text{Quant}(\Delta_{k+1,i})$$

and *not*

$$(np)^{-1} \sum_{i \in \mathcal{A}_{k+1}} \left( V_{k,i} + \text{Quant}(\Delta_{k+1,i}) \right).$$

Though the later solution may appear more natural, it would actually not only require to store all values $V_{k,i}$ for $i \in [n]^*$ on the central server, but also impair convergence in the heterogeneous setting. Indeed, even in the *uncompressed* regime, in which $\text{Quant}(\Delta_{k+1,i}) = \Delta_{k+1,i}$, our algorithm differs from a naive implementation of a distributed EM: FedEM computes

$$H_{k+1} = V_k - (np)^{-1} \sum_{i \in \mathcal{A}_{k+1}} V_{k,i} + (np)^{-1} \sum_{i \in \mathcal{A}_{k+1}} \left( \mathsf{S}_{k+1,i} - \widehat{S}_k \right)$$

while a naive distributed EM would compute

$$H_{k+1}^{\mathrm{dEM}} := (np)^{-1} \sum_{i \in \mathcal{A}_{k+1}} \left( \mathsf{S}_{k+1,i} - \widehat{S}_k \right) \ .$$

Such an update $H_{k+1}^{\mathrm{dEM}}$ is expected not to be robust to data heterogeneity as proved in [31] for the Stochastic Gradient algorithm in the FL setting.

The following theorem extends Theorem 1 to the partial participation regime. Its proof is in Appendix D.

**Theorem 4.** *Assume A1 to A9 and set $L^2 := n^{-1} \sum_{i=1}^n L_i^2$, $\sigma^2 := n^{-1} \sum_{i=1}^n \sigma_i^2$. Let $\{\widehat{S}_k, k \in [k_{\max}]\}$ be given by algorithm 1, run with $\alpha := (1+\omega)^{-1}$ and $\gamma_k = \gamma \in (0, \gamma_{\max}]$, where*

$$\gamma_{\max} := \frac{v_{\min}}{2L_{\dot{\mathrm{W}}}} \wedge \frac{p\sqrt{n}}{2\sqrt{2}L(1+\omega)\sqrt{\omega + (1-p)(1+\omega)/p}} \ .$$

*Denote by $K$ the uniform random variable on $[k_{\max} - 1]$. Then, taking $V_{0,i} := \mathsf{h}_i(\widehat{S}_0)$ for $i \in [n]^\star$, we get*

$$v_{\min}\left(1 - \gamma \frac{L_{\dot{\mathrm{W}}}}{v_{\min}}\right) \mathbb{E}\left[\|\mathsf{h}(\widehat{S}_K)\|^2\right] \leq \frac{\left(\mathrm{W}(\widehat{S}_0) - \min \mathrm{W}\right)}{\gamma k_{\max}} + \gamma L_{\dot{\mathrm{W}}} \frac{1 + 5\left(\omega + (1-p)(1+\omega)/p\right)}{n} \sigma^2 \ .$$

The above expressions can be simplified upon noting that $\omega + (1-p)(1+\omega)/p \leq (1+\omega)/p$. When $p = 1$, Theorem 1 and Theorem 4 coincide. More generally, Theorem 4 highlights that partial participation impacts both the limiting variance (which increases by a factor proportional to $p^{-1}$) and the maximal learning rate.

## B  An example of quantization mechanisms: the block-$p$-quantization

In this section, we recall the definition of a common lossy data compression mechanism in FL (see, e.g. [28]), called block-$p$-quantization, and demonstrate that such quantizations satisfy the assumptions required to derive our theoretical results.

**Block-$p$-quantization.**  Let $x \in \mathbb{R}^q$. Choose $\{q_\ell, \ell \in [m]^\star\}$ a sequence of positive integers such that $\sum_{\ell=1}^m q_\ell = q$; and $p \in \mathbb{N}^\star$. For $x \in \mathbb{R}^q$, we define the block partition

$$x = \begin{bmatrix} x_{(1)} \\ \cdots \\ x_{(m)} \end{bmatrix}, \ x_{(l)} \in \mathbb{R}^{q_\ell} \text{ for all } \ell \in [m]^\star.$$

For all $\ell \in [m]^\star$, set

$$\hat{X}_{(\ell)} := \|x_{(\ell)}\|_p \begin{bmatrix} \mathrm{sign}(x_{(\ell),1}) \\ \cdots \\ \mathrm{sign}(x_{(\ell),q_\ell}) \end{bmatrix} \odot \begin{bmatrix} U_{\ell,1} \\ \cdots \\ U_{\ell,q_\ell} \end{bmatrix} \qquad U_{\ell,j} \overset{indep}{\sim} \mathcal{B}\left(\frac{|x_{(\ell),j}|}{\|x_{(\ell)}\|_p}\right) \ , \qquad (13)$$

where $x_{(\ell)} = (x_{(\ell),1}, \cdots, x_{(\ell),q_\ell})^\top \in \mathbb{R}^{q_\ell}$ and $\mathcal{B}(u)$ denotes the Bernoulli random variable with success probability $u$. The block-$p$-quantization operator $\mathrm{Quant} : \mathbb{R}^q \to \mathbb{R}^q$ is defined by

$$\mathrm{Quant}(x) := \begin{bmatrix} \hat{X}_{(1)} \\ \cdots \\ \hat{X}_{(m)} \end{bmatrix} \ . \qquad (14)$$

The following Lemma ensures the block-$p$-quantization operator $\mathrm{Quant}$ satisfies the assumption A6 on the compression mechanism required by Theorem 1, Theorem 4 and Theorem 3.

**Lemma 5.** *Let $p \in \mathbb{N}^\star$ and $\{q_\ell, \ell \in [m]^\star\}$ be positive integers such that $\sum_{\ell=1}^m q_\ell = q$. For any $x \in \mathbb{R}^q$, we have*

$$\mathbb{E}\left[\mathrm{Quant}(x)\right] = x \ , \qquad \mathbb{E}\left[\|\mathrm{Quant}(x) - x\|^2\right] = \sum_{\ell=1}^m \left(\|x_{(\ell)}\|_1 \|x_{(\ell)}\|_p - \|x_{(\ell)}\|^2\right) \ ,$$

*where $\mathrm{Quant}$ is the block-$p$-quantization operator defined in (13) and (14). Thus, A6 holds. In particular, for $p = 2$, we may take $\omega = \max_{\ell \in [m]^*}(\sqrt{q_\ell} - 1)$.*

*Proof.* We start by noticing that, for all $\ell \in [m]^\star$, $(\mathrm{Quant}(x))_{(\ell)} = \hat{X}_{(\ell)}$. Furthermore,

$$
\mathbb{E}\left[\hat{X}_{(\ell)}\right] = \|x_{(\ell)}\|_p \begin{bmatrix} \mathrm{sign}(x_{(\ell),1}) \\ \cdots \\ \mathrm{sign}(x_{(\ell),q_\ell}) \end{bmatrix} \odot \begin{bmatrix} \mathbb{E}\left[U_{\ell,1}\right] \\ \cdots \\ \mathbb{E}\left[U_{\ell,q_\ell}\right] \end{bmatrix} = \|x_{(\ell)}\|_p \begin{bmatrix} \mathrm{sign}(x_{(\ell),1}) \\ \cdots \\ \mathrm{sign}(x_{(\ell),q_\ell}) \end{bmatrix} \odot \begin{bmatrix} \frac{|x_{(\ell),1}|}{\|x_{(\ell)}\|_p} \\ \cdots \\ \frac{|x_{(\ell),q_\ell}|}{\|x_{(\ell)}\|_p} \end{bmatrix}
$$

$$
= \begin{bmatrix} \mathrm{sign}(x_{(\ell),1}) \\ \cdots \\ \mathrm{sign}(x_{(\ell),q_\ell}) \end{bmatrix} \odot \begin{bmatrix} |x_{(\ell),1}| \\ \vdots \\ |x_{(\ell),q_\ell}| \end{bmatrix} = \begin{bmatrix} x_{(\ell),1} \\ \vdots \\ x_{(\ell),q_\ell} \end{bmatrix} = x_{(\ell)} \ ,
$$

which concludes the proof of the first statement. To prove the second statement, we write

$$
\|\mathrm{Quant}(x) - x\|^2 = \sum_{\ell=1}^m \|\hat{X}_{(\ell)} - x_{(\ell)}\|^2 = \sum_{\ell=1}^m \|x_{(\ell)}\|_p^2 \sum_{j=1}^{q_\ell} \left(U_{\ell,j} - \mathbb{E}\left[U_{\ell,j}\right]\right)^2 \ .
$$

Since $U_{\ell,j}$ is a Bernouilli random variable with parameter $|x_{(\ell),j}|/\|x_{(\ell)}\|_p$, it holds that

$$
\mathbb{E}\left[\left(U_{\ell,j} - \mathbb{E}\left[U_{\ell,j}\right]\right)^2\right] = \frac{|x_{(\ell),j}|\left(\|x_{(\ell)}\|_p - |x_{(\ell),j}|\right)}{\|x_{(\ell)}\|_p^2} \ .
$$

Hence

$$
\mathbb{E}\left[\|\mathrm{Quant}(x) - x\|^2\right] = \sum_{\ell=1}^m \sum_{j=1}^{q_\ell} \left\{|x_{(\ell),j}|\left(\|x_{(\ell)}\|_p - |x_{(\ell),j}|\right)\right\}
$$

$$
= \sum_{\ell=1}^m \left(\|x_{(\ell)}\|_1\|x_{(\ell)}\|_p - \|x_{(\ell)}\|^2\right) \ ,
$$

which proves the second statement. In the particular case where $p = 2$, using the fact that $\|x_{(\ell)}\|_1 \leq \sqrt{q_\ell}\|x_{(\ell)}\|$, we obtain that

$$
\mathbb{E}\left[\|\mathrm{Quant}(x) - x\|^2\right] \leq \sum_{\ell=1}^m (\sqrt{q_\ell} - 1)\|x_{(\ell)}\|^2 \leq \max_{\ell \in [m]^*}(\sqrt{q_\ell} - 1)\|x\|^2,
$$

which concludes the proof. □

## C Convergence analysis of `FedEM`

This section contains all the elements to derive the convergence analysis of `FedEM` developed in Section 2 in the full participation regime. The analysis is organized as follows. First, Appendix C.1 gives the pseudo code of the `FedEM` algorithm; Appendix C.2 introduces rigorous definitions for filtrations and a technical Lemma, and Appendix C.3 presents preliminary results. Then, the proof of Theorem 1 is given in Appendix C.4 and the proof of Corollary 2 is in Appendix C.5.

The assumptions A1 to A3 are assumed throughout this section.

## C.1 Pseudo code of the `FedEM` algorithm

For the sake of completeness of the supplementary material, we start by recalling the pseudo code which defines the `FedEM` sequence in the **full participation regime.** It is given in algorithm 3 below.

---

**Algorithm 3:** `FedEM`

---

**Data:** $k_{\max} \in \mathbb{N}^\star$; for $i \in [n]^\star$, $V_{0,i} \in \mathbb{R}^q$; $\widehat{S}_0 \in \mathbb{R}^q$; a positive sequence $\{\gamma_{k+1}, k \in [k_{\max} - 1]\}$; $\alpha > 0$

**Result:** The sequence: $\{\widehat{S}_k, k \in [k_{\max}]\}$

1 Set $V_0 = n^{-1} \sum_{i=1}^n V_{0,i}$ ;
2 **for** $k = 0, \ldots, k_{\max} - 1$ **do**
3    **for** $i = 1, \ldots, n$ **do**
4      *(worker #i)* ;
5      Sample $\mathsf{S}_{k+1,i}$, an approximation of $\bar{\mathsf{s}}_i \circ \mathsf{T}(\widehat{S}_k)$ ;
6      Set $\Delta_{k+1,i} = \mathsf{S}_{k+1,i} - V_{k,i} - \widehat{S}_k$ ;
7      Set $V_{k+1,i} = V_{k,i} + \alpha \operatorname{Quant}(\Delta_{k+1;i})$. Send $\operatorname{Quant}(\Delta_{k+1;i})$ to the central server ;
8    *(the central server)* ;
9    Compute $H_{k+1} = V_k + n^{-1} \sum_{i=1}^n \operatorname{Quant}(\Delta_{k+1;i})$ ;
10    Set $\widehat{S}_{k+1} = \widehat{S}_k + \gamma_{k+1} H_{k+1}$ ;
11    Set $V_{k+1} = V_k + \alpha n^{-1} \sum_{i=1}^n \operatorname{Quant}(\Delta_{k+1;i})$ ;
12    Send $\widehat{S}_{k+1}$ and $\mathsf{T}(\widehat{S}_{k+1})$ to the $n$ workers

---

## C.2 Notations and technical lemma

In this section, we start by introducing the appropriate filtrations employed later on to define conditional expectations. Then, we present a technical lemma used in the main proof of Theorem 1 (see Appendix C.4).

**Notations.** For any random variable $U$, we denote by $\sigma(U)$ the sigma-algebra generated by $U$. For $n$ sigma-algebras $\{\mathcal{F}_k, k \in [n]^\star\}$, we denote by $\bigvee_{k=1}^n \mathcal{F}_k$ the sigma-algebra generated by $\{\mathcal{F}_k, k \in [n]^\star\}$.

**Definition of filtrations.** Let us define the following filtrations. For any $i \in [n]^\star$, we set

$$\mathcal{F}_{0,i} = \mathcal{F}_{0,i}^+ := \sigma\left(\widehat{S}_0; V_{0,i}\right) \quad \text{and} \quad \mathcal{F}_0 := \bigvee_{i=1}^n \mathcal{F}_{0,i} \ .$$

Then, for all $k \geq 0$,

    (i) $\mathcal{F}_{k+1/2,i} := \mathcal{F}_{k,i}^+ \vee \sigma\left(\mathsf{S}_{k+1,i}\right)$,

    (ii) $\mathcal{F}_{k+1,i} := \mathcal{F}_{k+1/2,i} \vee \sigma\left(\operatorname{Quant}(\Delta_{k+1,i})\right)$,

    (iii) $\mathcal{F}_{k+1} := \bigvee_{i=1}^n \mathcal{F}_{k+1,i}$,

    (iv) $\mathcal{F}_{k+1,i}^+ := \mathcal{F}_{k+1,i} \vee \mathcal{F}_{k+1}$.

Note that, with these notations, for $k \geq 0$ and $i \in [n]^\star$, the random variables of the `FedEM` sequence defined in Algorithm 3 belong to the filtrations defined above as follows:

    (i) $\widehat{S}_k \in \mathcal{F}_{k,i}^+$, $\widehat{S}_k \in \mathcal{F}_k$,

    (ii) $\mathsf{S}_{k+1,i}, \Delta_{k+1,i} \in \mathcal{F}_{k+1/2,i}$,

    (iii) $V_{k+1,i} \in \mathcal{F}_{k+1,i}$,

    (iv) $\widehat{S}_{k+1}, H_{k+1}, V_{k+1} \in \mathcal{F}_{k+1}$.

Note also that we have the following inclusions for filtrations: $\mathcal{F}_k \subset \mathcal{F}_{k,i}^+ \subset \mathcal{F}_{k+1/2,i} \subset \mathcal{F}_{k+1,i} \subset \mathcal{F}_{k+1}$ for all $i \in [n]^\star$.

**Elementary lemma.** In the main proof of Theorem 1, we use the following elementary lemma.

**Lemma 6.** *For any $x, y \in \mathbb{R}^q$ and for any $\alpha \in \mathbb{R}$, one has:*

$$\|\alpha x + (1-\alpha)y\|^2 = \alpha\|x\|^2 + (1-\alpha)\|y\|^2 - \alpha(1-\alpha)\|x-y\|^2.$$

*Proof.* The LHS is equal to

$$\alpha^2\|x\|^2 + (1-\alpha)^2\|y\|^2 + 2\alpha(1-\alpha)\langle x, y\rangle \;.$$

The RHS is equal to

$$\alpha\|x\|^2 + (1-\alpha)\|y\|^2 - \alpha(1-\alpha)\left(\|x\|^2 + \|y\|^2 - 2\langle x, y\rangle\right) \;.$$

The proof is concluded upon noting that $\alpha - \alpha(1-\alpha) = \alpha^2$ and $(1-\alpha) - \alpha(1-\alpha) = (1-\alpha)^2$. $\quad\square$

## C.3 Preliminary results

In this section, we gather preliminary results on the control of the bias and variance of random variables of interest, which will be used in the main proof of Theorem 1. Namely, Proposition 8 controls the random field $H_{k+1}$, Proposition 10 controls the local increments $\Delta_{k+1,i}$ and Proposition 11 controls the memory term $V_{k,i}$.

### C.3.1 Results on the memory terms $V_k$.

Proposition 7 shows that, even if the central server only receives the variation $\alpha^{-1}(V_{k+1,i} - V_{k,i})$ from each local worker $\#i$, it is able to compute $n^{-1}\sum_{i=1}^{n} V_{k+1,i}$ as soon as the quantity $V_0$ is correctly initialized.

**Proposition 7.** *For any $k \in [k_{\max}]$, we have*

$$V_k = \frac{1}{n}\sum_{i=1}^{n} V_{k,i} \;.$$

*Proof.* The proof is by induction on $k$. When $k = 0$, the property holds true by Line 1 in algorithm 3. Assume that the property holds for $k \leq k_{\mathrm{in}} - 2$. Then by definition of $V_{k+1}$ and by the induction assumption:

$$V_{k+1} = V_k + \alpha\frac{1}{n}\sum_{i=1}^{n}\mathrm{Quant}(\Delta_{k+1,i}) = \frac{1}{n}\sum_{i=1}^{n}(V_{k,i} + \alpha\mathrm{Quant}(\Delta_{k+1,i}))$$

$$= \frac{1}{n}\sum_{i=1}^{n} V_{k+1,i} \;.$$

This concludes the induction. $\quad\square$

### C.3.2 Results on the random field $H_{k+1}$.

We compute in Proposition 8 the conditional expectation of $H_{k+1}$ with respect to the appropriate filtration $\mathcal{F}_k$ defined in Appendix C.2, as well as an upper bound on its variance. These results are combined in an upper bound on the conditional expectation of the square norm $\|H_{k+1}\|^2$ in Corollary 9.

Proposition 8 shows that the stochastic field $H_{k+1}$ is a (conditionally) unbiased estimator of $\mathsf{h}(\widehat{S}_k)$. In the case of no compression (i.e. $\omega = 0$), the conditional variance of $H_{k+1}$ is $\sigma^2/n$ where $\sigma^2$ is the mean variance of the approximations $\mathsf{S}_{k+1,i}$ over the $n$ workers (see A7); when $\sup_i \sigma_i^2 < \infty$, the variance is inversely proportional to the number of workers $n$.

**Proposition 8.** *Assume A6 and A7 and set $\sigma^2 := n^{-1}\sum_{i=1}^{n}\sigma_i^2$. For any $k \geq 0$,*

$$\mathbb{E}\left[H_{k+1}|\mathcal{F}_k\right] = \mathsf{h}(\widehat{S}_k) \;, \tag{15}$$

$$\mathbb{E}\left[\|H_{k+1} - \mathbb{E}\left[H_{k+1}|\mathcal{F}_k\right]\|^2|\mathcal{F}_k\right] \leq \frac{\omega}{n}\left(\frac{1}{n}\sum_{i=1}^{n}\mathbb{E}\left[\|\Delta_{k+1,i}\|^2|\mathcal{F}_k\right]\right) + \frac{\sigma^2}{n} \;. \tag{16}$$

*Proof.* Let $k \geq 0$. A6 guarantees

$$\mathbb{E}\left[\sum_{i=1}^n \text{Quant}(\Delta_{k+1,i}) \Big| \mathcal{F}_{k+1/2,i}\right] = \sum_{i=1}^n \mathbb{E}\left[\text{Quant}(\Delta_{k+1,i}) \big| \mathcal{F}_{k+1/2,i}\right]$$

$$= \sum_{i=1}^n \{\mathsf{S}_{k+1,i} - V_{k,i} - \widehat{S}_k\} . \tag{17}$$

Note also that, by A7, $\mathbb{E}\left[\mathsf{S}_{k+1,i} \big| \mathcal{F}_{k,i}^+\right] = \bar{\mathsf{s}}_i \circ \mathsf{T}(\widehat{S}_k)$, and that $V_k \in \mathcal{F}_k$ and $\mathcal{F}_k \subset \mathcal{F}_{k,i}^+ \subset \mathcal{F}_{k+1/2,i}$ (see Appendix C.2). Combined with (17) and using that $n^{-1}\sum_{i=1}^n V_{k,i} = V_k$ (see Proposition 7), this yields

$$\mathbb{E}\left[H_{k+1} | \mathcal{F}_k\right] = \mathbb{E}\left[n^{-1}\sum_{i=1}^n \text{Quant}(\Delta_{k+1,i}) \Big| \mathcal{F}_k\right] + V_k = \frac{1}{n}\sum_{i=1}^n \bar{\mathsf{s}}_i \circ \mathsf{T}(\widehat{S}_k) - \widehat{S}_k = \mathsf{h}(\widehat{S}_k) .$$

We now prove the second statement, and start by writing

$$H_{k+1} - \mathsf{h}(\widehat{S}_k) = \frac{1}{n}\sum_{i=1}^n \text{Quant}(\Delta_{k+1,i}) + V_k - \frac{1}{n}\sum_{i=1}^n \bar{\mathsf{s}}_i \circ \mathsf{T}(\widehat{S}_k) + \widehat{S}_k$$

$$= \frac{1}{n}\sum_{i=1}^n \left\{\text{Quant}(\Delta_{k+1,i}) - \mathbb{E}\left[\text{Quant}(\Delta_{k+1,i}) \big| \mathcal{F}_{k+1/2,i}\right]\right\}$$

$$+ \frac{1}{n}\sum_{i=1}^n \{\mathsf{S}_{k+1,i} - \bar{\mathsf{s}}_i \circ \mathsf{T}(\widehat{S}_k)\} ,$$

where we applied (17) to obtain the last equality. Using the fact that $\mathsf{S}_{k+1,i} - \bar{\mathsf{s}}_i \circ \mathsf{T}(\widehat{S}_k) \in \mathcal{F}_{k+1/2,i}$ and since, conditionally to $\mathcal{F}_k$, the workers are independent we have

$$\mathbb{E}\left[\|H_{k+1} - \mathsf{h}(\widehat{S}_k)\|^2 \Big| \mathcal{F}_k\right] = \frac{1}{n^2}\sum_{i=1}^n \mathbb{E}\left[\|\text{Quant}(\Delta_{k+1,i}) - \mathbb{E}\left[\text{Quant}(\Delta_{k+1,i}) \big| \mathcal{F}_{k+1/2,i}\right]\|^2 \big| \mathcal{F}_k\right]$$

$$+ \frac{1}{n^2}\sum_{i=1}^n \mathbb{E}\left[\|\mathsf{S}_{k+1,i} - \bar{\mathsf{s}}_i \circ \mathsf{T}(\widehat{S}_k)\|^2 \Big| \mathcal{F}_k\right] .$$

The second terme in the RHS is upped bounded by $n^{-1}\sigma^2$ (see A7). For the first term, using A6 and since $\Delta_{k+1,i} \in \mathcal{F}_{k+1/2,i}$, for any $i \in [n]^\star$ we have

$$\mathbb{E}\left[\|\text{Quant}(\Delta_{k+1,i}) - \mathbb{E}\left[\text{Quant}(\Delta_{k+1,i}) \big| \mathcal{F}_{k+1/2,i}\right]\|^2 \big| \mathcal{F}_{k+1/2,i}\right]$$
$$= \mathbb{E}\left[\|\text{Quant}(\Delta_{k+1,i})\|^2 \big| \mathcal{F}_{k+1/2,i}\right] - \|\Delta_{k+1,i}\|^2$$
$$\leq (1+\omega)\|\Delta_{k+1,i}\|^2 - \|\Delta_{k+1,i}\|^2 = \omega\|\Delta_{k+1,i}\|^2 ,$$

which concludes the proof upon conditioning with respect to $\mathcal{F}_k$. $\square$

**Corollary 9** (of Proposition 8)**.**

$$\mathbb{E}\left[\|H_{k+1}\|^2 \big| \mathcal{F}_k\right] \leq \|\mathsf{h}(\widehat{S}_k)\|^2 + \frac{\omega}{n}\left(\frac{1}{n}\sum_{i=1}^n \mathbb{E}\left[\|\Delta_{k+1,i}\|^2 \big| \mathcal{F}_k\right]\right) + \frac{\sigma^2}{n} .$$

### C.3.3   Results on the local increments $\Delta_{k+1,i}$.

We compute in Proposition 10 an upper bound on the second conditional moment of $\Delta_{k+1,i}$, with respect to the appropriate filtration $\mathcal{F}_k$ (see Appendix C.2).

**Proposition 10.** *Assume A7. For any $i \in [n]^\star$ and $k \in [k_{\max} - 1]$,*

$$\mathbb{E}\left[\|\Delta_{k+1,i}\|^2 \big| \mathcal{F}_k\right] \leq \|V_{k,i} - \mathsf{h}_i(\widehat{S}_k)\|^2 + \sigma_i^2 .$$

*Proof.* Let $i \in [n]^\star$ and $k \in [k_{\max} - 1]$. By A7, $\mathbb{E}\left[\mathsf{S}_{k+1,i} - \widehat{S}_k \middle| \mathcal{F}_{k,i}^+\right] = \mathsf{h}_i(\widehat{S}_k)$; in addition, $\widehat{S}_k \in \mathcal{F}_k$, $V_{k,i} \in \mathcal{F}_{k,i}^+$ and $\mathcal{F}_k \subset \mathcal{F}_{k,i}^+$. Hence, we get

$$
\begin{aligned}
\mathbb{E}\left[\|\Delta_{k+1,i}\|^2 \middle| \mathcal{F}_{k,i}^+\right] &= \mathbb{E}\left[\|\mathsf{S}_{k+1,i} - V_{k,i} - \widehat{S}_k\|^2 \middle| \mathcal{F}_{k,i}^+\right] \\
&= \|\mathsf{h}_i(\widehat{S}_k) - V_{k,i}\|^2 + \mathbb{E}\left[\|\mathsf{S}_{k+1,i} - \widehat{S}_k - \mathsf{h}_i(\widehat{S}_k)\|^2 \middle| \mathcal{F}_{k,i}^+\right] \\
&= \|\mathsf{h}_i(\widehat{S}_k) - V_{k,i}\|^2 + \mathbb{E}\left[\|\mathsf{S}_{k+1,i} - \bar{\mathsf{s}}_i \circ \mathsf{T}(\widehat{S}_k)\|^2 \middle| \mathcal{F}_{k,i}^+\right] \\
&\overset{A7}{\leq} \|\mathsf{h}_i(\widehat{S}_k) - V_{k,i}\|^2 + \sigma_i^2 \ .
\end{aligned}
\tag{18}
$$

The proof is concluded upon noting that $\mathcal{F}_k \subset \mathcal{F}_{k,i}^+$, $\widehat{S}_k \in \mathcal{F}_k$ and $V_{k,i} \in \mathcal{F}_k$. $\qquad\square$

### C.3.4  Results on the memory terms $V_{k,i}$.

Our final preliminary result is to compute in Proposition 11 an upper bound to control the conditional variance of the local memory terms $V_{k,i}$ with respect to the appropriate filtration $\mathcal{F}_k$ (see Appendix C.2).

**Proposition 11.** *Assume A5, A6 and A7; set $L^2 := n^{-1} \sum_{i=1}^n L_i^2$ and $\sigma^2 := n^{-1} \sum_{i=1}^n \sigma_i^2$. For any $k \geq 0$, set*

$$
G_k := \frac{1}{n} \sum_{i=1}^n \|V_{k,i} - \mathsf{h}_i(\widehat{S}_k)\|^2 \ .
$$

*For any $k \in [k_{\max} - 1]$ and $\alpha \in (0, (1/(1+\omega))]$, it holds that*

$$
\mathbb{E}[G_{k+1}|\mathcal{F}_k] \leq \left(1 - \frac{\alpha}{2} + 2\gamma_{k+1}^2 \frac{L^2}{\alpha} \frac{\omega}{n}\right) G_k + 2\gamma_{k+1}^2 \frac{L^2}{\alpha} \|\mathsf{h}(\widehat{S}_k)\|^2
$$
$$
+ 2\left(\alpha + \gamma_{k+1}^2 \frac{L^2}{\alpha} \frac{1+\omega}{n}\right) \sigma^2 \ .
$$

*Proof.* We start by computing an upper bound for the local conditional expectations $\mathbb{E}\left[\|V_{k+1,i} - \mathsf{h}_i(\widehat{S}_{k+1})\|^2 \middle| \mathcal{F}_k\right]$, $i \in [n]^\star$ and then derive the result of Proposition 11 by averaging over the $n$ local workers.

Let $i \in [n]^\star$; from Lemma 6, we have for any $s \in \mathbb{R}^q$

$$
\begin{aligned}
\left\|\mathbb{E}\left[V_{k+1,i} - s \middle| \mathcal{F}_{k+1/2,i}\right]\right\|^2 &= \|(1-\alpha)(V_{k,i} - s) + \alpha(\mathsf{S}_{k+1,i} - \widehat{S}_k - s)\|^2 \\
&= (1-\alpha)\|V_{k,i} - s\|^2 + \alpha\|\mathsf{S}_{k+1,i} - \widehat{S}_k - s\|^2 - \alpha(1-\alpha)\|\Delta_{k+1,i}\|^2 \ .
\end{aligned}
$$

On the other hand,

$$
\left\|V_{k+1,i} - \mathbb{E}\left[V_{k+1,i}|\mathcal{F}_{k+1/2,i}\right]\right\|^2 = \alpha^2 \left\|\mathrm{Quant}(\Delta_{k+1,i}) - \mathbb{E}\left[\mathrm{Quant}(\Delta_{k+1,i})\middle|\mathcal{F}_{k+1/2,i}\right]\right\|^2
$$

and by A6 (see the proof of Proposition 8 for the same computation)

$$
\mathbb{E}\left[\left\|V_{k+1,i} - \mathbb{E}\left[V_{k+1,i}|\mathcal{F}_{k+1/2,i}\right]\right\|^2 \middle| \mathcal{F}_{k+1/2,i}\right] \leq \alpha^2 \omega \|\Delta_{k+1,i}\|^2 \ .
$$

Hence

$$
\begin{aligned}
\mathbb{E}\left[\|V_{k+1,i} - s\|^2 \middle| \mathcal{F}_{k+1/2,i}\right] &\leq \mathbb{E}\left[\left\|V_{k+1,i} - s - \mathbb{E}\left[V_{k+1,i} - s \middle| \mathcal{F}_{k+1/2,i}\right]\right\|^2 \middle| \mathcal{F}_{k+1/2,i}\right] \\
&\quad + \mathbb{E}\left[\left\|\mathbb{E}\left[V_{k+1,i} - s\middle|\mathcal{F}_{k+1/2,i}\right]\right\|^2 \middle| \mathcal{F}_{k+1/2,i}\right] \\
&\leq (1-\alpha)\|V_{k,i} - s\|^2 + \alpha\|\mathsf{S}_{k+1,i} - \widehat{S}_k - s\|^2 + \alpha(\alpha(1+\omega) - 1)\|\Delta_{k+1,i}\|^2 \ . \quad (19)
\end{aligned}
$$

For any $\beta > 0$, using that $\|a + b\|^2 \leq (1 + \beta^2)\|a\|^2 + (1 + \beta^{-2})\|b\|^2$, we have

$$
\mathbb{E}\left[\|V_{k+1,i} - \mathsf{h}_i(\widehat{S}_{k+1})\|^2 \Big| \mathcal{F}_k\right]
$$

$$
\leq (1 + \beta^{-2})\mathbb{E}\left[\|V_{k+1,i} - \mathsf{h}_i(\widehat{S}_k)\|^2 \Big| \mathcal{F}_k\right] + (1 + \beta^2)\mathbb{E}\left[\|\mathsf{h}_i(\widehat{S}_k) - \mathsf{h}_i(\widehat{S}_{k+1})\|^2 \Big| \mathcal{F}_k\right]
$$

$$
\overset{A5}{\leq} (1 + \beta^{-2})\mathbb{E}\left[\mathbb{E}\left[\|V_{k+1,i} - \mathsf{h}_i(\widehat{S}_k)\|^2 \Big| \mathcal{F}_{k+1/2,i}\right] \Big| \mathcal{F}_k\right] + (1 + \beta^2)L_i^2 \gamma_{k+1}^2 \mathbb{E}[\|H_{k+1}\|^2 | \mathcal{F}_k]
$$

$$
\overset{(19)}{\leq} (1 + \beta^{-2})\bigg((1 - \alpha)\|V_{k,i} - \mathsf{h}_i(\widehat{S}_k)\|^2
$$

$$
+ \alpha\mathbb{E}[\|\mathsf{S}_{k+1,i} - \widehat{S}_k - \mathsf{h}_i(\widehat{S}_k)\|^2 | \mathcal{F}_k] + \alpha\,(\alpha(1 + \omega) - 1)\,\mathbb{E}\left[\|\Delta_{k+1,i}\|^2 \Big| \mathcal{F}_k\right]\bigg)
$$

$$
+ (1 + \beta^2)L_i^2 \gamma_{k+1}^2 \mathbb{E}\left[\|H_{k+1}\|^2 \Big| \mathcal{F}_k\right]\;,
$$

where we have used (19) with $s = \mathsf{h}_i(\widehat{S}_k) \in \mathcal{F}_k \subset \mathcal{F}_{k+1/2,i}$. Choose $\beta > 0$ such that

$$
\beta^{-2} := \begin{cases} \frac{\alpha}{2(1-\alpha)} & \text{if } \alpha \leq 2/3 \\ 1 & \text{if } \alpha \geq 2/3 \end{cases}
$$

which implies that $(1 + \beta^{-2})(1 - \alpha) \leq 1 - \alpha/2$; note also that $1 \leq 1 + \beta^{-2} \leq 2$. By Corollary 9, we have (remember that $\alpha(1 + \omega) - 1 \leq 0$)

$$
\mathbb{E}\left[\|V_{k+1,i} - \mathsf{h}_i(\widehat{S}_{k+1})\|^2 \Big| \mathcal{F}_k\right] \leq \left(1 - \frac{\alpha}{2}\right)\|V_{k,i} - \mathsf{h}_i(\widehat{S}_k)\|^2
$$

$$
+ 2\alpha\mathbb{E}\left[\|\mathsf{S}_{k+1,i} - \bar{\mathsf{s}}_i \circ \mathsf{T}(\widehat{S}_k)\|^2 \Big| \mathcal{F}_k\right] + \alpha\,(\alpha(1 + \omega) - 1)\,\mathbb{E}\left[\|\Delta_{k+1,i}\|^2 \Big| \mathcal{F}_k\right]
$$

$$
+ \frac{2}{\alpha}L_i^2 \gamma_{k+1}^2 \left(\frac{\omega}{n^2}\sum_{i=1}^{n}\mathbb{E}\left[\|\Delta_{k+1,i}\|^2 \Big| \mathcal{F}_k\right] + \|\mathsf{h}(\widehat{S}_k)\|^2 + \frac{\sigma^2}{n}\right)\;.
$$

Since $\alpha(1 + \omega) - 1 \leq 0$, using A7 and finally Proposition 10, we get:

$$
\mathbb{E}\left[\|V_{k+1,i} - \mathsf{h}_i(\widehat{S}_{k+1})\|^2 | \mathcal{F}_k\right] \leq \left(1 - \frac{\alpha}{2}\right)\|V_{k,i} - \mathsf{h}_i(\widehat{S}_k)\|^2 + 2\alpha\sigma_i^2
$$

$$
+ 2\gamma_{k+1}^2 \frac{L_i^2}{\alpha}\frac{\omega}{n^2}\sum_{i=1}^{n}\|h_i(\widehat{S}_k) - V_{k,i}\|^2 + 2\gamma_{k+1}^2 \frac{L_i^2}{\alpha}\|\mathsf{h}(\widehat{S}_k)\|^2
$$

$$
+ 2\gamma_{k+1}^2 \frac{L_i^2}{\alpha}\frac{1 + \omega}{n}\sigma^2\;.
$$

Overall, by averaging the previous inequality over all workers, we get:

$$
\mathbb{E}[G_{k+1} | \mathcal{F}_k] \leq \left(1 - \frac{\alpha}{2} + 2\gamma_{k+1}^2 \frac{L^2}{\alpha}\frac{\omega}{n}\right)G_k + 2\gamma_{k+1}^2 \frac{L^2}{\alpha}\|\mathsf{h}(\widehat{S}_k)\|^2
$$

$$
+ 2\left(\alpha + \gamma_{k+1}^2 \frac{L^2}{\alpha}\frac{1 + \omega}{n}\right)\sigma^2\;.
$$

$\square$

## C.4  Proof of Theorem 1

Equipped with the necessary results, we now provide the main proof of Theorem 1. We proceed in three steps, as follows. First, for $k \geq 1$, we compute an upper bound on the average decrement $\mathbb{E}\left[\mathsf{W}(\widehat{S}_{k+1}) \Big| \mathcal{F}_k\right] - \mathsf{W}(\widehat{S}_k)$ of the Lyapunov function W (defined in A4). Second, we introduce the maximal value of the learning rate. Third and finally, we deduce the result of Theorem 1 by computing the expectation w.r.t. a randomly chosen termination time $K$ in $[k_{\max} - 1]$; in this step, we restrict the computations to the case the step sizes are constant ($\gamma_{k+1} = \gamma$ for any $k \geq 0$).

**Step 1: Upper bound on the decrement.** Let $k \geq 0$; from A4, we have

$$
\begin{aligned}
\mathrm{W}(\widehat{S}_{k+1}) &\leq \mathrm{W}(\widehat{S}_k) + \left\langle \nabla \mathrm{W}(\widehat{S}_k), \widehat{S}_{k+1} - \widehat{S}_k \right\rangle + \frac{L_{\dot{\mathrm{W}}}}{2}\|\widehat{S}_{k+1} - \widehat{S}_k\|^2 \\
&\leq \mathrm{W}(\widehat{S}_k) - \gamma_{k+1}\left\langle B(\widehat{S}_k)\,\mathsf{h}(\widehat{S}_k), H_{k+1} \right\rangle + \frac{L_{\dot{\mathrm{W}}}}{2}\gamma_{k+1}^2\|H_{k+1}\|^2 \ .
\end{aligned} \tag{20}
$$

Since $\widehat{S}_k \in \mathcal{F}_k$, by Proposition 8 and A4 we have

$$
\mathbb{E}\left[\left\langle B(\widehat{S}_k)\,\mathsf{h}(\widehat{S}_k), H_{k+1}\right\rangle \middle| \mathcal{F}_k\right] = \left\langle B(\widehat{S}_k)\,\mathsf{h}(\widehat{S}_k), \mathsf{h}(\widehat{S}_k)\right\rangle \geq v_{\min}\|\mathsf{h}(\widehat{S}_k)\|^2. \tag{21}
$$

Hence, combining (20) and (21), we have

$$
\mathbb{E}\left[\mathrm{W}(\widehat{S}_{k+1})\middle|\mathcal{F}_k\right] \leq \mathrm{W}(\widehat{S}_k) - \gamma_{k+1}v_{\min}\|\mathsf{h}(\widehat{S}_k)\|^2 + \gamma_{k+1}^2\frac{L_{\dot{\mathrm{W}}}}{2}\mathbb{E}\left[\|H_{k+1}\|^2\middle|\mathcal{F}_k\right]
$$

$$
\leq \mathrm{W}(\widehat{S}_k) - \gamma_{k+1}v_{\min}\|\mathsf{h}(\widehat{S}_k)\|^2 + \gamma_{k+1}^2\frac{L_{\dot{\mathrm{W}}}}{2}\mathbb{E}\left[\|H_{k+1} - \mathbb{E}\left[H_{k+1}|\mathcal{F}_k\right]\|^2\middle|\mathcal{F}_k\right] + \gamma_{k+1}^2\frac{L_{\dot{\mathrm{W}}}}{2}\|\mathsf{h}(\widehat{S}_k)\|^2
$$

$$
\leq \mathrm{W}(\widehat{S}_k) - \gamma_{k+1}v_{\min}\left(1 - \gamma_{k+1}\frac{L_{\dot{\mathrm{W}}}}{2v_{\min}}\right)\|\mathsf{h}(\widehat{S}_k)\|^2 + \gamma_{k+1}^2\frac{L_{\dot{\mathrm{W}}}}{2}\mathbb{E}\left[\|H_{k+1} - \mathbb{E}\left[H_{k+1}|\mathcal{F}_k\right]\|^2\middle|\mathcal{F}_k\right] \ .
$$

Applying Proposition 8, we obtain that

$$
\begin{aligned}
\mathbb{E}\left[\mathrm{W}(\widehat{S}_{k+1})\middle|\mathcal{F}_k\right] &\leq \mathrm{W}(\widehat{S}_k) - \gamma_{k+1}v_{\min}\left(1 - \gamma_{k+1}\frac{L_{\dot{\mathrm{W}}}}{2v_{\min}}\right)\|\mathsf{h}(\widehat{S}_k)\|^2 \\
&\quad + \gamma_{k+1}^2\frac{L_{\dot{\mathrm{W}}}}{2}\frac{\omega}{n}\left(\frac{1}{n}\sum_{i=1}^n\mathbb{E}\left[\|\Delta_{k+1,i}\|^2\middle|\mathcal{F}_k\right]\right) + \gamma_{k+1}^2\frac{L_{\dot{\mathrm{W}}}}{2n}\sigma^2 \ .
\end{aligned} \tag{22}
$$

Finally, using Proposition 10 and (22), we get:

$$
\begin{aligned}
\mathbb{E}[\mathrm{W}(\widehat{S}_{k+1})|\mathcal{F}_k] &\leq \mathrm{W}(\widehat{S}_k) - \gamma_{k+1}v_{\min}\left(1 - \gamma_{k+1}\frac{L_{\dot{\mathrm{W}}}}{2v_{\min}}\right)\|\mathsf{h}(\widehat{S}_k)\|^2 \\
&\quad + \gamma_{k+1}^2\frac{L_{\dot{\mathrm{W}}}}{2}\frac{\omega}{n}G_k + \gamma_{k+1}^2\frac{L_{\dot{\mathrm{W}}}}{2n}(1+\omega)\sigma^2 \ ,
\end{aligned} \tag{23}
$$

where

$$
G_k := \frac{1}{n}\sum_{i=1}^n\|V_{k,i} - \mathsf{h}_i(\widehat{S}_k)\|^2 \ .
$$

**Step 2: Maximal learning rate $\gamma_{k+1}$ when $\omega \neq 0$.** From Proposition 11, for any non-increasing positive sequence $\{\gamma_k, k \in [k_{\max} - 1]\}$ such that

$$
\gamma_{k+1}^2 \leq \frac{\alpha^2}{8L^2}\frac{n}{\omega},
$$

and for any positive sequence $\{C_k, k \in [k_{\max} - 1]\}$, it holds

$$
\begin{aligned}
C_{k+1}\mathbb{E}\left[G_{k+1}|\mathcal{F}_k\right] &\leq C_{k+1}\left(1 - \frac{\alpha}{4}\right)G_k \\
&\quad + C_{k+1}\gamma_{k+1}^2\frac{2}{\alpha}L^2\|\mathsf{h}(\widehat{S}_k)\|^2 + 2C_{k+1}\left(\alpha + \gamma_{k+1}^2\frac{L^2}{\alpha}\frac{1+\omega}{n}\right)\sigma^2 \ .
\end{aligned} \tag{24}
$$

Combining equations (23) and (24), we thus have

$$
\begin{aligned}
\mathbb{E}[\mathrm{W}(\widehat{S}_{k+1})|\mathcal{F}_k] + C_{k+1}\mathbb{E}\left[G_{k+1}|\mathcal{F}_k\right] &\leq \mathrm{W}(\widehat{S}_k) + C_k G_k \\
&\quad - \gamma_{k+1}v_{\min}\left(1 - \gamma_{k+1}\frac{L_{\dot{\mathrm{W}}}}{2v_{\min}} - \frac{C_{k+1}}{v_{\min}}\gamma_{k+1}\frac{2}{\alpha}L^2\right)\|\mathsf{h}(\widehat{S}_k)\|^2 \\
&\quad + \left(\gamma_{k+1}^2\frac{L_{\dot{\mathrm{W}}}}{2}\frac{\omega}{n} - C_k + C_{k+1} - C_{k+1}\frac{\alpha}{4}\right)G_k \\
&\quad + \left\{2\alpha C_{k+1} + \gamma_{k+1}^2\frac{(1+\omega)}{n}\left(\frac{L_{\dot{\mathrm{W}}}}{2} + 2C_{k+1}\frac{L^2}{\alpha}\right)\right\}\sigma^2 \ .
\end{aligned}
$$

We choose the sequence $\{C_k\}$ as follows:

$$C_k := \gamma_k^2 \frac{2L_{\dot{W}}}{\alpha} \frac{\omega}{n} \; ;$$

the sequence satisfies $C_{k+1} \leq C_k$ (since $\gamma_{k+1} \leq \gamma_k$) and $\gamma_{k+1}^2 L_{\dot{W}} \omega/(2n) \leq C_{k+1}\alpha/4$. By convention, $\gamma_0 \in [\gamma_1, +\infty)$. Therefore

$$\mathbb{E}[W(\widehat{S}_{k+1})|\mathcal{F}_k] + \gamma_{k+1}^2 \frac{2L_{\dot{W}}}{\alpha}\frac{\omega}{n}\mathbb{E}[G_{k+1}|\mathcal{F}_k] \leq W(\widehat{S}_k) + \gamma_k^2 \frac{2L_{\dot{W}}}{\alpha}\frac{\omega}{n}G_k \tag{25}$$

$$- \gamma_{k+1}v_{\min}\left(1 - \gamma_{k+1}\frac{L_{\dot{W}}}{2v_{\min}}\left\{1 + 8\gamma_{k+1}^2\frac{\omega}{\alpha^2 n}L^2\right\}\right)\|h(\widehat{S}_k)\|^2 \tag{26}$$

$$+ 4\gamma_{k+1}^2 L_{\dot{W}}\frac{\omega}{n}\left\{1 + \frac{(1+\omega)}{8\omega}\left(1 + \gamma_{k+1}^2 8\frac{L^2}{\alpha^2}\frac{\omega}{n}\right)\right\}\sigma^2 \; . \tag{27}$$

**Step 3: Computing the expectation.** Let us apply the expectations, sum from $k = 0$ to $k = k_{\max} - 1$, and divide by $k_{\max}$. This yields

$$\frac{v_{\min}}{k_{\max}}\sum_{k=0}^{k_{\max}-1}\gamma_{k+1}\left(1 - \gamma_{k+1}\frac{L_{\dot{W}}}{2v_{\min}}\left\{1 + 8\gamma_{k+1}^2\frac{\omega}{\alpha^2 n}L^2\right\}\right)\|h(\widehat{S}_k)\|^2$$

$$\leq k_{\max}^{-1}\left\{W(\widehat{S}_0) + \gamma_0^2\frac{2L_{\dot{W}}}{\alpha}\frac{\omega}{n}G_0 - \mathbb{E}\left[W(\widehat{S}_{k_{\max}})\right] - \gamma_{k_{\max}}^2\frac{2L_{\dot{W}}}{\alpha}\frac{\omega}{n}\mathbb{E}[G_{k_{\max}}]\right\}$$

$$+ 4L_{\dot{W}}\frac{\omega}{n}\frac{1}{k_{\max}}\sum_{k=0}^{k_{\max}-1}\gamma_{k+1}^2\left\{1 + \frac{(1+\omega)}{8\omega}\left(1 + \gamma_{k+1}^2 8\frac{L^2}{\alpha^2}\frac{\omega}{n}\right)\right\}\sigma^2 \; .$$

We now focus on the case when $\gamma_{k+1} = \gamma$ for any $k \geq 0$. Denote by $K$ a uniform random variable on $[k_{\max} - 1]$, independent of the path $\{\widehat{S}_k, k \in [k_{\max}]\}$. Since $\gamma^2 \leq \alpha^2 n/(8L^2\omega)$, we have

$$1 + 8\gamma^2\frac{\omega}{\alpha^2 n}L^2 \leq 2 \; .$$

This yields

$$v_{\min}\gamma\left(1 - \gamma\frac{L_{\dot{W}}}{v_{\min}}\right)\mathbb{E}\left[\|h(\widehat{S}_K)\|^2\right]$$

$$\leq k_{\max}^{-1}\left\{W(\widehat{S}_0) + \gamma^2\frac{2L_{\dot{W}}}{\alpha}\frac{\omega}{n}G_0 - \mathbb{E}\left[W(\widehat{S}_{k_{\max}})\right] - \gamma^2\frac{2L_{\dot{W}}}{\alpha}\frac{\omega}{n}\mathbb{E}[G_{k_{\max}}]\right\}$$

$$+ 4L_{\dot{W}}\frac{\omega}{n}\gamma^2\left\{1 + \frac{(1+\omega)}{4\omega}\right\}\sigma^2 \; . \tag{28}$$

Note that $4(1 + (1+\omega)/(4\omega)) = (5\omega + 1)/\omega$.

**Step 4. Conclusion (when $\omega \neq 0$).** By choosing $V_{0,i} = h_i$ for any $i \in [n]^\star$, we have $G_0 = 0$. The roots of $\gamma \mapsto \gamma(1 - \gamma L_{\dot{W}}/v_{\min})$ are 0 and $v_{\min}/L_{\dot{W}}$ and its maximum is reached at $v_{\min}/(2L_{\dot{W}})$: this function is increasing on $(0, v_{\min}/(2L_{\dot{W}})]$. We therefore choose $\gamma \in (0, \gamma_{\max}(\alpha)]$ where

$$\gamma_{\max}(\alpha) := \min\left(\frac{v_{\min}}{2L_{\dot{W}}}; \frac{\alpha}{2\sqrt{2}L}\frac{\sqrt{n}}{\sqrt{\omega}}\right)$$

Finally, since $\alpha \in (0, 1/(1+\omega)]$, we choose $\alpha = 1/(1+\omega)$. This yields

$$\gamma_{\max} := \min\left(\frac{v_{\min}}{2L_{\dot{W}}}; \frac{1}{2\sqrt{2}L}\frac{\sqrt{n}}{\sqrt{\omega}(1+\omega)}\right) \; .$$

**Case $\omega = 0$.** From (23), applying the expectation we have

$$\gamma_{k+1}v_{\min}\left(1 - \gamma_{k+1}\frac{L_{\dot{W}}}{2v_{\min}}\right)\mathbb{E}\left[\|h(\widehat{S}_k)\|^2\right] \leq \mathbb{E}\left[W(\widehat{S}_k)\right] - \mathbb{E}\left[W(\widehat{S}_{k+1})\right] + \gamma_{k+1}^2\frac{L_{\dot{W}}\sigma^2}{2n} \; .$$

We now sum from $k = 0$ to $k = k_{\max} - 1$ and then divide by $k_{\max}$. In the case $\gamma_{k+1} = \gamma$, we have

$$\gamma v_{\min}\left(1 - \gamma\frac{L_{\dot{W}}}{2v_{\min}}\right)\mathbb{E}\left[\|h(\widehat{S}_K)\|^2\right] \leq k_{\max}^{-1}\left(\mathbb{E}\left[W(\widehat{S}_0)\right] - \min W\right) + \gamma^2\frac{L_{\dot{W}}\sigma^2}{2n} \; . \tag{29}$$

**Remark on the maximal learning rate.** The condition $\gamma_{k+1} \leq \frac{\alpha}{2\sqrt{2}L}\frac{\sqrt{n}}{\sqrt{\omega}}$ is used twice in the proof:

1. To ensure that $\left(1 - \gamma_{k+1}\frac{L_{\dot{W}}}{2v_{\min}}\left\{1 + 8\gamma_{k+1}^2\frac{\omega}{\alpha^2 n}L^2\right\}\right) \geq \left(1 - \gamma_{k+1}\frac{L_{\dot{W}}}{v_{\min}}\right)$ in order to obtain Equation (28).

2. To ensure that the process $(G_k)_{k\geq 0}$ is "pseudo-contractive" (i.e., satisfies a recursion of the form $u_{k+1} \leq \rho u_k + v_k$, with $\rho < 1$) in Proposition 11.

A more detailed analysis can get rid of this condition (and thus the dependency $\gamma = O_{\omega\to\infty}(\omega^{-3/2})$, as we recall that $\alpha^1 \propto_{\omega\to\infty} \omega$) for the *first point*. Indeed, we ultimately only require

$$\left(1 - \gamma_{k+1}\frac{L_{\dot{W}}}{2v_{\min}}\left\{1 + 8\gamma_{k+1}^2\frac{\omega}{\alpha^2 n}L^2\right\}\right) \geq \frac{1}{2} \tag{30}$$

to conclude the proof. This is for example satisfied if $\gamma_{k+1}\frac{L_{\dot{W}}}{2v_{\min}} \leq \frac{1}{4}$ and $8\gamma_{k+1}^3\frac{L_{\dot{W}}}{2v_{\min}}\frac{\omega}{\alpha^2 n}L^2 \leq \frac{1}{4}$. This approach results in a better asymptotic dependency of the maximal learning rate w.r.t. $\omega$ to obtain Equation (30): $\gamma = O_{\omega\to\infty}(\omega^{-1})$. *However*, the condition $\gamma_{k+1} \leq \frac{\alpha}{2\sqrt{2}L}\frac{\sqrt{n}}{\sqrt{\omega}}$ seems to be *necessary* to obtain the *second point* and Proposition 11. The possibility of providing a similar result to Proposition 11 without the $\omega^{-3/2}$ dependency, is an interesting open problem.

### C.5 Proof of Corollary 2

In (8), the RHS is of the form $A/\gamma + \gamma B$ for some positive constants $A, B$: we have $A/\gamma + \gamma B \geq 2\sqrt{AB}$ with equality reached with $\gamma_\star := \sqrt{A/B}$. Hence, we set

$$\gamma_\star := \frac{1}{\sigma}\left(\frac{n\left(\mathrm{W}(\widehat{S}_0) - \min \mathrm{W}\right)}{L_{\dot{W}}(1 + 5\omega)}\right)^{1/2}\frac{1}{\sqrt{k_{\max}}} .$$

If $\gamma_\star \leq \gamma_{\max}$, then let us apply (8) with $\gamma = \gamma_\star$ which yields a RHS given by $2\sqrt{A/B}$ i.e.

$$2\sigma\left(\left(\mathrm{W}(\widehat{S}_0) - \min \mathrm{W}\right)L_{\dot{W}}\frac{(1 + 5\omega)}{n}\right)^{1/2}\frac{1}{\sqrt{k_{\max}}} .$$

If $\gamma_\star \geq \gamma_{\max}$, we write

$$\frac{A}{\gamma_{\max}} + B\gamma_{\max} \leq \frac{A}{\gamma_{\max}} + \frac{A}{\gamma_{\max}}\frac{\gamma_{\max}^2 B}{A} = \frac{A}{\gamma_{\max}} + \frac{A}{\gamma_{\max}}\frac{\gamma_{\max}^2}{\gamma_\star^2} \leq 2\frac{A}{\gamma_{\max}} .$$

and the RHS is upper bounded by

$$2\frac{\mathrm{W}(\widehat{S}_0) - \min \mathrm{W}}{\gamma_{\max}k_{\max}} .$$

Finally, in the LHS of (8), we have

$$1 - \gamma\frac{L_{\dot{W}}}{v_{\min}} \geq 1 - \gamma_{\max}\frac{L_{\dot{W}}}{v_{\min}} \geq 1 - \frac{v_{\min}}{2L_{\dot{W}}}\frac{L_{\dot{W}}}{v_{\min}} = \frac{1}{2} .$$

This concludes the proof.

## D  Partial Participation case

In this section, we generalize the result of Theorem 1 to the *partial participation case*. This extra scheme could be incorporated into the main proof, but we choose to present it separately to improve the readability of the main proof in Appendix C. We first provide an equivalent description of algorithm 1 in Appendix D.1; algorithm 4 will be used throughout this section. Then, we introduce a new family of filtrations. In Appendix D.3, we first establish preliminary results and then give the proof of Theorem 4 in Appendix D.4.

The assumptions A1 to A3 hold throughout this section.

## D.1 An equivalent algorithm

In this Section, we describe an equivalent algorithm, that outputs the same result as Algorithm 1, and for which the analysis is conducted.

---

**Algorithm 4:** FedEM with partial participation

**Data:** $k_{\max} \in \mathbb{N}^\star$; for $i \in [n]^\star$, $V_{0,i} \in \mathbb{R}^q$; $\widehat{S}_0 \in \mathbb{R}^q$; a positive sequence $\{\gamma_{k+1}, k \in [k_{\max} - 1]\}$; $\alpha > 0$; $p \in (0,1)$.

**Result:** The FedEM-PP sequence: $\{\widehat{S}_k, k \in [k_{\max}]\}$

1  Set $V_0 = n^{-1} \sum_{i=1}^{n} V_{0,i}$ ;
2  **for** $k = 0, \ldots, k_{\max} - 1$ **do**
3       **for** $i = 1, \ldots, n$ **do**
4           *(worker #i)*;
5           Sample $\mathsf{S}_{k+1,i}$, an approximation of $\bar{\mathsf{s}}_i \circ \mathsf{T}(\widehat{S}_k)$ ;
6           Set $\Delta_{k+1,i} = \mathsf{S}_{k+1,i} - V_{k,i} - \widehat{S}_k$ ;
7           Sample a Bernoulli r.v. $B_{k+1,i}$ with success probability $p$ ;
8           Set $V_{k+1,i} = V_{k,i} + \alpha\, B_{k+1,i} \mathrm{Quant}(\Delta_{k+1,i})$. ;
9           Send $B_{k+1,i} \mathrm{Quant}(\Delta_{k+1,i})$ to the central server ;
10      *(the central server)* ;
11      Set $H_{k+1} = V_k + (np)^{-1} \sum_{i=1}^{n} B_{k+1,i} \mathrm{Quant}(\Delta_{k+1,i})$ ;
12      Set $\widehat{S}_{k+1} = \widehat{S}_k + \gamma_{k+1} H_{k+1}$ ;
13      Set $V_{k+1} = V_k + \alpha n^{-1} \sum_{i=1}^{n} B_{k+1,i} \mathrm{Quant}(\Delta_{k+1,i})$ ;
14      Send $\widehat{S}_{k+1}$ and $\mathsf{T}(\widehat{S}_{k+1})$ to the $n$ workers

---

## D.2 Notations

Let us introduce a new sequence of filtrations. For any $i \in [n]^\star$, we set

$$\mathcal{F}_{0,i} = \mathcal{F}_{0,i}^+ := \sigma\left(\widehat{S}_0; V_{0,i}\right) \text{ and } \qquad \mathcal{F}_0 := \bigvee_{i=1}^{n} \mathcal{F}_{0,i} \ .$$

Then, for all $k \geq 0$,

   (i) $\mathcal{F}_{k+1/3,i} := \mathcal{F}_{k,i}^+ \vee \sigma\left(\mathsf{S}_{k+1,i}\right)$,

   (ii) $\mathcal{F}_{k+2/3,i} := \mathcal{F}_{k+1/3,i} \vee \sigma\left(\mathrm{Quant}(\Delta_{k+1,i})\right)$,

   (iii) $\mathcal{F}_{k+1,i} := \mathcal{F}_{k+2/3,i} \vee \sigma\left(B_{k+1,i}\right)$,

   (iv) $\mathcal{F}_{k+1} := \bigvee_{i=1}^{n} \mathcal{F}_{k+1,i}$,

   (v) $\mathcal{F}_{k+1,i}^+ := \mathcal{F}_{k+1,i} \vee \mathcal{F}_{k+1}$.

Note that, with these notations, for $k \geq 0$ and $i \in [n]^\star$, the random variables of the FedEM sequence defined in algorithm 4 belong to the filtrations defined above as follows:

   (i) $\widehat{S}_k \in \mathcal{F}_{k,i}^+$, $\widehat{S}_k \in \mathcal{F}_k$,

   (ii) $\mathsf{S}_{k+1,i}, \Delta_{k+1,i} \in \mathcal{F}_{k+1/3,i}$,

   (iii) $V_{k+1,i} \in \mathcal{F}_{k+1,i}$,

   (iv) $\widehat{S}_{k+1}, H_{k+1}, V_{k+1} \in \mathcal{F}_{k+1}$.

Note also that we have the following inclusions for filtrations: $\mathcal{F}_k \subset \mathcal{F}_{k,i}^+ \subset \mathcal{F}_{k+1/3,i} \subset \mathcal{F}_{k+2/3,i} \subset \mathcal{F}_{k+1,i} \subset \mathcal{F}_{k+1}$ for all $i \in [n]^\star$.

## D.3 Preliminary results

In this section, we extend Proposition 7, Proposition 8 (that controls the random field $H_{k+1}$) and Proposition 11 (that controls the memory term $V_{k,i}$). We start by verifying the simple following

proposition, that ensures that the global variable $V_k$ corresponds to the mean of the local control variables $(V_{k,i})_{i \in [n]^\star}$.

**Proposition 12.** *For any $k \in [k_{\max}]$,*

$$V_k = \frac{1}{n} \sum_{i=1}^{n} V_{k,i} \ .$$

*Proof.* By definition of $V_0$, the property holds true when $k = 0$. Assume this holds true for $k \in [k_{\max} - 1]$. We write

$$V_{k+1} = V_k + \frac{\alpha}{n} \sum_{i=1}^{n} B_{k+1,i} \operatorname{Quant}(\Delta_{k+1,i})$$

$$= \frac{1}{n} \sum_{i=1}^{n} V_{k,i} + \frac{1}{n} \sum_{i=1}^{n} (V_{k+1,i} - V_{k,i})$$

$$= \frac{1}{n} \sum_{i=1}^{n} V_{k+1,i} \ .$$

This concludes the induction. $\qquad\qquad\qquad\qquad\qquad\qquad\qquad\qquad\qquad\qquad\square$

We now prove that the unbiased character of $H_k$ is preserved, and we provide a new control on its second order moment. Proposition 13 is Proposition 8 with $\omega$ replaced with $\omega_p$. When $p = 1$, Proposition 13 and Proposition 8 are the same.

**Proposition 13.** *Assume A6, A7 and A9. Set $\sigma^2 := n^{-1} \sum_{i=1}^{n} \sigma_i^2$. For any $k \in [k_{\max} - 1]$, we have*

$$\mathbb{E}\left[H_{k+1} | \mathcal{F}_k\right] = \mathsf{h}(\widehat{S}_k) \ ,$$

*and*

$$\mathbb{E}\left[\|H_{k+1} - \mathbb{E}\left[H_{k+1} | \mathcal{F}_k\right]\|^2 | \mathcal{F}_k\right] \leq \frac{\omega_p}{n} \frac{1}{n} \sum_{i=1}^{n} \mathbb{E}\left[\|\Delta_{k+1,i}\|^2 | \mathcal{F}_k\right] + \frac{\sigma^2}{n} \ ,$$

*where*

$$\omega_p := \frac{1-p}{p}(1+\omega) + \omega \ . \tag{31}$$

*Proof.* Let $k \in [k_{\max} - 1]$. By definition, we have

$$H_{k+1} = V_k + \frac{1}{np} \sum_{i=1}^{n} B_{k+1,i} \operatorname{Quant}(\Delta_{k+1,i})$$

where the Bernoulli random variables $\{B_{k+1,i}, i \in [n]^\star\}$ are independent with the same success probability $p$. By definition of the filtrations, we have $B_{k+1,i} \in \mathcal{F}_{k+1,i}$, $\operatorname{Quant}(\Delta_{k+1,i}) \in \mathcal{F}_{k+2/3,i}$, $V_k \in \mathcal{F}_k$ and $\Delta_{k+1,i} \in \mathcal{F}_{k+1/3,i}$; and the inclusions $\mathcal{F}_k \subset \mathcal{F}_{k+1/3,i} \subset \mathcal{F}_{k+2/3,i} \subset \mathcal{F}_{k+1,i}$. Therefore,

$$\mathbb{E}\left[H_{k+1} | \mathcal{F}_k\right] = V_k + \frac{1}{np} \sum_{i=1}^{n} \mathbb{E}\left[\mathbb{E}\left[B_{k+1,i} | \mathcal{F}_{k+2/3,i}\right] \operatorname{Quant}(\Delta_{k+1,i}) | \mathcal{F}_k\right]$$

$$= V_k + \frac{1}{n} \sum_{i=1}^{n} \mathbb{E}\left[\mathbb{E}\left[\operatorname{Quant}(\Delta_{k+1,i}) | \mathcal{F}_{k+1/3,i}\right] | \mathcal{F}_k\right] = V_k + \frac{1}{n} \sum_{i=1}^{n} \mathbb{E}\left[\Delta_{k+1,i} | \mathcal{F}_k\right]$$

$$= V_k + \frac{1}{n} \sum_{i=1}^{n} \left(\mathbb{E}\left[\mathsf{S}_{k+1,i} | \mathcal{F}_k\right] - \widehat{S}_k - V_{k,i}\right)$$

$$= \frac{1}{n} \sum_{i=1}^{n} \mathsf{h}_i(\widehat{S}_k) = \mathsf{h}(\widehat{S}_k) \ ,$$

where we used $\mathbb{E}\left[B_{k+1,i}\middle|\mathcal{F}_{k+2/3,i}\right] = p$ (see A9), A6, A7 and Proposition 12. This concludes the proof of the first statement of Proposition 13. For the second point, we write

$$H_{k+1} - \mathsf{h}(\widehat{S}_k) = \frac{1}{n}\sum_{i=1}^{n}\Xi_{k+1,i}$$

$$\Xi_{k+1,i} := \mathsf{S}_{k+1,i} - \mathbb{E}\left[\mathsf{S}_{k+1,i}\middle|\mathcal{F}_{k,i}^+\right]$$
$$+ \mathrm{Quant}(\Delta_{k+1,i}) - \mathbb{E}\left[\mathrm{Quant}(\Delta_{k+1,i})\middle|\mathcal{F}_{k+1/3,i}\right]$$
$$+ \frac{1}{p}\left(B_{k+1,i} - \mathbb{E}\left[B_{k+1,i}\middle|\mathcal{F}_{k+2/3,i}\right]\right)\mathrm{Quant}(\Delta_{k+1,i})\ ;$$

note indeed that $\mathsf{h}_i(\widehat{S}_k) = \mathbb{E}\left[\mathsf{S}_{k+1,i}\middle|\mathcal{F}_{k,i}^+\right] - \widehat{S}_k$, $\mathbb{E}\left[\mathrm{Quant}(\Delta_{k+1,i})\middle|\mathcal{F}_{k+1/3,i}\right] = \Delta_{k+1,i}$, $\Delta_{k+1,i} = V_{k,i} + \mathsf{S}_{k+1,i} - \widehat{S}_k$, $V_k = n^{-1}\sum_{i=1}^{n}V_{k,i}$ and $p = \mathbb{E}\left[B_{k+1,i}\middle|\mathcal{F}_{k+2/3,i}\right]$. Write $H_{k+1} - \mathsf{h}(\widehat{S}_k) = \frac{1}{n}\sum_{i=1}^{n}\Xi_{k+1,i}$. Since the workers are independent, we have

$$\mathbb{E}\left[\left\|H_{k+1} - \mathsf{h}(\widehat{S}_k)\right\|^2\middle|\mathcal{F}_k\right] = \frac{1}{n^2}\sum_{i=1}^{n}\mathbb{E}\left[\left\|\Xi_{k+1,i}\right\|^2\middle|\mathcal{F}_k\right]\ .$$

Fix $i \in [n]^\star$. $\Xi_{k+1,i}$ is the sum of three terms $\sum_{\ell=1}^{3}\Xi_{k+1,i,\ell}$ and observe that for any $\ell \neq \ell'$ we have

$$\mathbb{E}\left[\langle\Xi_{k+1,i,\ell},\Xi_{k+1,i,\ell'}\rangle\middle|\mathcal{F}_k\right] = 0\ .$$

Therefore $\mathbb{E}\left[\left\|\Xi_{k+1,i}\right\|^2\middle|\mathcal{F}_k\right] = \sum_{\ell=1}^{3}\mathbb{E}\left[\left\|\Xi_{k+1,i,\ell}\right\|^2\middle|\mathcal{F}_k\right]$. We have by A7

$$\mathbb{E}\left[\left\|\mathsf{S}_{k+1,i} - \mathbb{E}\left[\mathsf{S}_{k+1,i}\middle|\mathcal{F}_{k,i}^+\right]\right\|^2\middle|\mathcal{F}_k\right] \leq \sigma_i^2\ ;$$

by A6,

$$\mathbb{E}\left[\left\|\mathrm{Quant}(\Delta_{k+1,i}) - \mathbb{E}\left[\mathrm{Quant}(\Delta_{k+1,i})\middle|\mathcal{F}_{k+1/3,i}\right]\right\|^2\middle|\mathcal{F}_k\right] \leq \omega\mathbb{E}\left[\left\|\Delta_{k+1,i}\right\|^2\middle|\mathcal{F}_k\right]\ ;$$

and by A6 and A9

$$\mathbb{E}\left[\frac{1}{p^2}\left(B_{k+1,i} - \mathbb{E}\left[B_{k+1,i}\middle|\mathcal{F}_{k+2/3,i}\right]\right)^2\left\|\mathrm{Quant}(\Delta_{k+1,i})\right\|^2\middle|\mathcal{F}_k\right]$$
$$\leq \frac{1-p}{p}\mathbb{E}\left[\left\|\mathrm{Quant}(\Delta_{k+1,i})\right\|^2\middle|\mathcal{F}_k\right]$$
$$\leq \frac{1-p}{p}(1+\omega)\mathbb{E}\left[\left\|\Delta_{k+1,i}\right\|^2\middle|\mathcal{F}_k\right]\ .$$

This concludes the proof. $\square$

**Proposition 14.** *Assume A7 and set $\sigma^2 := n^{-1}\sum_{i=1}^{n}\sigma_i^2$. For any $k \in [k_{\max} - 1]$,*

$$\frac{1}{n}\sum_{i=1}^{n}\mathbb{E}\left[\left\|\Delta_{k+1,i}\right\|^2\middle|\mathcal{F}_k\right] \leq \frac{1}{n}\sum_{i=1}^{n}\left\|V_{k,i} - \mathsf{h}_i(\widehat{S}_k)\right\|^2 + \sigma^2\ .$$

The proof is on the same lines as the proof of Proposition 10 and is omitted.

Proposition 15 extends Proposition 11: the result is similar but with $\alpha$ replaced with $\alpha p$ and $\omega$ by $\omega_p$.

**Proposition 15.** *Assume A5, A6, A7 and A9; set $L^2 := n^{-1}\sum_{i=1}^{n}L_i^2$ and $\sigma^2 := n^{-1}\sum_{i=1}^{n}\sigma_i^2$. Choose $\alpha \in (0, 1/(1+\omega)]$. For any $k \geq 0$, define*

$$G_k := \frac{1}{n}\sum_{i=1}^{n}\left\|V_{k,i} - \mathsf{h}_i(\widehat{S}_k)\right\|^2\ .$$

*We have, for any $k \in [k_{\max} - 1]$*

$$\mathbb{E}\left[G_{k+1}|\mathcal{F}_k\right] \le \left(1 - \frac{\alpha p}{2} + 2\gamma_{k+1}^2 \frac{L^2}{\alpha p}\frac{\omega_p}{n}\right)G_k + 2\gamma_{k+1}^2\frac{L^2}{\alpha p}\|\mathsf{h}(\widehat{S}_k)\|^2$$
$$+ 2\left(\alpha p + \gamma_{k+1}^2\frac{L^2}{\alpha p}\frac{\omega_p}{n}\right)\sigma^2 \;,$$

*where $\omega_p$ is defined in Proposition 13.*

*Proof.* Let $i \in [n]^\star$. We follow the same line of the proof as Proposition 11: for any $\beta > 0$, using that $\|a+b\|^2 \le (1+\beta^2)\|a\|^2 + (1+\beta^{-2})\|b\|^2$, we have

$$\mathbb{E}\left[\|V_{k+1,i} - \mathsf{h}_i(\widehat{S}_{k+1})\|^2\Big|\mathcal{F}_k\right]$$
$$\le (1+\beta^{-2})\mathbb{E}\left[\|V_{k+1,i} - \mathsf{h}_i(\widehat{S}_k)\|^2\Big|\mathcal{F}_k\right] + (1+\beta^2)\mathbb{E}\left[\|\mathsf{h}_i(\widehat{S}_k) - \mathsf{h}_i(\widehat{S}_{k+1})\|^2\Big|\mathcal{F}_k\right]$$
$$\overset{A5}{\le} (1+\beta^{-2})\mathbb{E}\left[\|V_{k+1,i} - \mathsf{h}_i(\widehat{S}_k)\|^2\Big|\mathcal{F}_k\right] + (1+\beta^2)L_i^2\gamma_{k+1}^2\mathbb{E}\left[\|H_{k+1}\|^2\Big|\mathcal{F}_k\right]\;.$$

We then provide a control for $\mathbb{E}\left[\|V_{k+1,i} - \mathsf{h}_i(\widehat{S}_k)\|^2\Big|\mathcal{F}_k\right]$. Recall that:

$$V_{k+1,i} = V_{k,i} + \alpha\, B_{k+1,i}\mathrm{Quant}(\Delta_{k+1;i}).$$

We write $f(B_{k+1,i}) = f(1)\mathbb{1}_{B_{k+1,i}=1} + f(0)\mathbb{1}_{B_{k+1,i}=0}$ for any measurable positive function $f$; and then use $\mathbb{E}\left[\mathbb{1}_{B_{k+1,i}}\Big|\mathcal{F}_{k+2/3,i}\right] = p$ (see A9), $\mathrm{Quant}(\Delta_{k+1,i}), \widehat{S}_k, V_{k,i} \in \mathcal{F}_{k+2/3,i}$ . We get

$$\mathbb{E}\left[\|V_{k+1,i} - \mathsf{h}_i(\widehat{S}_k)\|^2|\mathcal{F}_k\right]$$
$$= p\mathbb{E}\left[\|V_{k,i} - \mathsf{h}_i(\widehat{S}_k) - \alpha\mathrm{Quant}(\Delta_{k+1,i})\|^2\Big|\mathcal{F}_k\right] + (1-p)\|V_{k,i} - \mathsf{h}_i(\widehat{S}_k)\|^2$$
$$\overset{(19)}{=} p(1-\alpha)\,\|V_{k,i} - \mathsf{h}_i(\widehat{S}_k)\|^2 + \alpha p\,\mathbb{E}\left[\|\mathsf{S}_{k+1,i} - \widehat{S}_k - \mathsf{h}_i(\widehat{S}_k)\|^2\Big|\mathcal{F}_k\right]$$
$$+ \alpha p\,(\alpha(1+\omega)-1)\,\mathbb{E}\left[\|\Delta_{k+1,i}\|^2\Big|\mathcal{F}_k\right] + (1-p)\,\|V_{k,i} - \mathsf{h}_i(\widehat{S}_k)\|^2$$
$$= (1-\alpha p)\,\|V_{k,i} - \mathsf{h}_i(\widehat{S}_k)\|^2$$
$$+ \alpha p\,\mathbb{E}\left[\|\mathsf{S}_{k+1,i} - \widehat{S}_k - \mathsf{h}_i(\widehat{S}_k)\|^2\Big|\mathcal{F}_k\right] + \alpha p\,(\alpha(1+\omega)-1)\,\mathbb{E}\left[\|\Delta_{k+1,i}\|^2\Big|\mathcal{F}_k\right]\;.$$

The end of the proof is identical to the proof of Proposition 11: we choose $\beta_p > 0$ such that $\beta_p^{-2} = 1$ if $\alpha p \ge 2/3$ and $\beta_p^{-2} = \frac{\alpha p}{2(1-\alpha p)}$ if $\alpha p \le 2/3$. We have

$$(1-\alpha p)(1+\beta_p^{-2}) \le 1 - \frac{\alpha p}{2}\;, \qquad (1+\beta_p^2) \le \frac{2}{\alpha p}\;, \qquad 1 \le 1+\beta_p^{-2} \le 2\;;$$

and this yields

$$\mathbb{E}\left[\|V_{k+1,i} - \mathsf{h}_i(\widehat{S}_{k+1})\|^2\Big|\mathcal{F}_k\right] \le \left(1 - \frac{\alpha p}{2}\right)\,\|V_{k,i} - \mathsf{h}_i(\widehat{S}_k)\|^2$$
$$+ 2\alpha p\,\mathbb{E}\left[\|\mathsf{S}_{k+1,i} - \bar{\mathsf{s}}_i \circ \mathsf{T}(\widehat{S}_k)\|^2\Big|\mathcal{F}_k\right] + \alpha p\,(\alpha(1+\omega)-1)\,\mathbb{E}\left[\|\Delta_{k+1,i}\|^2\Big|\mathcal{F}_k\right]$$
$$+ \frac{2}{\alpha p}L_i^2\gamma_{k+1}^2\mathbb{E}\left[\|H_{k+1}\|^2\Big|\mathcal{F}_k\right]\;.$$

By definition of the conditional expectation and Proposition 13 we have

$$\mathbb{E}\left[\|H_{k+1}\|^2\Big|\mathcal{F}_k\right] = \|\mathbb{E}\left[H_{k+1}|\mathcal{F}_k\right]\|^2 + \mathbb{E}\left[\|H_{k+1} - \mathbb{E}\left[H_{k+1}|\mathcal{F}_k\right]\|^2\Big|\mathcal{F}_k\right]$$
$$= \|\mathsf{h}(\widehat{S}_k)\|^2 + \mathbb{E}\left[\|H_{k+1} - \mathsf{h}(\widehat{S}_k)\|^2\Big|\mathcal{F}_k\right]\;.$$

Since $(\alpha(1+\omega)-1) \le 0$, using A7 and Proposition 13 again, we get:

$$\mathbb{E}\left[G_{k+1}|\mathcal{F}_k\right] \le \left(1 - \frac{\alpha p}{2}\right)G_k + 2\alpha p\sigma^2 + \frac{2}{\alpha p}L^2\gamma_{k+1}^2\frac{1}{n}\left(\sigma^2 + \omega_p\frac{1}{n}\sum_{i=1}^n\mathbb{E}\left[\|\Delta_{k+1,i}\|^2\Big|\mathcal{F}_k\right]\right)\;.$$

Finally, from Proposition 14,

$$\frac{1}{n}\sum_{i=1}^{n}\mathbb{E}\left[\|\Delta_{k+1,i}\|^2\big|\mathcal{F}_k\right] \leq G_k + \sigma^2 .$$

This concludes the proof.

$\square$

### D.4   Proof of Theorem 4

Throughout this proof, set

$$\omega_p := \frac{1-p}{p}(1+\omega) + \omega .$$

**Step 1: Upper bound on the decrement.** Let $k \geq 0$. Following the same lines as in the proof of Theorem 1, we have

$$\mathbb{E}\left[W(\widehat{S}_{k+1})\Big|\mathcal{F}_k\right]$$
$$\leq W(\widehat{S}_k) - \gamma_{k+1}v_{\min}\left(1 - \gamma_{k+1}\frac{L_{\dot{W}}}{2v_{\min}}\right)\|h(\widehat{S}_k)\|^2 + \gamma_{k+1}^2\frac{L_{\dot{W}}}{2}\mathbb{E}\left[\|H_{k+1} - \mathbb{E}\left[H_{k+1}|\mathcal{F}_k\right]\|^2\big|\mathcal{F}_k\right] .$$

Applying Proposition 13 and Proposition 14, we obtain that

$$\mathbb{E}\left[W(\widehat{S}_{k+1})\Big|\mathcal{F}_k\right] \leq W(\widehat{S}_k) - \gamma_{k+1}v_{\min}\left(1 - \gamma_{k+1}\frac{L_{\dot{W}}}{2v_{\min}}\right)\|h(\widehat{S}_k)\|^2$$
$$+ \gamma_{k+1}^2\frac{L_{\dot{W}}}{2}\frac{\omega_p}{n}G_k + \gamma_{k+1}^2\frac{L_{\dot{W}}}{2n}(1+\omega_p)\sigma^2 , \quad (32)$$

where

$$G_k := \frac{1}{n}\sum_{i=1}^{n}\|V_{k,i} - h_i(\widehat{S}_k)\|^2 .$$

**Step 2: Maximal learning rate $\gamma_{k+1}$ when $\omega \neq 0$.** From Proposition 11, for any non-increasing positive sequence $\{\gamma_k, k \in [k_{\max} - 1]\}$ such that

$$\gamma_{k+1}^2 \leq \frac{\alpha^2 p^2}{8L^2}\frac{n}{\omega_p},$$

and for any positive sequence $\{C_k, k \in [k_{\max} - 1]\}$, it holds

$$C_{k+1}\mathbb{E}\left[G_{k+1}|\mathcal{F}_k\right] \leq C_{k+1}\left(1 - \frac{\alpha p}{4}\right)G_k$$
$$+ C_{k+1}\gamma_{k+1}^2\frac{2}{\alpha p}L^2\|h(\widehat{S}_k)\|^2 + 2C_{k+1}\left(\alpha p + \gamma_{k+1}^2\frac{L^2}{\alpha p}\frac{1+\omega_p}{n}\right)\sigma^2 . \quad (33)$$

Combining equations (32) and (33), we thus have

$$\mathbb{E}[W(\widehat{S}_{k+1})|\mathcal{F}_k] + C_{k+1}\mathbb{E}\left[G_{k+1}|\mathcal{F}_k\right] \leq W(\widehat{S}_k) + C_k G_k$$
$$- \gamma_{k+1}v_{\min}\left(1 - \gamma_{k+1}\frac{L_{\dot{W}}}{2v_{\min}} - \frac{C_{k+1}}{v_{\min}}\gamma_{k+1}\frac{2}{\alpha p}L^2\right)\|h(\widehat{S}_k)\|^2$$
$$+ \left(\gamma_{k+1}^2\frac{L_{\dot{W}}}{2}\frac{\omega_p}{n} - C_k + C_{k+1} - C_{k+1}\frac{\alpha p}{4}\right)G_k$$
$$+ \left\{2\alpha p C_{k+1} + \gamma_{k+1}^2\frac{(1+\omega_p)}{n}\left(\frac{L_{\dot{W}}}{2} + 2C_{k+1}\frac{L^2}{\alpha p}\right)\right\}\sigma^2 .$$

We choose the sequence $\{C_k\}$ as follows:

$$C_k := \gamma_k^2\frac{2L_{\dot{W}}}{\alpha p}\frac{\omega_p}{n} ;$$

the sequence satisfies $C_{k+1} \leq C_k$ (since $\gamma_{k+1} \leq \gamma_k$) and $\gamma_{k+1}^2 L_{\dot{W}} \omega_p/(2n) \leq C_{k+1}\alpha p/4$. By convention, $\gamma_0 \in [\gamma_1, +\infty)$. Therefore

$$\mathbb{E}[\mathrm{W}(\widehat{S}_{k+1})|\mathcal{F}_k] + \gamma_{k+1}^2 \frac{2L_{\dot{W}}}{\alpha p}\frac{\omega_p}{n}\mathbb{E}[G_{k+1}|\mathcal{F}_k] \leq \mathrm{W}(\widehat{S}_k) + \gamma_k^2 \frac{2L_{\dot{W}}}{\alpha p}\frac{\omega_p}{n}G_k$$
$$- \gamma_{k+1}v_{\min}\left(1 - \gamma_{k+1}\frac{L_{\dot{W}}}{2v_{\min}}\left\{1 + 8\gamma_{k+1}^2\frac{\omega_p}{\alpha^2 p^2 n}L^2\right\}\right)\|\mathsf{h}(\widehat{S}_k)\|^2$$
$$+ 4\gamma_{k+1}^2 L_{\dot{W}}\frac{\omega_p}{n}\left\{1 + \frac{(1+\omega_p)}{8\omega_p}\left(1 + \gamma_{k+1}^2 8\frac{L^2}{\alpha^2 p^2}\frac{\omega_p}{n}\right)\right\}\sigma^2 .$$

**Step 3: Computing the expectation.** Let us apply the expectations, sum from $k = 0$ to $k = k_{\max} - 1$, and divide by $k_{\max}$. This yields

$$\frac{v_{\min}}{k_{\max}}\sum_{k=0}^{k_{\max}-1}\gamma_{k+1}\left(1 - \gamma_{k+1}\frac{L_{\dot{W}}}{2v_{\min}}\left\{1 + 8\gamma_{k+1}^2\frac{\omega_p}{\alpha^2 p^2 n}L^2\right\}\right)\|\mathsf{h}(\widehat{S}_k)\|^2$$
$$\leq k_{\max}^{-1}\left\{\mathrm{W}(\widehat{S}_0) + \gamma_0^2\frac{2L_{\dot{W}}}{\alpha}\frac{\omega_p}{n}G_0 - \mathbb{E}\left[\mathrm{W}(\widehat{S}_{k_{\max}})\right] - \gamma_{k_{\max}}^2\frac{2L_{\dot{W}}}{\alpha p}\frac{\omega_p}{n}\mathbb{E}[G_{k_{\max}}]\right\}$$
$$+ 4L_{\dot{W}}\frac{\omega_p}{n}\frac{1}{k_{\max}}\sum_{k=0}^{k_{\max}-1}\gamma_{k+1}^2\left\{1 + \frac{(1+\omega_p)}{8\omega}\left(1 + \gamma_{k+1}^2 8\frac{L^2}{\alpha^2 p^2}\frac{\omega_p}{n}\right)\right\}\sigma^2 .$$

We now focus on the case when $\gamma_{k+1} = \gamma$ for any $k \geq 0$. Denote by $K$ a uniform random variable on $[k_{\max} - 1]$, independent of the path $\{\widehat{S}_k, k \in [k_{\max}]\}$. Since $\gamma^2 \leq \alpha^2 p^2 n/(8L^2\omega_p)$, we have

$$1 + 8\gamma^2\frac{\omega_p}{\alpha^2 p^2 n}L^2 \leq 2 .$$

This yields

$$v_{\min}\gamma\left(1 - \gamma\frac{L_{\dot{W}}}{v_{\min}}\right)\mathbb{E}\left[\|\mathsf{h}(\widehat{S}_K)\|^2\right]$$
$$\leq k_{\max}^{-1}\left\{\mathrm{W}(\widehat{S}_0) + \gamma^2\frac{2L_{\dot{W}}}{\alpha p}\frac{\omega_p}{n}G_0 - \mathbb{E}\left[\mathrm{W}(\widehat{S}_{k_{\max}})\right] - \gamma^2\frac{2L_{\dot{W}}}{\alpha p}\frac{\omega_p}{n}\mathbb{E}[G_{k_{\max}}]\right\}$$
$$+ 4L_{\dot{W}}\frac{\omega_p}{n}\gamma^2\left\{1 + \frac{(1+\omega_p)}{4\omega_p}\right\}\sigma^2 .$$

Note that $4(1 + (1+\omega_p)/(4\omega_p)) = (5\omega_p + 1)/\omega_p$.

**Step 4. Conclusion (when $\omega \neq 0$)** By choosing $V_{0,i} = \mathsf{h}_i$ for any $i \in [n]^\star$, we have $G_0 = 0$. The roots of $\gamma \mapsto \gamma(1 - \gamma L_{\dot{W}}/v_{\min})$ are $0$ and $v_{\min}/L_{\dot{W}}$ and its maximum is reached at $v_{\min}/(2L_{\dot{W}})$: this function is increasing on $(0, v_{\min}/(2L_{\dot{W}})]$. We therefore choose $\gamma \in (0, \gamma_{\max}(\alpha)]$ where

$$\gamma_{\max}(\alpha) := \min\left(\frac{v_{\min}}{2L_{\dot{W}}}; \frac{\alpha p}{2\sqrt{2}L}\frac{\sqrt{n}}{\sqrt{\omega_p}}\right)$$

Finally, since $\alpha \in (0, 1/(1+\omega)]$, we choose $\alpha = 1/(1+\omega)$. This yields

$$\gamma_{\max} := \min\left(\frac{v_{\min}}{2L_{\dot{W}}}; \frac{p}{2\sqrt{2}L}\frac{\sqrt{n}}{\sqrt{\omega_p}(1+\omega)}\right) .$$

# E Convergence Analysis of `VR-FedEM`

The assumptions A1 to A3 hold throughout this section. We will use the notations

$$L_i^2 := m^{-1}\sum_{j=1}^m L_{ij}^2 , \qquad L^2 := n^{-1}\sum_{i=1}^n L_i^2 , \tag{34}$$

where $L_{ij}$ is defined in A8, and

$$\mathsf{h}_i(s) := \frac{1}{m}\sum_{j=1}^m \bar{\mathsf{s}}_{ij} \circ \mathsf{T}(s) - s , \qquad \mathsf{h}(s) := \frac{1}{n}\sum_{i=1}^n \mathsf{h}_i(s) .$$

### E.1 Notations and elementary result

Let us define the following filtrations: for any $i \in [n]^\star$ and $t \in [k_{\mathrm{out}}]^\star$, $k \in [k_{\max} - 1]$, set

$$\mathcal{F}_{1,0,i} = \mathcal{F}_{1,0,i}^+ := \sigma\left(\widehat{S}_{\mathrm{init}}; V_{1,0,i}\right) , \qquad \mathcal{F}_{1,0} := \bigvee_{i=1}^{n} \mathcal{F}_{1,0,i} ,$$

$$\mathcal{F}_{t,k+1/2,i} := \mathcal{F}_{t,k,i}^+ \vee \sigma\left(\mathcal{B}_{t,k+1,i}\right) , \qquad \mathcal{F}_{t,k+1,i} := \mathcal{F}_{t,k+1/2,i} \vee \sigma\left(\mathrm{Quant}(\Delta_{t,k+1,i})\right) ,$$

$$\mathcal{F}_{t,k+1} := \bigvee_{i=1}^{n} \mathcal{F}_{t,k+1,i} , \qquad \mathcal{F}_{t,k+1,i}^+ := \mathcal{F}_{t,k+1} .$$

With these notations, for $t \in [k_{\mathrm{out}}]^\star$, $k \in [k_{\max} - 1]$ and $i \in [n]^\star$, $\widehat{S}_{t,k+1} \in \mathcal{F}_{t,k+1,i}^+$, $S_{t,k+1,i} \in \mathcal{F}_{t,k+1/2,i}, \Delta_{t,k+1,i} \in \mathcal{F}_{t,k+1/2,i}$, $V_{t,k+1,i} \in \mathcal{F}_{t,k+1,i}$, $\widehat{S}_{t,k+1} \in \mathcal{F}_{t,k+1}$ $H_{t,k+1} \in \mathcal{F}_{t,k+1}$, and $V_{t,k+1} \in \mathcal{F}_{t,k+1}$.

### E.2 Computed conditional expectations complexity.

In this section, we provide a discussion on the computed conditional expectations complexity $\mathcal{K}_{\mathrm{CE}}$ that was removed from the main text due to spaces constraints.

The number of calls to conditional expectations (i.e., computing $\bar{s}_{ij}$) to perform $k_{\mathrm{out}}$ outer steps of algorithm 2, each composed of $k_{\mathrm{in}}$ inner iterations, with $n$ workers and mini-batches of size b is

$$nmk_{\mathrm{out}} + n(2\mathrm{b})k_{\mathrm{in}}k_{\mathrm{out}} = nk_{\mathrm{in}}k_{\mathrm{out}}\left(\frac{m}{k_{\mathrm{in}}} + 2b\right) ;$$

it corresponds to one full pass on the data at the beginning of each outer loop and two batches of size b on each worker $i \in [n]^\star$, at each inner iteration. In oder to reach an accuracy $\epsilon$, we need $(k_{\mathrm{in}}k_{\mathrm{out}}\gamma)^{-1} = O(\epsilon)$ with the parameter choices in Theorem 3 (esp. on b) we thus have

$$\mathcal{K}_{\mathrm{CE}}(\epsilon) = O\left(\frac{n}{\epsilon\gamma}\left(\frac{m}{k_{\mathrm{in}}} + 2\frac{k_{\mathrm{in}}}{(1+\omega)^2}\right)\right) .$$

This complexity is minimized with $k_{\mathrm{in}} = (1+\omega)\sqrt{m/2}$. We then obtain an overall complexity $\mathcal{K}_{\mathrm{CE}}$ of $O\left(\frac{\sqrt{m}}{\epsilon\gamma}\frac{n}{(1+\omega)}\right)$. We stress the following two points:

1. **Dependency w.r.t. $m$**: the complexity increases as $\sqrt{m}$. For $n = 1, \omega = 0$, this yields a scaling equal to $\sqrt{m}\epsilon^{-1}$ that corresponds to the optimal $\mathcal{K}_{\mathrm{CE}}$ of SPIDER-EM [10];
2. **Dependency w.r.t. $\omega$**. Again, the dependency on $\omega$ depends on the regime for $\gamma$. In the (worst case regime), $\gamma = O(\sqrt{n}/\omega^{3/2})$, we get

$$\mathcal{K}_{\mathrm{CE}}(\epsilon) = O\left(\frac{\sqrt{m}\sqrt{n}\sqrt{\omega}}{\epsilon}\right)$$

when $\epsilon \to 0$ and $\omega, n \to \infty$, which corresponds to a sublinear increase w.r.t. $\omega$ (that compares to a linear increase in the cost of each communication).

### E.3 Preliminary results

#### E.3.1 Results on the minibatch $\mathcal{B}_{t,k+1}$

The proof of the following proposition is given in [10, Lemma 4]. It establishes the bias and the variance of the sum along the random set of indices $\mathcal{B}_{t,k+1}$ conditionally to the past.

**Proposition 16.** *Let $\mathcal{B}$ be a minibatch of size b, sampled at random (with or without replacement) from $[m]^\star$. It holds for any $i \in [n]^\star$ and $s \in \mathbb{R}^q$,*

$$\mathbb{E}\left[\frac{1}{b}\sum_{j \in \mathcal{B}} \bar{s}_{ij} \circ \mathsf{T}(s)\right] = \frac{1}{m}\sum_{j=1}^{m} \bar{s}_{ij} \circ \mathsf{T}(s) ;$$

*and for any $s, s' \in \mathbb{R}^q$,*

$$\mathbb{E}\left[\left\|\frac{1}{b}\sum_{j\in\mathcal{B}}\{\bar{s}_{ij}\circ\mathsf{T}(s)-s)-(\bar{s}_{ij}\circ\mathsf{T}(s')-s')\}\right.\right.$$

$$\left.\left.-\frac{1}{m}\sum_{j=1}^{m}\{(\bar{s}_{ij}\circ\mathsf{T}(s)-s)-(\bar{s}_{ij}\circ\mathsf{T}(s')-s')\}\right\|^2\right]\leq\frac{L_i^2}{b}\|s-s'\|^2\ .$$

### E.3.2    Results on the statistics $\mathsf{S}_{t,k,i}$

Proposition 17 shows that for $k\geq 1$, $\mathsf{S}_{t,k+1,i}$ is a biased approximation of $m^{-1}\sum_{j=1}^{m}\bar{s}_{ij}\circ\mathsf{T}(\widehat{S}_{t,k})$; and this bias is canceled at the beginning of each outer loop since $\mathsf{S}_{t,1,i}=m^{-1}\sum_{j=1}^{m}\bar{s}_{ij}\circ\mathsf{T}(\widehat{S}_{t,0})$. Corollary 18 establishes an upper bound for the conditional variance and the mean squared error of $\mathsf{S}_{t,k+1,i}$.

Let us comment the definition of $\mathsf{S}_{t,k+1,i}$. For any $t\in[k_{\mathrm{out}}]^\star$, $k\in[k_{\mathrm{in}}-1]$ and $i\in[n]^\star$,

$$\mathsf{S}_{t,k+1,i}=\frac{1}{b}\sum_{j\in\mathcal{B}_{t,k+1,i}}\bar{s}_{ij}\circ\mathsf{T}(\widehat{S}_{t,k})+\Upsilon_{t,k+1,i}\ ,\quad\Upsilon_{t,k+1,i}:=\mathsf{S}_{t,k,i}-\frac{1}{b}\sum_{j\in\mathcal{B}_{t,k+1,i}}\bar{s}_{ij}\circ\mathsf{T}(\widehat{S}_{t,k-1})\ .$$

It is easily seen that

$$\Upsilon_{t,k+1,i}=\Upsilon_{t,k,i}+\frac{1}{b}\sum_{j\in\mathcal{B}_{t,k,i}}\bar{s}_{ij}\circ\mathsf{T}(\widehat{S}_{t,k-1})-\frac{1}{b}\sum_{j\in\mathcal{B}_{t,k+1,i}}\bar{s}_{ij}\circ\mathsf{T}(\widehat{S}_{t,k-1})\ ,$$

and since $\Upsilon_{t,1,i}=\mathsf{S}_{t,0,i}-b^{-1}\sum_{j\in\mathcal{B}_{t,1,i}}\bar{s}_{ij}\circ\mathsf{T}(\widehat{S}_{t,-1})$, we have by using Proposition 17,

$$\Upsilon_{t,k,i}=\sum_{\ell=1}^{k}\left\{\frac{1}{b}\sum_{j\in\mathcal{B}_{t,\ell,i}}\bar{s}_{ij}\circ\mathsf{T}(\widehat{S}_{t,\ell-1})-\frac{1}{b}\sum_{j\in\mathcal{B}_{t,\ell+1,i}}\bar{s}_{ij}\circ\mathsf{T}(\widehat{S}_{t,\ell-1})\right\}$$

$$+\frac{1}{m}\sum_{j=1}^{m}\bar{s}_{ij}\circ\mathsf{T}(\widehat{S}_{t,-1})-\frac{1}{b}\sum_{j\in\mathcal{B}_{t,1,i}}\bar{s}_{ij}\circ\mathsf{T}(\widehat{S}_{t,-1})\ .$$

We have $\mathbb{E}\left[\Upsilon_{t,k,i}|\mathcal{F}_{t,0}\right]=0$ but conditionally to the past $\mathcal{F}_{t,k-1,i}^+$, the variable $\Upsilon_{t,k,i}$ is *not* centered.

**Proposition 17.** *For any $t\in[k_{\mathrm{out}}]^\star$ and $i\in[n]^\star$,*

$$\mathsf{S}_{t,1,i}-\frac{1}{m}\sum_{j=1}^{m}\bar{s}_{ij}\circ\mathsf{T}(\widehat{S}_{t,0})=\mathsf{S}_{t,0,i}-\frac{1}{m}\sum_{j=1}^{m}\bar{s}_{ij}\circ\mathsf{T}(\widehat{S}_{t,-1})=0\ .$$

*For any $t\in[k_{\mathrm{out}}]^\star$, $k\in[k_{\mathrm{in}}-1]$ and $i\in[n]^\star$, we have*

$$\mathbb{E}\left[\mathsf{S}_{t,k+1,i}\Big|\mathcal{F}_{t,k,i}^+\right]-\frac{1}{m}\sum_{j=1}^{m}\bar{s}_{ij}\circ\mathsf{T}(\widehat{S}_{t,k})=\mathsf{S}_{t,k,i}-\frac{1}{m}\sum_{j=1}^{m}\bar{s}_{ij}\circ\mathsf{T}(\widehat{S}_{t,k-1})\ .$$

*Proof.* Let $t\in[k_{\mathrm{out}}]^\star$ and $i\in[n]^\star$. We have by definition of $\mathsf{S}_{t,1,i}$ and $\mathsf{S}_{t,0,i}$

$$\mathsf{S}_{t,1,i}=\mathsf{S}_{t,0,i}+b^{-1}\sum_{j\in\mathcal{B}_{t,1,i}}\left(\bar{s}_{ij}\circ\mathsf{T}(\widehat{S}_{t,0})-\bar{s}_{ij}\circ\mathsf{T}(\widehat{S}_{t,-1})\right)=\mathsf{S}_{t,0,i}=\frac{1}{m}\sum_{j=1}^{m}\bar{s}_{ij}\circ\mathsf{T}(\widehat{S}_{t,0})$$

where we used that $\widehat{S}_{t,0}=\widehat{S}_{t,-1}$.

Let $k\in[k_{\mathrm{in}}-1]$. By definition of $\mathsf{S}_{t,k+1,i}$, we have

$$\mathsf{S}_{t,k+1,i}-\mathsf{S}_{t,k,i}=b^{-1}\sum_{j\in\mathcal{B}_{t,k+1,i}}\left(\bar{s}_{ij}\circ\mathsf{T}(\widehat{S}_{t,k})-\bar{s}_{ij}\circ\mathsf{T}(\widehat{S}_{t,k-1})\right)\ .$$

Since $\widehat{S}_{t,k}, \widehat{S}_{t,k-1} \in \mathcal{F}_{t,k,i}^+$, we have by Proposition 16

$$\mathbb{E}\left[\mathsf{b}^{-1}\sum_{j\in\mathcal{B}_{t,k+1,i}}\left(\bar{\mathsf{s}}_{ij}\circ\mathsf{T}(\widehat{S}_{t,k})-\bar{\mathsf{s}}_{ij}\circ\mathsf{T}(\widehat{S}_{t,k-1})\right)\Big|\mathcal{F}_{t,k,i}^+\right]$$

$$=\frac{1}{m}\sum_{j=1}^{m}\left(\bar{\mathsf{s}}_{ij}\circ\mathsf{T}(\widehat{S}_{t,k})-\bar{\mathsf{s}}_{ij}\circ\mathsf{T}(\widehat{S}_{t,k-1})\right)$$

and the proof follows. $\qquad\square$

**Corollary 18** (of Proposition 17). *Assume A8. For any $t\in[k_{\mathrm{out}}]^\star$, $k\in[k_{\mathrm{in}}-1]$ and $i\in[n]^\star$,*

$$\mathbb{E}\left[\|\mathsf{S}_{t,k+1,i}-\mathbb{E}\left[\mathsf{S}_{t,k+1,i}|\mathcal{F}_{t,k,i}\right]\|^2\big|\mathcal{F}_{t,k}\right]\le\frac{L_i^2}{\mathsf{b}}\gamma_{t,k}^2\|H_{t,k}\|^2\ ,$$

$$\mathbb{E}\left[\|\mathsf{S}_{t,k+1,i}-\frac{1}{m}\sum_{j=1}^{m}\bar{\mathsf{s}}_{ij}\circ\mathsf{T}(\widehat{S}_{t,k})\|^2\Big|\mathcal{F}_{t,0}\right]\le\frac{L_i^2}{\mathsf{b}}\sum_{\ell=1}^{k}\gamma_{t,\ell}^2\mathbb{E}\left[\|H_{t,\ell}\|^2\big|\mathcal{F}_{t,0}\right]\ .$$

*By convention, $H_{t,0}=0$ and $\sum_{\ell=1}^{0}a_\ell=0$.*

*Proof.* Note that $\widehat{S}_{t,k},\widehat{S}_{t,k-1}\in\mathcal{F}_{t,k}$. By Proposition 17, we have

$$\mathbb{E}\left[\|\mathsf{S}_{t,k+1,i}-\mathbb{E}\left[\mathsf{S}_{t,k+1,i}\big|\mathcal{F}_{t,k,i}^+\right]\|^2\big|\mathcal{F}_{t,k}\right]$$

$$=\mathbb{E}\left[\Big\|\frac{1}{\mathsf{b}}\sum_{j\in\mathcal{B}_{t,k+1,i}}\left(\bar{\mathsf{s}}_{ij}\circ\mathsf{T}(\widehat{S}_{t,k})-\bar{\mathsf{s}}_{ij}\circ\mathsf{T}(\widehat{S}_{t,k-1})\right)-\frac{1}{m}\sum_{j=1}^{m}\left(\bar{\mathsf{s}}_{ij}\circ\mathsf{T}(\widehat{S}_{t,k})-\bar{\mathsf{s}}_{ij}\circ\mathsf{T}(\widehat{S}_{t,k-1})\right)\Big\|^2\Big|\mathcal{F}_{t,k}\right]\ .$$

By Proposition 16, it holds

$$\mathbb{E}\left[\|\mathsf{S}_{t,k+1,i}-\mathbb{E}\left[\mathsf{S}_{t,k+1,i}|\mathcal{F}_{t,k,i}\right]\|^2|\mathcal{F}_{t,k}\right]\le\frac{L_i^2}{\mathsf{b}}\|\widehat{S}_{t,k}-\widehat{S}_{t,k-1}\|^2=\frac{L_i^2}{\mathsf{b}}\gamma_{t,k}^2\|H_{t,k}\|^2\ ;$$

with the convention that $H_{t,0}=0$ since $\widehat{S}_{t,0}=\widehat{S}_{t,-1}$. The proof of the first statement is concluded.

For the second statement, by definition of the conditional expectation and since $\widehat{S}_{t,k}\in\mathcal{F}_{t,k}\subset\mathcal{F}_{t,k,i}^+$, it holds

$$\mathbb{E}\left[\|\mathsf{S}_{t,k+1,i}-\frac{1}{m}\sum_{j=1}^{m}\bar{\mathsf{s}}_{ij}\circ\mathsf{T}(\widehat{S}_{t,k})\|^2\Big|\mathcal{F}_{t,k}\right]=\mathbb{E}\left[\|\mathsf{S}_{t,k+1,i}-\mathbb{E}\left[\mathsf{S}_{t,k+1,i}\big|\mathcal{F}_{t,k,i}^+\right]\|^2\big|\mathcal{F}_{t,k}\right]$$

$$+\mathbb{E}\left[\|\mathbb{E}\left[\mathsf{S}_{t,k+1,i}\big|\mathcal{F}_{t,k,i}^+\right]-\frac{1}{m}\sum_{j=1}^{m}\bar{\mathsf{s}}_{ij}\circ\mathsf{T}(\widehat{S}_{t,k})\|^2\Big|\mathcal{F}_{t,k}\right]\ .$$

By Proposition 17,

$$\left\|\mathbb{E}\left[\mathsf{S}_{t,k+1,i}\big|\mathcal{F}_{t,k,i}^+\right]-\frac{1}{m}\sum_{j=1}^{m}\bar{\mathsf{s}}_{ij}\circ\mathsf{T}(\widehat{S}_{t,k})\right\|^2=\left\|\mathsf{S}_{t,k,i}-\frac{1}{m}\sum_{j=1}^{m}\bar{\mathsf{s}}_{ij}\circ\mathsf{T}(\widehat{S}_{t,k-1})\right\|^2\ .$$

Hence, by using $\mathsf{S}_{t,1,i}-m^{-1}\sum_{j=1}^{m}\bar{\mathsf{s}}_{ij}\circ\mathsf{T}(\widehat{S}_{t,0})=0$ (see Proposition 17), we have

$$\mathbb{E}\left[\left\|\mathsf{S}_{t,k+1,i}-\frac{1}{m}\sum_{j=1}^{m}\bar{\mathsf{s}}_{ij}\circ\mathsf{T}(\widehat{S}_{t,k})\right\|^2\Big|\mathcal{F}_{t,0}\right]$$

$$\le\frac{L_i^2}{\mathsf{b}}\gamma_{t,k}^2\mathbb{E}\left[\|H_{t,k}\|^2\big|\mathcal{F}_{t,0}\right]+\mathbb{E}\left[\left\|\mathsf{S}_{t,k,i}-\frac{1}{m}\sum_{j=1}^{m}\bar{\mathsf{s}}_{ij}\circ\mathsf{T}(\widehat{S}_{t,k-1})\right\|^2\Big|\mathcal{F}_{t,0}\right]$$

$$\le\frac{L_i^2}{\mathsf{b}}\sum_{\ell=1}^{k}\gamma_{t,\ell}^2\mathbb{E}\left[\|H_{t,\ell}\|^2\big|\mathcal{F}_{t,0}\right]\ .$$

$\qquad\square$

### E.3.3 Results on $\Delta_{t,k+1,i}$

Proposition 19 provides an upper bound for the mean value of the conditional variance of $\Delta_{t,k+1,\cdot}$ and for its $L_2$-moment. Proposition 20 prepares the control of the varianc of the random field $H_{t,k+1}$ upon noting that

$$H_{t,k+1} - \mathbb{E}\left[H_{t,k+1}|\mathcal{F}_{t,k}\right] = \frac{1}{n}\sum_{i=1}^{n}\left(\mathrm{Quant}(\Delta_{t,k+1,i}) - \mathbb{E}\left[\Delta_{t,k+1,i}|\mathcal{F}_{t,k}\right]\right) .$$

**Proposition 19.** *Assume A8. For any* $t \in [k_{\mathrm{out}}]^\star$ *and* $k \in [k_{\mathrm{in}} - 1]$,

$$\frac{1}{n}\sum_{i=1}^{n}\mathbb{E}\left[\|\Delta_{t,k+1,i}\|^2\big|\mathcal{F}_{t,0}\right]$$

$$\leq 2\frac{L^2}{\mathsf{b}}\sum_{\ell=1}^{k}\gamma_{t,\ell}^2\mathbb{E}\left[\|H_{t,\ell}\|^2\big|\mathcal{F}_{t,0}\right] + \frac{2}{n}\sum_{i=1}^{n}\mathbb{E}\left[\|\mathsf{h}_i(\widehat{S}_{t,k}) - V_{t,k,i}\|^2\Big|\mathcal{F}_{t,0}\right] .$$

*In addition,*

$$\frac{1}{n}\sum_{i=1}^{n}\mathbb{E}\left[\|\Delta_{t,k+1,i} - \mathbb{E}\left[\Delta_{t,k+1,i}|\mathcal{F}_{t,k}\right]\|^2\big|\mathcal{F}_{t,k}\right] \leq \frac{L^2}{\mathsf{b}}\gamma_{t,k}^2\|H_{t,k}\|^2 .$$

*Proof.* Let $i \in [n]^\star$, $t \in [k_{\mathrm{out}}]^\star$ and $k \in [k_{\mathrm{in}} - 1]$. We write

$$\Delta_{t,k+1,i} = \mathsf{S}_{t,k+1,i} - \frac{1}{m}\sum_{j=1}^{m}\bar{\mathsf{s}}_{ij}\circ\mathsf{T}(\widehat{S}_{t,k}) + \mathsf{h}_i(\widehat{S}_{t,k}) - V_{t,k,i} .$$

When $k = 0$, we have $\mathsf{S}_{t,1,i} - \frac{1}{m}\sum_{j=1}^{m}\bar{\mathsf{s}}_{ij}\circ\mathsf{T}(\widehat{S}_{t,0}) = 0$ (see Proposition 17) so that $\Delta_{t,1,i} = \mathsf{h}_i(\widehat{S}_{t,0}) - V_{t,0,i}$. For $k \geq 1$, we write

$$\mathbb{E}\left[\|\Delta_{t,k+1,i}\|^2\big|\mathcal{F}_{t,0}\right] \leq 2\mathbb{E}\left[\left\|\mathsf{S}_{t,k+1,i} - \frac{1}{m}\sum_{j=1}^{m}\bar{\mathsf{s}}_{ij}\circ\mathsf{T}(\widehat{S}_{t,k})\right\|^2\Bigg|\mathcal{F}_{t,0}\right]$$

$$+ 2\mathbb{E}\left[\|\mathsf{h}_i(\widehat{S}_{t,k}) - V_{t,k,i}\|^2\Big|\mathcal{F}_{t,0}\right]$$

and the proof of the first statement is concluded by Corollary 18.

By definition of $\Delta_{t,k+1,i}$, it holds

$$\Delta_{t,k+1,i} - \mathbb{E}\left[\Delta_{t,k+1,i}|\mathcal{F}_{t,k}\right] = \mathsf{S}_{t,k+1,i} - \mathbb{E}\left[\mathsf{S}_{t,k+1,i}|\mathcal{F}_{t,k}\right] . \tag{35}$$

The proof is concluded by (35) and Corollary 18. $\qquad\square$

**Proposition 20.** *Assume A6 and A8. For any* $t \in [k_{\mathrm{out}}]^\star$ *and* $k \in [k_{\mathrm{in}} - 1]$,

$$\frac{1}{n}\sum_{i=1}^{n}\mathbb{E}\left[\|\mathrm{Quant}(\Delta_{t,k+1,i}) - \mathbb{E}\left[\Delta_{t,k+1,i}|\mathcal{F}_{t,k}\right]\|^2\big|\mathcal{F}_{t,0}\right] \leq \frac{\omega}{n}\sum_{i=1}^{n}\mathbb{E}\left[\|\Delta_{t,k+1,i}\|^2\big|\mathcal{F}_{t,0}\right]$$

$$+ \frac{L^2}{\mathsf{b}}\gamma_{t,k}^2\mathbb{E}\left[\|H_{t,k}\|^2\big|\mathcal{F}_{t,0}\right] .$$

*Proof.* Let $i \in [n]^\star$, $t \in [k_{\mathrm{out}}]^\star$ and $k \in [k_{\mathrm{in}} - 1]$. We write

$$\mathrm{Quant}(\Delta_{t,k+1,i}) - \mathbb{E}\left[\Delta_{t,k+1,i}|\mathcal{F}_{t,k}\right] = \mathrm{Quant}(\Delta_{t,k+1,i}) - \Delta_{t,k+1,i} + \Delta_{t,k+1,i} - \mathbb{E}\left[\Delta_{t,k+1,i}|\mathcal{F}_{t,k}\right] ;$$

and use the property

$$\mathbb{E}\left[\|\mathrm{Quant}(\Delta_{t,k+1,i}) - \mathbb{E}\left[\Delta_{t,k+1,i}|\mathcal{F}_{t,k}\right]\|^2\big|\mathcal{F}_{t,0}\right] = \mathbb{E}\left[\|\mathrm{Quant}(\Delta_{t,k+1,i}) - \Delta_{t,k+1,i}\|^2\big|\mathcal{F}_{t,0}\right]$$

$$+ \mathbb{E}\left[\|\Delta_{t,k+1,i} - \mathbb{E}\left[\Delta_{t,k+1,i}|\mathcal{F}_{t,k}\right]\|^2\big|\mathcal{F}_{t,0}\right] .$$

By A6 and $\mathcal{F}_{t,k} \subset \mathcal{F}_{t,k+1/2,i}$, we have

$$\mathbb{E}\left[\|\mathrm{Quant}(\Delta_{t,k+1,i}) - \Delta_{t,k+1,i}\|^2\big|\mathcal{F}_{t,0}\right]$$
$$= \mathbb{E}\left[\mathbb{E}\left[\|\mathrm{Quant}(\Delta_{t,k+1,i}) - \Delta_{t,k+1,i}\|^2\big|\mathcal{F}_{t,k+1/2,i}\right]\big|\mathcal{F}_{t,0}\right] \le \omega\mathbb{E}\left[\|\Delta_{t,k+1,i}\|^2\big|\mathcal{F}_{t,0}\right] \,;$$

in addition, by Proposition 19,

$$n^{-1}\sum_{i=1}^n \mathbb{E}\left[\|\Delta_{t,k+1,i} - \mathbb{E}\left[\Delta_{t,k+1,i}|\mathcal{F}_{t,k}\right]\|^2\big|\mathcal{F}_{t,0}\right] \le \frac{L^2}{\mathsf{b}}\gamma_{t,k}^2\mathbb{E}\left[\|H_{t,k}\|^2\big|\mathcal{F}_{t,0}\right] \,.$$

This concludes the proof. $\qquad\qquad\square$

### E.3.4 Results on the memory terms $V_{t,k+1,i}$

Lemma 21 proves that the memory term $V_{t,k+1}$ computed by the central server is the mean value of the local $V_{t,k+1,i}$ computed by each worker #$i$. Proposition 22 establishes a contraction-like inequality on the mean quantity $n^{-1}\sum_{i=1}^n \|V_{t,k+1,i} - \mathsf{h}_i(\widehat{S}_{t,k+1})\|^2$ thus providing the intuition that $V_{t,k+1,i}$ approximates $\mathsf{h}_i(\widehat{S}_{t,k+1})$.

**Lemma 21.** *For any $t \in [k_{\mathrm{out}}]^\star$ and $k \in [k_{\mathrm{in}} - 1]$,*

$$V_{t,k+1} = \frac{1}{n}\sum_{i=1}^n V_{t,k+1,i} \,, \qquad V_{t,0} = \frac{1}{n}\sum_{i=1}^n V_{t,0,i} \,.$$

*Proof.* The proof is by induction on $t$ and $k$. Consider the case $t = 1$. When $k = 0$, the property holds true by Line 1 in algorithm 2. Assume that the property holds for $k \le k_{\mathrm{in}} - 2$. Then by definition of $V_{1,k+1}$ and by the induction assumption:

$$V_{1,k+1} = V_{1,k} + \alpha\frac{1}{n}\sum_{i=1}^n \mathrm{Quant}(\Delta_{1,k+1,i}) = \frac{1}{n}\sum_{i=1}^n (V_{1,k,i} + \alpha\mathrm{Quant}(\Delta_{1,k+1,i}))$$
$$= \frac{1}{n}\sum_{i=1}^n V_{1,k+1,i} \,.$$

By Lines 18 and 21 in algorithm 2 and by the induction on $k$, we obtain

$$V_{2,0} = V_{1,k_{\mathrm{in}}} = \frac{1}{n}\sum_{i=1}^n V_{1,k_{\mathrm{in}},i} = \frac{1}{n}\sum_{i=1}^n V_{2,0,i} \,.$$

Assume that for $t \in [k_{\mathrm{out}} - 1]^\star$ we have $V_{t,0} = n^{-1}\sum_{i=1}^n V_{t,0,i}$. As in the case $t = 1$, we prove by induction on $k$ that for any $k \in [k_{\mathrm{in}} - 1]$, $V_{t,k+1} = n^{-1}\sum_{i=1}^n V_{t,k+1,i}$ (details are omitted). This implies, by using Lines 18 and 21 of algorithm 2, that

$$V_{t+1,0} = V_{t,k_{\mathrm{in}}} = \frac{1}{n}\sum_{i=1}^n V_{t,k_{\mathrm{in}},i} = \frac{1}{n}\sum_{i=1}^n V_{t+1,0,i} \,.$$

This concludes the induction. $\qquad\qquad\square$

**Proposition 22.** *Assume A6 and A8. Let $\alpha \in \left(0, (1 + \omega)^{-1}\right]$. For any $t \in [k_{\mathrm{out}}]^\star$, $k \in [k_{\mathrm{in}} - 1]$ and $i \in [n]^\star$, it holds*

$$\mathbb{E}\left[V_{t,k+1,i}\big|\mathcal{F}_{t,k+1/2,i}\right] = (1 - \alpha)V_{t,k,i} + \alpha\left(\mathsf{S}_{t,k+1,i} - \widehat{S}_{t,k}\right) \,,$$

*Define for $t \in [k_{\mathrm{out}}]^\star$ and $k \in [k_{\mathrm{in}}]$*

$$G_{t,k} := \frac{1}{n}\sum_{i=1}^n \|V_{t,k,i} - \mathsf{h}_i(\widehat{S}_{t,k})\|^2 \,.$$

*We have*

$$\mathbb{E}\left[G_{t,k+1}|\mathcal{F}_{t,0}\right] \leq (1 - \alpha/2)\mathbb{E}\left[G_{t,k}|\mathcal{F}_{t,0}\right]$$

$$+ \frac{2}{\alpha}L^2\gamma_{t,k+1}^2\mathbb{E}\left[\|H_{t,k+1}\|^2|\mathcal{F}_{t,0}\right] + 2\alpha\frac{L^2}{\mathsf{b}}\sum_{\ell=1}^{k}\gamma_{t,\ell}^2\mathbb{E}\left[\|H_{t,\ell}\|^2|\mathcal{F}_{t,0}\right]$$

$$+ \alpha\left(\alpha(1+\omega) - 1\right)\frac{1}{n}\sum_{i=1}^{n}\mathbb{E}\left[\|\Delta_{t,k+1,i}\|^2|\mathcal{F}_{t,0}\right] .$$

*Proof.* Let $t \in [k_{\mathrm{out}}]^\star$, $k \in [k_{\mathrm{in}} - 1]$ and $i \in [n]^\star$. By definition of $V_{t,k+1,i}$, $\Delta_{t,k+1,i}$ and by A6, it holds

$$\mathbb{E}\left[V_{t,k+1,i}|\mathcal{F}_{t,k+1/2,i}\right] = V_{t,k,i} + \alpha\mathbb{E}\left[\mathrm{Quant}(\Delta_{t,k+1,i})|\mathcal{F}_{t,k+1/2,i}\right]$$

$$= V_{t,k,i} + \alpha\left(\mathsf{S}_{t,k+1,i} - \widehat{S}_{t,k} - V_{t,k,i}\right) .$$

This concludes the proof of the first statement. For the second statement, we write for any $\beta > 0$:

$$\|V_{t,k+1,i} - \mathsf{h}_i(\widehat{S}_{t,k+1})\|^2 \leq (1 + \beta^2)\|\mathsf{h}_i(\widehat{S}_{t,k+1}) - \mathsf{h}_i(\widehat{S}_{t,k})\|^2 + (1 + \beta^{-2})\|V_{t,k+1,i} - \mathsf{h}_i(\widehat{S}_{t,k})\|^2$$

$$\leq (1 + \beta^2)L_i^2\gamma_{t,k+1}^2\|H_{t,k+1}\|^2 + (1 + \beta^{-2})\|V_{t,k+1,i} - \mathsf{h}_i(\widehat{S}_{t,k})\|^2 , \tag{36}$$

where we used A8 and the definition of $\widehat{S}_{t,k+1}$ in the last inequality. For any $s \in \mathbb{R}^q$

$$\mathbb{E}\left[\|V_{t,k+1,i} - s\|^2|\mathcal{F}_{t,k+1/2,i}\right] = \mathbb{E}\left[\|V_{t,k+1,i} - \mathbb{E}\left[V_{t,k+1,i}|\mathcal{F}_{t,k+1/2,i}\right]\|^2|\mathcal{F}_{t,k+1/2,i}\right]$$

$$+ \|\mathbb{E}\left[V_{t,k+1,i} - s|\mathcal{F}_{t,k+1/2,i}\right]\|^2 . \tag{37}$$

On one hand,

$$\|V_{t,k+1,i} - \mathbb{E}\left[V_{t,k+1,i}|\mathcal{F}_{t,k+1/2,i}\right]\|^2 = \alpha^2\|\mathrm{Quant}(\Delta_{t,k+1,i}) - \mathbb{E}\left[\mathrm{Quant}(\Delta_{t,k+1,i})|\mathcal{F}_{t,k+1/2,i}\right]\|^2$$

and by A6,

$$\mathbb{E}\left[\|V_{t,k+1,i} - \mathbb{E}\left[V_{t,k+1,i}|\mathcal{F}_{t,k+1/2,i}\right]\|^2|\mathcal{F}_{t,k+1/2,i}\right] \leq \alpha^2\omega\|\Delta_{t,k+1,i}\|^2 . \tag{38}$$

On the other hand, for any $s \in \mathbb{R}^q$, and using Lemma 6

$$\|\mathbb{E}\left[V_{t,k+1,i} - s|\mathcal{F}_{t,k+1/2,i}\right]\|^2 = \|(1 - \alpha)\left(V_{t,k,i} - s\right) + \alpha\left(\mathsf{S}_{t,k+1,i} - \widehat{S}_{t,k} - s\right)\|^2$$

$$= (1 - \alpha)\|V_{t,k,i} - s\|^2 + \alpha\|\mathsf{S}_{t,k+1,i} - \widehat{S}_{t,k} - s\|^2 - \alpha(1 - \alpha)\|V_{t,k,i} - \mathsf{S}_{t,k+1,i} + \widehat{S}_{t,k}\|^2$$

$$= (1 - \alpha)\|V_{t,k,i} - s\|^2 + \alpha\|\mathsf{S}_{t,k+1,i} - \widehat{S}_{t,k} - s\|^2 - \alpha(1 - \alpha)\|\Delta_{t,k+1,i}\|^2 . \tag{39}$$

Let us combine (36) to (39), the last one being applied with $s \leftarrow \mathsf{h}_i(\widehat{S}_{t,k}) \in \mathcal{F}_{t,k,i}^+ \subseteq \mathcal{F}_{t,k+1/2,i}$. Since

$$\|\mathsf{S}_{t,k+1,i} - \widehat{S}_{t,k} - \mathsf{h}_i(\widehat{S}_{t,k})\|^2 = \|\mathsf{S}_{t,k+1,i} - \frac{1}{m}\sum_{j=1}^{m}\bar{\mathsf{s}}_{ij}\circ\mathsf{T}(\widehat{S}_{t,k})\|^2 ,$$

we write

$$\mathbb{E}\left[\|V_{t,k+1,i} - \mathsf{h}_i(\widehat{S}_{t,k+1})\|^2|\mathcal{F}_{t,k}\right] \leq (1 + \beta^2)L_i^2\gamma_{t,k+1}^2\mathbb{E}\left[\|H_{t,k+1}\|^2|\mathcal{F}_{t,k}\right]$$

$$+ (1 + \beta^{-2})\left\{\alpha^2\omega\mathbb{E}\left[\|\Delta_{t,k+1,i}\|^2|\mathcal{F}_{t,k}\right] + (1 - \alpha)\|V_{t,k,i} - \mathsf{h}_i(\widehat{S}_{t,k})\|^2\right.$$

$$+ \alpha\mathbb{E}\left[\|\mathsf{S}_{t,k+1,i} - \frac{1}{m}\sum_{j=1}^{m}\bar{\mathsf{s}}_{ij}\circ\mathsf{T}(\widehat{S}_{t,k})\|^2\Big|\mathcal{F}_{t,k}\right] - \alpha(1 - \alpha)\mathbb{E}\left[\|\Delta_{t,k+1,i}\|^2|\mathcal{F}_{t,k}\right]\right\} .$$

Choose $\beta^2 > 0$ such that

$$\beta^{-2} := \begin{cases} 1 & \text{if } \alpha \geq 2/3 \\ \frac{\alpha}{2(1-\alpha)} & \text{if } \alpha \leq 2/3 \end{cases}$$

This implies that

$$(1 + \beta^{-2})(1 - \alpha) \leq 1 - \frac{\alpha}{2} \ , \qquad 1 + \beta^2 \leq \frac{2}{\alpha} \ , \qquad 1 + \beta^{-2} \leq 2 \ .$$

Hence,

$$\mathbb{E}\left[\|V_{t,k+1,i} - \mathsf{h}_i(\widehat{S}_{t,k+1})\|^2 \Big| \mathcal{F}_{t,k}\right] \leq (1 - \alpha/2)\|V_{t,k,i} - \mathsf{h}_i(\widehat{S}_{t,k})\|^2$$

$$+ \frac{2}{\alpha} L_i^2 \gamma_{t,k+1}^2 \mathbb{E}\left[\|H_{t,k+1}\|^2 \big| \mathcal{F}_{t,k}\right] + 2\alpha \mathbb{E}\left[\|\mathsf{S}_{t,k+1,i} - \frac{1}{m}\sum_{j=1}^m \bar{\mathsf{s}}_{ij} \circ \mathsf{T}(\widehat{S}_{t,k})\|^2 \Big| \mathcal{F}_{t,k}\right]$$

$$+ \alpha\left(\alpha\omega - 1 + \alpha\right) \mathbb{E}\left[\|\Delta_{t,k+1,i}\|^2 \big| \mathcal{F}_{t,k}\right] \ ;$$

(in the last equality, we use $1 + \beta^{-2} \geq 1$ since $\alpha\omega - 1 + \alpha \leq 0$). Finally, by using Corollary 18, we have

$$\mathbb{E}\left[\|V_{t,k+1,i} - \mathsf{h}_i(\widehat{S}_{t,k+1})\|^2 \Big| \mathcal{F}_{t,0}\right] \leq (1 - \alpha/2)\mathbb{E}\left[\|V_{t,k,i} - \mathsf{h}_i(\widehat{S}_{t,k})\|^2 \Big| \mathcal{F}_{t,0}\right]$$

$$+ \frac{2}{\alpha} L_i^2 \gamma_{t,k+1}^2 \mathbb{E}\left[\|H_{t,k+1}\|^2 \big| \mathcal{F}_{t,0}\right] + 2\alpha\frac{L_i^2}{\mathsf{b}}\sum_{\ell=1}^k \gamma_{t,\ell}^2 \mathbb{E}\left[\|H_{t,\ell}\|^2 \big| \mathcal{F}_{t,0}\right]$$

$$+ \alpha\left(\alpha\omega - 1 + \alpha\right) \mathbb{E}\left[\|\Delta_{t,k+1,i}\|^2 \big| \mathcal{F}_{t,0}\right] \ .$$

The proof is concluded. $\qquad\qquad\qquad\qquad\qquad\qquad\qquad\qquad\qquad\qquad\qquad\qquad\qquad\square$

### E.3.5   Results on the random field $H_{t,k+1}$

Proposition 23 shows that the random field $H_{t,k+1}$ is a biased approximation of the field $\mathsf{h}(\widehat{S}_{t,k})$, and this bias is canceled at the beginning of each outer loop. Observe also that the bias exists even when there is no compression: when $\omega = 0$ (so that $\mathrm{Quant}(u) = u$) we have

$$\mathbb{E}\left[H_{t,k+1}|\mathcal{F}_{t,k}\right] - \mathsf{h}(\widehat{S}_{t,k}) = H_{t,k} - \mathsf{h}(\widehat{S}_{t,k-1}) \ ,$$

and the bias is again canceled at the beginning of each outer loop. Proposition 24 provides an upper bound for the variance and the mean squared error of the random field $H_{t,k+1}$. In the case of no compression ($\omega = 0$) and of a single worker ($n = 1$) so that VR-FedEM is SPIDER-EM, Proposition 24 retrieves the variance and the mean squared error of the random field $H_{t,k+1}$ in SPIDER-EM (see [10, Proposition 13]).

**Proposition 23.** *Assume A6. For any* $t \in [k_{\mathrm{out}}]^\star$, $\mathbb{E}\left[H_{t,2}|\mathcal{F}_{t,0}\right] - \mathsf{h}(\widehat{S}_{t,1}) = \mathbb{E}\left[H_{t,1}|\mathcal{F}_{t,0}\right] - \mathsf{h}(\widehat{S}_{t,0}) = 0$ *and for any* $k \in [k_{\mathrm{in}} - 1]^\star$,

$$\mathbb{E}\left[H_{t,k+1}|\mathcal{F}_{t,k}\right] - \mathsf{h}(\widehat{S}_{t,k}) = H_{t,k} - \mathsf{h}(\widehat{S}_{t,k-1}) - n^{-1}\sum_{i=1}^n \left(\mathrm{Quant}(\Delta_{t,k,i}) - \Delta_{t,k,i}\right)$$

$$= n^{-1}\sum_{i=1}^n \left(\mathbb{E}\left[\mathsf{S}_{t,k+1,i}|\mathcal{F}_{t,k}\right] - m^{-1}\sum_{j=1}^m \bar{\mathsf{s}}_{ij} \circ \mathsf{T}(\widehat{S}_{t,k})\right) \ .$$

*Proof.* Let $t \in [k_{\mathrm{out}}]^\star$.

• By definition of $H_{t,1}$ and $\Delta_{t,1,i}$, by A6 and by Lemma 21, we have

$$\mathbb{E}\left[H_{t,1}|\mathcal{F}_{t,0}\right] = V_{t,0} + n^{-1}\sum_{i=1}^n \mathbb{E}\left[\mathrm{Quant}(\Delta_{t,1,i})|\mathcal{F}_{t,0}\right] = V_{t,0} + n^{-1}\sum_{i=1}^n \mathbb{E}\left[\Delta_{t,1,i}|\mathcal{F}_{t,0}\right]$$

$$= V_{t,0} + n^{-1}\sum_{i=1}^n \left(\mathbb{E}\left[\mathsf{S}_{t,1,i}|\mathcal{F}_{t,0}\right] - \widehat{S}_{t,0} - V_{t,0,i}\right)$$

$$= n^{-1}\sum_{i=1}^n \mathbb{E}\left[\mathsf{S}_{t,1,i}|\mathcal{F}_{t,0}\right] - \widehat{S}_{t,0} \ .$$

By Proposition 17 $n^{-1}\sum_{i=1}^{n}\mathbb{E}\left[\mathsf{S}_{t,1,i}|\mathcal{F}_{t,0}\right]-\widehat{S}_{t,0}=\mathsf{h}(\widehat{S}_{t,0})$.

$\bullet$ Consider the case $k=1$. We have by definition of $H_{t,2}$

$$\mathbb{E}\left[H_{t,2}|\mathcal{F}_{t,1}\right]-\mathsf{h}(\widehat{S}_{t,1})=\frac{1}{n}\sum_{i=1}^{n}\left(\mathbb{E}\left[\mathsf{S}_{t,2,i}|\mathcal{F}_{t,1}\right]-m^{-1}\sum_{j=1}^{m}\bar{\mathsf{s}}_{ij}\circ\mathsf{T}(\widehat{S}_{t,1})\right);$$

Proposition 17 concludes the proof.

$\bullet$ Let $k\geq 2$. As in the case $k=0$, we have

$$\mathbb{E}\left[H_{t,k+1}|\mathcal{F}_{t,k}\right]=V_{t,k}+n^{-1}\sum_{i=1}^{n}\mathbb{E}\left[\mathrm{Quant}(\Delta_{t,k+1,i})|\mathcal{F}_{t,k}\right]=V_{t,k}+n^{-1}\sum_{i=1}^{n}\mathbb{E}\left[\Delta_{t,k+1,i}|\mathcal{F}_{t,k}\right]$$

$$=V_{t,k}+n^{-1}\sum_{i=1}^{n}\left(\mathbb{E}\left[\mathsf{S}_{t,k+1,i}|\mathcal{F}_{t,k}\right]-\widehat{S}_{t,k}-V_{t,k,i}\right)$$

$$=n^{-1}\sum_{i=1}^{n}\mathbb{E}\left[\mathsf{S}_{t,k+1,i}|\mathcal{F}_{t,k}\right]-\widehat{S}_{t,k},$$

so that

$$\mathbb{E}\left[H_{t,k+1}|\mathcal{F}_{t,k}\right]-\mathsf{h}(\widehat{S}_{t,k})=n^{-1}\sum_{i=1}^{n}\left(\mathbb{E}\left[\mathsf{S}_{t,k+1,i}|\mathcal{F}_{t,k}\right]-m^{-1}\sum_{j=1}^{m}\bar{\mathsf{s}}_{ij}\circ\mathsf{T}(\widehat{S}_{t,k})\right).\qquad(40)$$

By Proposition 17, upon noting that $\mathcal{F}_{t,k}\subset\mathcal{F}_{t,k,i}^{+}$ and $\mathsf{S}_{t,k,i},\widehat{S}_{t,k-1}\in\mathcal{F}_{t,k}$, we have

$$n^{-1}\sum_{i=1}^{n}\mathbb{E}\left[\mathsf{S}_{t,k+1,i}|\mathcal{F}_{t,k}\right]-m^{-1}\sum_{j=1}^{m}\bar{\mathsf{s}}_{ij}\circ\mathsf{T}(\widehat{S}_{t,k})=n^{-1}\sum_{i=1}^{n}\left(\mathsf{S}_{t,k,i}-m^{-1}\sum_{j=1}^{m}\bar{\mathsf{s}}_{ij}\circ\mathsf{T}(\widehat{S}_{t,k-1})\right).$$
$$(41)$$

On the other hand, observe that

$$H_{t,k}=V_{t,k-1}+n^{-1}\sum_{i=1}^{n}\mathrm{Quant}(\Delta_{t,k,i})$$

$$=V_{t,k-1}+n^{-1}\sum_{i=1}^{n}\mathsf{S}_{t,k,i}-\widehat{S}_{t,k-1}-n^{-1}\sum_{i=1}^{n}V_{t,k-1,i}+n^{-1}\sum_{i=1}^{n}\left(\mathrm{Quant}(\Delta_{t,k,i})-\Delta_{t,k,i}\right)$$

$$=n^{-1}\sum_{i=1}^{n}\mathsf{S}_{t,k,i}-\widehat{S}_{t,k-1}+n^{-1}\sum_{i=1}^{n}\left(\mathrm{Quant}(\Delta_{t,k,i})-\Delta_{t,k,i}\right),$$

where we used Lemma 21. This yields

$$H_{t,k}-\mathsf{h}(\widehat{S}_{t,k-1})$$
$$=n^{-1}\sum_{i=1}^{n}\left(\mathsf{S}_{t,k,i}-m^{-1}\sum_{j=1}^{m}\bar{\mathsf{s}}_{ij}\circ\mathsf{T}(\widehat{S}_{t,k-1})\right)+n^{-1}\sum_{i=1}^{n}\left(\mathrm{Quant}(\Delta_{t,k,i})-\Delta_{t,k,i}\right).\quad(42)$$

The proof is concluded by combining (40), (41) and (42). $\qquad\qquad\square$

**Proposition 24.** *Assume A6 and A8. For any $t\in[k_{\mathrm{out}}]^{\star}$,*

$$\mathbb{E}\left[\|H_{t,1}-\mathsf{h}(\widehat{S}_{t,0})\|^{2}\Big|\mathcal{F}_{t,0}\right]\leq\frac{\omega}{n}\left(\frac{1}{n}\sum_{i=1}^{n}\|V_{t,0,i}-\mathsf{h}_{i}(\widehat{S}_{t,0})\|^{2}\right),$$

*and for any $k\in[k_{\mathrm{in}}-1]^{\star}$,*

$$\mathbb{E}\left[\|H_{t,k+1}-\mathsf{h}(\widehat{S}_{t,k})\|^{2}\Big|\mathcal{F}_{t,0}\right]\leq\frac{\omega}{n}\frac{1}{n}\sum_{i=1}^{n}\mathbb{E}\left[\|\Delta_{t,k+1,i}\|^{2}\big|\mathcal{F}_{t,0}\right]+\frac{L^{2}}{n\mathsf{b}}\sum_{\ell=1}^{k}\gamma_{t,\ell}^{2}\mathbb{E}\left[\|H_{t,\ell}\|^{2}\big|\mathcal{F}_{t,0}\right],$$

$$\mathbb{E}\left[\|\mathbb{E}\left[H_{t,k+1}|\mathcal{F}_{t,k}\right]-\mathsf{h}(\widehat{S}_{t,k})\|^{2}\Big|\mathcal{F}_{t,0}\right]\leq\frac{L^{2}}{n\mathsf{b}}\sum_{\ell=1}^{k-1}\gamma_{t,\ell}^{2}\mathbb{E}\left[\|H_{t,\ell}\|^{2}\big|\mathcal{F}_{t,0}\right].$$

*Proof.* • **Case** $k = 1$**.** From Proposition 23 and the definition of $H_{t,1}$, we have

$$H_{t,1} - \mathsf{h}(\widehat{S}_{t,0}) = H_{t,1} - \mathbb{E}\left[\,[H_{t,1}|\mathcal{F}_{t,0}\,\right] = n^{-1}\sum_{i=1}^{n}\left(\text{Quant}(\Delta_{t,1,i}) - \mathbb{E}\left[\,\text{Quant}(\Delta_{t,1,i})|\mathcal{F}_{t,0}\,\right]\right)$$

$$= n^{-1}\sum_{i=1}^{n}\left(\text{Quant}(\Delta_{t,1,i}) - \mathbb{E}\left[\,\Delta_{t,1,i}|\mathcal{F}_{t,0}\,\right]\right)\;,$$

where we used $\mathbb{E}\left[\,\text{Quant}(\Delta_{t,1,i})\big|\mathcal{F}_{t,1/2,i}\,\right] = \Delta_{t,1,i}$ and $\mathcal{F}_{t,0} \subset \mathcal{F}_{t,1/2,i}$ in the last equality. In addition, since $\widehat{S}_{t,0} = \widehat{S}_{t,-1}$, we have (see Proposition 17)

$$\mathsf{S}_{t,1,i} = \mathsf{S}_{t,0,i} = \mathsf{h}_i(\widehat{S}_{t,0}) + \widehat{S}_{t,0}\;.$$

Hence,

$$\Delta_{t,1,i} = \mathsf{S}_{t,1,i} - \widehat{S}_{t,0} - V_{t,0,i} = \mathsf{h}_i(\widehat{S}_{t,0}) - V_{t,0,i}\;.$$

Therefore, $\mathbb{E}\left[\,\Delta_{t,1,i}|\mathcal{F}_{t,0}\,\right] = \Delta_{t,1,i} = \mathsf{h}_i(\widehat{S}_{t,0}) - V_{t,0,i}$. Since the workers are independent, we write

$$\mathbb{E}\left[\,\|H_{t,1} - \mathsf{h}(\widehat{S}_{t,0})\|^2\big|\mathcal{F}_{t,0}\,\right] = \frac{1}{n^2}\sum_{i=1}^{n}\mathbb{E}\left[\,\|\text{Quant}(\mathsf{h}_i(\widehat{S}_{t,0}) - V_{t,0,i}) - \left(\mathsf{h}_i(\widehat{S}_{t,0}) - V_{t,0,i}\right)\|^2\big|\mathcal{F}_{t,0}\,\right]\;.$$

By A6, this yields

$$\mathbb{E}\left[\,\|H_{t,1} - \mathsf{h}(\widehat{S}_{t,0})\|^2\big|\mathcal{F}_{t,0}\,\right] \leq \frac{\omega}{n}\frac{1}{n}\sum_{i=1}^{n}\|\mathsf{h}_i(\widehat{S}_{t,0}) - V_{t,0,i}\|^2\;.$$

• **Case** $k \geq 1$**.** Let $t \in [k_{\text{out}}]^\star$ and $k \in [k_{\text{in}} - 1]^\star$. We write

$$\mathbb{E}\left[\,\|H_{t,k+1} - \mathsf{h}(\widehat{S}_{t,k})\|^2\big|\mathcal{F}_{t,0}\,\right] = \mathbb{E}\left[\,\|H_{t,k+1} - \mathbb{E}\left[\,H_{t,k+1}|\mathcal{F}_{t,k}\,\right]\|^2\big|\mathcal{F}_{t,0}\,\right]$$

$$+ \mathbb{E}\left[\,\|\mathbb{E}\left[\,H_{t,k+1}|\mathcal{F}_{t,k}\,\right] - \mathsf{h}(\widehat{S}_{t,k})\|^2\big|\mathcal{F}_{t,0}\,\right]\;. \quad (43)$$

Let us first consider the bias term. From Proposition 17, Proposition 23 and the definition of $\mathsf{S}_{t,k+1,i}$ (remember that $\mathsf{S}_{t,k,i}$, $\widehat{S}_{t,k}$ and $\widehat{S}_{t,k-1}$ are in $\mathcal{F}_{t,k,i}^+ \supset \mathcal{F}_{t,k}$), it holds

$$\mathbb{E}\left[\,\left\|\mathbb{E}\left[\,H_{t,k+1}|\mathcal{F}_{t,k}\,\right] - \mathsf{h}(\widehat{S}_{t,k})\right\|^2\big|\mathcal{F}_{t,0}\,\right]$$

$$= \mathbb{E}\left[\,\left\|n^{-1}\sum_{i=1}^{n}(\mathbb{E}\left[\,\mathsf{S}_{t,k+1,i}|\mathcal{F}_{t,k}\,\right] - m^{-1}\sum_{j=1}^{m}\bar{\mathsf{s}}_{ij}\circ\mathsf{T}(\widehat{S}_{t,k}))\right\|^2\big|\mathcal{F}_{t,0}\,\right]$$

$$\leq \mathbb{E}\left[\,\left\|n^{-1}\sum_{i=1}^{n}(\mathsf{S}_{t,k,i} - m^{-1}\sum_{j=1}^{m}\bar{\mathsf{s}}_{ij}\circ\mathsf{T}(\widehat{S}_{t,k-1}))\right\|^2\big|\mathcal{F}_{t,0}\,\right]\;.$$

By Proposition 17 again, the RHS is zero when $k = 1$; when $k \geq 2$, by Corollary 18 and the independence of the workers, we have yields

$$\mathbb{E}\left[\,\|\mathbb{E}\left[\,H_{t,k+1}|\mathcal{F}_{t,k}\,\right] - \mathsf{h}(\widehat{S}_{t,k})\|^2\big|\mathcal{F}_{t,0}\,\right] \leq \frac{L^2}{n\mathsf{b}}\sum_{\ell=1}^{k-1}\gamma_{t,\ell}^2\mathbb{E}\left[\,\|H_{t,\ell}\|^2\big|\mathcal{F}_{t,0}\,\right]\;. \quad (44)$$

Let us now consider the variance term. We have from the definition of $H_{t,k+1}$ and A6

$$H_{t,k+1} - \mathbb{E}\left[\,H_{t,k+1}|\mathcal{F}_{t,k}\,\right] = \frac{1}{n}\sum_{i=1}^{n}\left(\text{Quant}(\Delta_{t,k+1,i}) - \mathbb{E}\left[\,\Delta_{t,k+1,i}|\mathcal{F}_{t,k}\,\right]\right)$$

and here again, by the independence of the workers

$$\mathbb{E}\left[\,\|H_{t,k+1} - \mathbb{E}\left[\,H_{t,k+1}|\mathcal{F}_{t,k}\,\right]\|^2\big|\mathcal{F}_{t,0}\,\right]$$

$$\leq \frac{1}{n^2}\sum_{i=1}^{n}\mathbb{E}\left[\,\|\text{Quant}(\Delta_{t,k+1,i}) - \mathbb{E}\left[\,\Delta_{t,k+1,i}|\mathcal{F}_{t,k}\,\right]\|^2\big|\mathcal{F}_{t,0}\,\right]\;. \quad (45)$$

The proof follows from (43) to (45) and Proposition 20. $\qquad\square$

### E.4 Proof of Theorem 3

Theorem 3 is a corollary of the more general following proposition.

**Proposition 25.** *Assume A1 to 3, A4, A6 and A8. Set $L^2 := n^{-1}m^{-1}\sum_{i=1}^{n}\sum_{j=1}^{m}L_{ij}^2$. Let $\{\widehat{S}_{t,k}, t \in [k_{\mathrm{out}}]^\star, k \in [k_{\mathrm{in}} - 1]\}$ be given by algorithm 2 run with any $\alpha \leq 1/(1 + \omega)$, and $\mathsf{b} \geq 1$, with $V_{1,0,i} = \mathsf{h}_i(\widehat{S}_{1,0})$ for any $i \in [n]^\star$. Let $(\tau, K)$ be a uniform random variable on $[k_{\mathrm{out}}]^\star \times [k_{\mathrm{in}} - 1]$, independent of $\{\widehat{S}_{t,k}, t \in [k_{\mathrm{out}}]^\star, k \in [k_{\mathrm{in}} - 1]\}$. Then, it holds*

$$v_{\min}\left(1 - \gamma\Lambda_\star\right)\mathbb{E}\left[\|H_{\tau,K+1}\|^2\right] \leq \gamma^{-1}k_{\mathrm{in}}^{-1}k_{\mathrm{out}}^{-1}\left(\mathbb{E}\left[\mathrm{W}(\widehat{S}_{1,0})\right] - \min\mathrm{W}\right),$$

*where*

$$\Lambda_\star := \frac{L_{\dot{\mathrm{W}}}}{2v_{\min}} + 2\sqrt{2}\frac{v_{\max}}{v_{\min}}\frac{L}{\sqrt{n}\alpha}\left(\omega + \frac{k_{\mathrm{in}}\alpha^2}{8\mathsf{b}}(1 + 10\omega)\right)^{1/2}.$$

The proof of Theorem 3 from Proposition 25 (which corresponds to particular choices of $\mathsf{b}$, $\alpha$, etc. is detailed in Appendix E.5).

### E.4.1 Control of $H_{\tau,K}$

Let $t \in [k_{\mathrm{out}}]^\star$ and $k \in [k_{\mathrm{in}} - 1]$. By A4, we have

$$\mathrm{W}(\widehat{S}_{t,k+1}) \leq \mathrm{W}(\widehat{S}_{t,k}) + \left\langle \nabla\mathrm{W}(\widehat{S}_{t,k}), \widehat{S}_{t,k+1} - \widehat{S}_{t,k}\right\rangle + \frac{L_{\dot{\mathrm{W}}}}{2}\|\widehat{S}_{t,k+1} - \widehat{S}_{t,k}\|^2.$$

Since $\widehat{S}_{t,k+1} - \widehat{S}_{t,k} = \gamma_{t,k+1}H_{t,k+1}$, we have using again A4

$$\mathrm{W}(\widehat{S}_{t,k+1}) \leq \mathrm{W}(\widehat{S}_{t,k}) - \gamma_{t,k+1}\left\langle B(\widehat{S}_{t,k})\mathsf{h}(\widehat{S}_{t,k}), H_{t,k+1}\right\rangle + \frac{L_{\dot{\mathrm{W}}}}{2}\gamma_{t,k+1}^2\|H_{t,k+1}\|^2.$$

We have the inequality, for any $\beta > 0$:

$$-\langle Bh, H\rangle \leq -\langle BH, H\rangle - \langle B(h - H), H\rangle \leq -\langle BH, H\rangle + \frac{\beta^2}{2}\|H\|^2 + \frac{1}{2\beta^2}\|B(H - h)\|^2.$$

By A4 again, this inequality yields for any $\beta_{t,k+1} > 0$ after applying the conditional expectation

$$\mathbb{E}\left[\mathrm{W}(\widehat{S}_{t,k+1})\Big|\mathcal{F}_{t,0}\right] \leq \mathbb{E}\left[\mathrm{W}(\widehat{S}_{t,k})\Big|\mathcal{F}_{t,0}\right] - \gamma_{t,k+1}v_{\min}\Lambda_{t,k+1}\mathbb{E}\left[\|H_{t,k+1}\|^2\Big|\mathcal{F}_{t,0}\right]$$
$$+ \frac{\gamma_{t,k+1}}{2\beta_{t,k+1}^2}v_{\max}^2\mathbb{E}\left[\|H_{t,k+1} - \mathsf{h}(\widehat{S}_{t,k})\|^2\Big|\mathcal{F}_{t,0}\right], \quad (46)$$

where

$$\Lambda_{t,k+1} := 1 - \gamma_{t,k+1}\frac{L_{\dot{\mathrm{W}}}}{2v_{\min}} - \frac{\beta_{t,k+1}^2}{2v_{\min}}.$$

By (46) and Proposition 24, it holds

$$\mathbb{E}\left[\mathrm{W}(\widehat{S}_{t,k+1})|\mathcal{F}_{t,0}\right] \leq \mathbb{E}\left[\mathrm{W}(\widehat{S}_{t,k})|\mathcal{F}_{t,0}\right] - \gamma_{t,k+1}v_{\min}\Lambda_{t,k+1}\mathbb{E}\left[\|H_{t,k+1}\|^2|\mathcal{F}_{t,0}\right]$$
$$+ \frac{\gamma_{t,k+1}}{2\beta_{t,k+1}^2}v_{\max}^2\frac{L^2}{n\mathsf{b}}\sum_{\ell=1}^{k}\gamma_{t,\ell}^2\mathbb{E}\left[\|H_{t,\ell}\|^2|\mathcal{F}_{t,0}\right]$$
$$+ \frac{\gamma_{t,k+1}}{2\beta_{t,k+1}^2}v_{\max}^2\frac{\omega}{n}\frac{1}{n}\sum_{i=1}^{n}\mathbb{E}\left[\|\Delta_{t,k+1,i}\|^2|\mathcal{F}_{t,0}\right]. \quad (47)$$

Set

$$G_{t,k} := \frac{1}{n}\sum_{i=1}^{n}\|V_{t,k,i} - \mathsf{h}_i(\widehat{S}_{t,k})\|^2.$$

From Proposition 19, we obtain

$$
\mathbb{E}\left[\mathrm{W}(\widehat{S}_{t,k+1})|\mathcal{F}_{t,0}\right] \le \mathbb{E}\left[\mathrm{W}(\widehat{S}_{t,k})|\mathcal{F}_{t,0}\right] - \gamma_{t,k+1}v_{\min}\Lambda_{t,k+1}\mathbb{E}\left[\|H_{t,k+1}\|^2|\mathcal{F}_{t,0}\right]
$$
$$
+ \frac{\gamma_{t,k+1}}{2\beta_{t,k+1}^2}v_{\max}^2\frac{L^2}{n\mathsf{b}}(1+2\omega)\sum_{\ell=1}^{k}\gamma_{t,\ell}^2\mathbb{E}\left[\|H_{t,\ell}\|^2|\mathcal{F}_{t,0}\right] + \frac{\gamma_{t,k+1}}{\beta_{t,k+1}^2}v_{\max}^2\frac{\omega}{n}\mathbb{E}\left[G_{t,k}|\mathcal{F}_{t,0}\right] . \quad (48)
$$

Assume that $k \mapsto \gamma_{t,k+1}/\beta_{t,k+1}^2$ is a non-increasing sequence and set

$$
C_{t,k+1} := \frac{2\omega}{\alpha n}v_{\max}^2\frac{\gamma_{t,k+1}}{\beta_{t,k+1}^2} . \quad (49)
$$

From Proposition 22, since $\alpha \in (0, 1/(1+\omega)]$, we have

$$
C_{t,k+1}\mathbb{E}\left[G_{t,k+1}|\mathcal{F}_{t,0}\right] \le (1-\alpha/2)C_{t,k+1}\mathbb{E}\left[G_{t,k}|\mathcal{F}_{t,0}\right] + \frac{2}{\alpha}L^2\gamma_{t,k+1}^2 C_{t,k+1}\mathbb{E}\left[\|H_{t,k+1}\|^2|\mathcal{F}_{t,0}\right]
$$
$$
+ 2\alpha\frac{L^2}{\mathsf{b}}C_{t,k+1}\sum_{\ell=1}^{k}\gamma_{t,\ell}^2\mathbb{E}\left[\|H_{t,\ell}\|^2|\mathcal{F}_{t,0}\right] . \quad (50)
$$

Upon noting that by definition of $C_{t,k+1}$ we have (remember that $C_{t,k+1} \le C_{t,k}$)

$$
(1-\alpha/2)C_{t,k+1} - C_{t,k} + \frac{\gamma_{t,k+1}}{\beta_{t,k+1}^2}v_{\max}^2\frac{\omega}{n} \le 0 ,
$$

this yields from (48) and (50)

$$
\mathbb{E}\left[\mathrm{W}(\widehat{S}_{t,k+1})|\mathcal{F}_{t,0}\right] + C_{t,k+1}\mathbb{E}\left[G_{t,k+1}|\mathcal{F}_{t,0}\right] \le \mathbb{E}\left[\mathrm{W}(\widehat{S}_{t,k})|\mathcal{F}_{t,0}\right] + C_{t,k}\mathbb{E}\left[G_{t,k}|\mathcal{F}_{t,0}\right]
$$
$$
- \left(\gamma_{t,k+1}v_{\min}\Lambda_{t,k+1} - \frac{2}{\alpha}L^2\gamma_{t,k+1}^2 C_{t,k+1}\right)\mathbb{E}\left[\|H_{t,k+1}\|^2|\mathcal{F}_{t,0}\right]
$$
$$
+ \left(\frac{\gamma_{t,k+1}}{2\beta_{t,k+1}^2}v_{\max}^2\frac{L^2}{n\mathsf{b}}(1+2\omega) + 2\alpha\frac{L^2}{\mathsf{b}}C_{t,k+1}\right)\sum_{\ell=1}^{k}\gamma_{t,\ell}^2\mathbb{E}\left[\|H_{t,\ell}\|^2|\mathcal{F}_{t,0}\right] .
$$

Let us restrict the computations to the case $\gamma_{t,k} = \gamma$, $\beta_{t,k} = \beta$ (which implies $C_{t,k+1} = C_{t,k} =: C$); we obtain

$$
\gamma v_{\min}\left(1 - \gamma\frac{L_{\dot{\mathrm{W}}}}{2v_{\min}} - \frac{\beta^2}{2v_{\min}} - \frac{\gamma^2}{\beta^2}\frac{4v_{\max}^2}{v_{\min}}L^2\frac{\omega}{\alpha^2 n}\right)\mathbb{E}\left[\|H_{t,k+1}\|^2|\mathcal{F}_{t,0}\right]
$$
$$
\le \mathbb{E}\left[\mathrm{W}(\widehat{S}_{t,k})|\mathcal{F}_{t,0}\right] + C\mathbb{E}\left[G_{t,k}|\mathcal{F}_{t,0}\right] - \mathbb{E}\left[\mathrm{W}(\widehat{S}_{t,k+1})|\mathcal{F}_{t,0}\right] - C\mathbb{E}\left[G_{t,k+1}|\mathcal{F}_{t,0}\right]
$$
$$
+ \frac{\gamma^3}{2\beta^2}v_{\max}^2\frac{L^2}{n\mathsf{b}}(1+10\omega)\sum_{\ell=1}^{k}\mathbb{E}\left[\|H_{t,\ell}\|^2|\mathcal{F}_{t,0}\right] .
$$

We now sum from $k = 0$ to $k = k_{\mathrm{in}} - 1$ and divide by $k_{\mathrm{in}}$:

$$
\gamma v_{\min}\left(1 - \gamma\frac{L_{\dot{\mathrm{W}}}}{2v_{\min}} - \frac{\beta^2}{2v_{\min}} - \frac{\gamma^2}{\beta^2}\frac{4v_{\max}^2}{v_{\min}}L^2\frac{\omega}{\alpha^2 n}\right)\frac{1}{k_{\mathrm{in}}}\sum_{k=1}^{k_{\mathrm{in}}}\mathbb{E}\left[\|H_{t,k}\|^2|\mathcal{F}_{t,0}\right]
$$
$$
\le k_{\mathrm{in}}^{-1}\mathbb{E}\left[\mathrm{W}(\widehat{S}_{t,0})|\mathcal{F}_{t,0}\right] + \frac{C}{k_{\mathrm{in}}}\mathbb{E}\left[G_{t,0}|\mathcal{F}_{t,0}\right]
$$
$$
- k_{\mathrm{in}}^{-1}\mathbb{E}\left[\mathrm{W}(\widehat{S}_{t,k_{\mathrm{in}}})|\mathcal{F}_{t,0}\right] - \frac{C}{k_{\mathrm{in}}}\mathbb{E}\left[G_{t,k_{\mathrm{in}}}|\mathcal{F}_{t,0}\right]
$$
$$
+ \frac{\gamma^3}{2\beta^2}v_{\max}^2\frac{L^2}{n\mathsf{b}}(1+10\omega)\sum_{k=1}^{k_{\mathrm{in}}}\mathbb{E}\left[\|H_{t,k}\|^2|\mathcal{F}_{t,0}\right] .
$$

As a conclusion, we have

$$\gamma v_{\min} \left( 1 - \gamma \frac{L_{\dot{\mathsf{W}}}}{2v_{\min}} - \gamma \bar{\Lambda} \right) \frac{1}{k_{\mathrm{in}}} \sum_{k=0}^{k_{\mathrm{in}}-1} \mathbb{E} \left[ \|H_{t,k+1}\|^2 | \mathcal{F}_{t,0} \right]$$

$$\leq k_{\mathrm{in}}^{-1} \mathbb{E} \left[ \mathrm{W}(\widehat{S}_{t,0}) | \mathcal{F}_{t,0} \right] + \frac{C}{k_{\mathrm{in}}} \mathbb{E} \left[ G_{t,0} | \mathcal{F}_{t,0} \right]$$

$$- k_{\mathrm{in}}^{-1} \mathbb{E} \left[ \mathrm{W}(\widehat{S}_{t,k_{\mathrm{in}}}) | \mathcal{F}_{t,0} \right] - \frac{C}{k_{\mathrm{in}}} \mathbb{E} \left[ G_{t,k_{\mathrm{in}}} | \mathcal{F}_{t,0} \right] .$$

where

$$\bar{\Lambda} := \frac{\beta^2}{2v_{\min}\gamma} + \frac{\gamma}{\beta^2} \frac{4v_{\max}^2}{v_{\min}} L^2 \frac{\omega}{\alpha^2 n} + \frac{\gamma}{2\beta^2} \frac{v_{\max}^2}{v_{\min}} \frac{L^2 k_{\mathrm{in}}}{n\mathsf{b}} (1 + 10\omega) .$$

Next, we sum from $t = 1$ to $t = k_{\mathrm{out}}$, divide by $k_{\mathrm{out}}$.

$$\gamma v_{\min} \left( 1 - \gamma \frac{L_{\dot{\mathsf{W}}}}{2v_{\min}} - \gamma \bar{\Lambda} \right) \frac{1}{k_{\mathrm{out}} k_{\mathrm{in}}} \sum_{k=1}^{k_{\mathrm{out}}} \sum_{k=1}^{k_{\mathrm{in}}} \mathbb{E} \left[ \|H_{t,k+1}\|^2 \right]$$

$$\leq k_{\mathrm{in}}^{-1} k_{\mathrm{out}}^{-1} \left( \mathbb{E} \left[ \mathrm{W}(\widehat{S}_{1,0}) \right] - \min \mathrm{W} \right) + \frac{C}{k_{\mathrm{in}} k_{\mathrm{out}}} \mathbb{E} \left[ G_{1,0} \right] . \quad (51)$$

Finally, we apply the expectation, with $(\tau, K)$ a uniform random variable on $[k_{\mathrm{out}}]^\star \times [k_{\mathrm{in}} - 1]$, independent of $\{\widehat{S}_{t,k}, t \in [k_{\mathrm{out}}]^\star, k \in [k_{\mathrm{in}} - 1]\}$, upon noting that $G_{t,k_{\mathrm{in}}} = G_{t+1,0}$ and $\widehat{S}_{t,k_{\mathrm{in}}} = \widehat{S}_{t+1,0}$, this yields

$$\gamma v_{\min} \left( 1 - \gamma \frac{L_{\dot{\mathsf{W}}}}{2v_{\min}} - \gamma \bar{\Lambda} \right) \mathbb{E} \left[ \|H_{\tau,K+1}\|^2 \right]$$

$$\leq k_{\mathrm{in}}^{-1} k_{\mathrm{out}}^{-1} \left( \mathbb{E} \left[ \mathrm{W}(\widehat{S}_{1,0}) \right] - \min \mathrm{W} \right) + \frac{C}{k_{\mathrm{in}} k_{\mathrm{out}}} \mathbb{E} \left[ G_{1,0} \right] . \quad (52)$$

**Impact of initialization.** With $V_{1,0,i} = \mathsf{h}_i(\widehat{S}_{1,0})$ for any $i \in [n]^\star$, we have $G_{1,0} = 0$.

**Choice of $\beta$.** The latter inequality is true for all parameter $\beta^2 > 0$ (coming from Young's inequality). We can thus optimize the value of $\beta^2$ to minimize the value of $\bar{\Lambda}$. We here discuss this choice. First, to ensure that $\bar{\Lambda}$ is independent of $\gamma$, we introduce $\mathsf{a}$, and set $\beta^2 = \mathsf{a}\gamma$ so that

$$\bar{\Lambda} = \frac{\mathsf{a}}{2v_{\min}} + \frac{1}{\mathsf{a}} \frac{4v_{\max}^2}{v_{\min}} L^2 \frac{\omega}{\alpha^2 n} + \frac{1}{2\mathsf{a}} \frac{v_{\max}^2}{v_{\min}} \frac{L^2 k_{\mathrm{in}}}{n\mathsf{b}} (1 + 10\omega)$$

$$= \frac{\mathsf{a}}{2v_{\min}} + \frac{4}{\mathsf{a}} \frac{v_{\max}^2}{v_{\min}} \frac{L^2}{n\alpha^2} \left( \omega + \frac{k_{\mathrm{in}}\alpha^2}{8\mathsf{b}} (1 + 10\omega) \right) .$$

Next, we optimize the value of $\mathsf{a}$.[2] Upon noting that $\mathsf{a} \mapsto A\mathsf{a} + B/\mathsf{a}$ (for $A, B > 0$) is lower bounded by $2\sqrt{AB}$ and its minimizer is $\mathsf{a}_\star := \sqrt{B/A}$, we choose

$$\mathsf{a}_\star := 2\sqrt{2}v_{\max} \frac{L}{\sqrt{n}\alpha} \left( \omega + \frac{k_{\mathrm{in}}\alpha^2}{8\mathsf{b}} (1 + 10\omega) \right)^{1/2} .$$

and obtain

$$\bar{\Lambda} = 2\sqrt{2} \frac{v_{\max}}{v_{\min}} \frac{L}{\sqrt{n}\alpha} \left( \omega + \frac{k_{\mathrm{in}}\alpha^2}{8\mathsf{b}} (1 + 10\omega) \right)^{1/2} . \quad (53)$$

Combining Equation (53) and Equation (52), we obtain

$$v_{\min} (1 - \gamma\Lambda_\star) \mathbb{E} \left[ \|H_{\tau,K+1}\|^2 \right] \leq \gamma^{-1} k_{\mathrm{in}}^{-1} k_{\mathrm{out}}^{-1} \left( \mathbb{E} \left[ \mathrm{W}(\widehat{S}_{1,0}) \right] - \min \mathrm{W} \right) ,$$

where

$$\Lambda_\star := \frac{L_{\dot{\mathsf{W}}}}{2v_{\min}} + 2\sqrt{2} \frac{v_{\max}}{v_{\min}} \frac{L}{\sqrt{n}\alpha} \left( \omega + \frac{k_{\mathrm{in}}\alpha^2}{8\mathsf{b}} (1 + 10\omega) \right)^{1/2} , \quad (54)$$

which is the result of Proposition 25.

---

[2]Remark that this optimization step is crucial to optimize the dependency of $\bar{\Lambda}$ w.r.t. $\omega$: this ensures that $\bar{\Lambda} \propto \omega^{3/2}$.

### E.5 Proof of Theorem 3 (Equation (11)) from Proposition 25

We apply Proposition 25 with: $b := \lceil \frac{k_{in}}{(1+\omega)^2} \rceil$ and the largest possible learning rate $\alpha = (1+\omega)^{-1}$: this gives in Equation (54)

$$
\begin{aligned}
\Lambda_\star &= \frac{L_{\dot{W}}}{2v_{\min}} + 2\sqrt{2}\frac{v_{\max}}{v_{\min}}\frac{L}{\sqrt{n}}(1+\omega)\left(\omega + \frac{1+10\omega}{8}\right)^{1/2} \\
&= \frac{L_{\dot{W}}}{2v_{\min}}\left(1 + 4\sqrt{2}\frac{v_{\max}}{L_{\dot{W}}}\frac{L}{\sqrt{n}}(1+\omega)\left(\omega + \frac{1+10\omega}{8}\right)^{1/2}\right) .
\end{aligned}
$$

Next, we choose $\gamma$ to be the largest possible value to ensure $(1 - \gamma\Lambda_\star) \geq \frac{1}{2}$. For all $t, k$,

$$
\gamma_{t,k} = \gamma := \frac{1}{2\Lambda_\star} = \frac{v_{\min}}{L_{\dot{W}}}\left(1 + 4\sqrt{2}\frac{v_{\max}}{L_{\dot{W}}}\frac{L}{\sqrt{n}}(1+\omega)\left(\omega + \frac{1+10\omega}{8}\right)^{1/2}\right)^{-1} .
$$

This gives the first result of Theorem 3, namely Equation (11). We give the proof of the second result, Equation (12) in the following subsection.

### E.6 Proof of Theorem 3 (Equation (12)): control on $h(\widehat{S}_{\tau,K})$

We now establish (12) for $\gamma_{t,k} = \gamma$. Let $t \in [k_{out}]^\star$ and $k \in [k_{in} - 1]$. We have

$$
\|h(\widehat{S}_{t,k})\|^2 \leq 2\|\mathbb{E}[H_{t,k+1}|\mathcal{F}_{t,k}]\|^2 + 2\|h(\widehat{S}_{t,k}) - \mathbb{E}[H_{t,k+1}|\mathcal{F}_{t,k}]\|^2 . \tag{55}
$$

Let us consider the first term in (55). By Jensen's inequality and the tower property of conditional expectations

$$
\mathbb{E}\left[\|\mathbb{E}[H_{t,k+1}|\mathcal{F}_{t,k}]\|^2|\mathcal{F}_{t,0}\right] \leq \mathbb{E}\left[\mathbb{E}\left[\|H_{t,k+1}\|^2|\mathcal{F}_{t,k}\right]|\mathcal{F}_{t,0}\right] = \mathbb{E}\left[\|H_{t,k+1}\|^2|\mathcal{F}_{t,0}\right] .
$$

Let us now consider the second term in (55). By Proposition 23 and Proposition 24, we have

$$
\mathbb{E}\left[\|\mathbb{E}[H_{t,k+1}|\mathcal{F}_{t,k}] - h(\widehat{S}_{t,k})\|^2|\mathcal{F}_{t,0}\right] \leq \left\{\begin{array}{ll} \gamma^2\frac{L^2}{nb}\sum_{\ell=1}^{k-1}\mathbb{E}\left[\|H_{t,\ell}\|^2|\mathcal{F}_{t,0}\right] & \text{when } k \geq 2 \\ 0 & \text{when } k \in \{0,1\} \end{array}\right. .
$$

Therefore, we write

$$
\mathbb{E}\left[\|h(\widehat{S}_{t,k})\|^2\right] \leq 2\mathbb{E}\left[\|H_{t,k+1}\|^2\right] + 2\gamma^2\frac{L^2}{nb}\sum_{\ell=1}^{k-1}\mathbb{E}\left[\|H_{t,\ell}\|^2\right]
$$

We now sum from $k = 0$ to $k = k_{in} - 1$, then from $t = 1$ to $t = k_{out}$, and finally we divide by $k_{in}k_{out}$. This yields

$$
\begin{aligned}
\mathbb{E}\left[\|h(\widehat{S}_{\tau,K})\|^2\right] &\leq 2\mathbb{E}\left[\|H_{\tau,K+1}\|^2\right] + 2\gamma^2\frac{L^2}{nb}\frac{1}{k_{in}k_{out}}\sum_{t=1}^{k_{out}}\sum_{k=2}^{k_{in}-1}\sum_{\ell=1}^{k-1}\mathbb{E}\left[\|H_{t,\ell}\|^2\right] \\
&\leq 2\mathbb{E}\left[\|H_{\tau,K+1}\|^2\right] + 2\gamma^2\frac{L^2}{nb}\frac{1}{k_{out}}\sum_{t=1}^{k_{out}}\sum_{k=1}^{k_{in}-2}\mathbb{E}\left[\|H_{t,k}\|^2\right] \\
&\leq 2\mathbb{E}\left[\|H_{\tau,K+1}\|^2\right] + 2\gamma^2\frac{L^2}{n}\frac{k_{in}}{b}\mathbb{E}\left[\|H_{\tau,K+1}\|^2\right] \\
&\leq 2\left(1 + \gamma^2\frac{L^2}{n}\frac{k_{in}}{b}\right)\mathbb{E}\left[\|H_{\tau,K+1}\|^2\right] .
\end{aligned}
$$

### E.7 On the convergence of the $V_{t,k,i}$'s

In this subsection, we provide a complementary result, to support the assertion made in the text, that the variable $V_{t,k,i}$ approximates $h_i(\widehat{S}_{t,k})$. Recall that for $t \in [k_{out}]^\star$ and $k \in [k_{in}]$, $G_{t,k} := \frac{1}{n}\sum_{i=1}^{n}\|V_{t,k,i} - h_i(\widehat{S}_{t,k})\|^2$ .

**Proposition 26.** *When running [algorithm 2](#) with a constant step size $\gamma$ equal to*

$$\gamma := \frac{v_{\min}}{L_{\dot{W}}} \left( 1 + 4\sqrt{2} \frac{v_{\max}}{L_{\dot{W}}} \frac{L}{\sqrt{n}} (1+\omega) \left( \omega + \frac{1+10\omega}{8} \right)^{1/2} \right)^{-1} ,$$

*with* $\mathsf{b} := \lceil \frac{k_{\mathrm{in}}}{(1+\omega)^2} \rceil$ *and* $\alpha := 1/(\omega+1)$, *we have*

$$\frac{1}{k_{\mathrm{out}} k_{\mathrm{in}}} \sum_{t=1}^{k_{\mathrm{out}}} \sum_{k=1}^{k_{\mathrm{in}}} \mathbb{E}[G_{t,k}] \leq \frac{2(1+\omega)}{k_{\mathrm{in}} k_{\mathrm{out}}} \mathbb{E}[G_{1,0}] + 16 \frac{\gamma}{k_{\mathrm{in}} k_{\mathrm{out}}} \frac{(1+\omega)^2 L^2}{v_{\min}} \left( \mathbb{E}\left[ \mathrm{W}(\widehat{S}_{1,0}) \right] - \min \mathrm{W} \right) .$$

In words, the Cesaro average $\frac{1}{k_{\mathrm{out}} k_{\mathrm{in}}} \sum_{t=1}^{k_{\mathrm{out}}} \sum_{k=1}^{k_{\mathrm{in}}} \mathbb{E}[G_{t,k}]$ decreases proportionally to the number of iterations $k_{\mathrm{in}} k_{\mathrm{out}}$. Consequently, the average squared distance between $V_{t,k,i}$ and $\mathsf{h}_i(\widehat{S}_{t,k})$ (i.e., $G_{t,k}$), converges to 0 in the sense of Cesaro.

*Proof.* From Proposition [22](#), we have that, $t \in [k_{\mathrm{out}}]^\star$ and $k \in [k_{\mathrm{in}}]$, and any $\alpha \leq (\omega+1)^{-1}$:

$$\mathbb{E}\left[ G_{t,k+1} | \mathcal{F}_{t,0} \right] \leq (1 - \alpha/2) \mathbb{E}\left[ G_{t,k} | \mathcal{F}_{t,0} \right]$$
$$+ \frac{2}{\alpha} L^2 \gamma_{t,k+1}^2 \mathbb{E}\left[ \|H_{t,k+1}\|^2 | \mathcal{F}_{t,0} \right] + 2\alpha \frac{L^2}{\mathsf{b}} \sum_{\ell=1}^{k} \gamma_{t,\ell}^2 \mathbb{E}\left[ \|H_{t,\ell}\|^2 | \mathcal{F}_{t,0} \right] .$$

Equivalently:

$$\alpha/2 \mathbb{E}\left[ G_{t,k} | \mathcal{F}_{t,0} \right] \leq \mathbb{E}\left[ G_{t,k} | \mathcal{F}_{t,0} \right] - \mathbb{E}\left[ G_{t,k+1} | \mathcal{F}_{t,0} \right]$$
$$+ \frac{2}{\alpha} L^2 \gamma_{t,k+1}^2 \mathbb{E}\left[ \|H_{t,k+1}\|^2 | \mathcal{F}_{t,0} \right] + 2\alpha \frac{L^2}{\mathsf{b}} \sum_{\ell=1}^{k} \gamma_{t,\ell}^2 \mathbb{E}\left[ \|H_{t,\ell}\|^2 | \mathcal{F}_{t,0} \right] .$$

Summing from $k = 0$ to $k = k_{\mathrm{in}} - 1$, we get, with $\gamma_{t,k+1}^2 = \gamma$:

$$\frac{\alpha}{2} \sum_{k=0}^{k_{\mathrm{in}}-1} \mathbb{E}\left[ G_{t,k} | \mathcal{F}_{t,0} \right] \leq \mathbb{E}\left[ G_{t,0} | \mathcal{F}_{t,0} \right] - \mathbb{E}\left[ G_{t,k_{\mathrm{in}}} | \mathcal{F}_{t,0} \right]$$
$$+ \frac{2}{\alpha} L^2 \gamma^2 \sum_{k=1}^{k_{\mathrm{in}}} \mathbb{E}\left[ \|H_{t,k}\|^2 | \mathcal{F}_{t,0} \right] + 2\alpha \frac{L^2}{\mathsf{b}} \sum_{k=1}^{k_{\mathrm{in}}-1} \sum_{\ell=1}^{k} \gamma^2 \mathbb{E}\left[ \|H_{t,\ell}\|^2 | \mathcal{F}_{t,0} \right]$$
$$\leq \mathbb{E}\left[ G_{t,0} | \mathcal{F}_{t,0} \right] - \mathbb{E}\left[ G_{t,k_{\mathrm{in}}} | \mathcal{F}_{t,0} \right]$$
$$+ \frac{2}{\alpha} L^2 \gamma^2 \sum_{k=1}^{k_{\mathrm{in}}} \mathbb{E}\left[ \|H_{t,k}\|^2 | \mathcal{F}_{t,0} \right] + 2\alpha \frac{L^2 k_{\mathrm{in}}}{\mathsf{b}} \sum_{k=1}^{k_{\mathrm{in}}} \gamma^2 \mathbb{E}\left[ \|H_{t,k}\|^2 | \mathcal{F}_{t,0} \right]$$
$$\leq \mathbb{E}\left[ G_{t,0} | \mathcal{F}_{t,0} \right] - \mathbb{E}\left[ G_{t,k_{\mathrm{in}}} | \mathcal{F}_{t,0} \right]$$
$$+ \frac{2}{\alpha} L^2 \gamma^2 \left( 1 + \frac{\alpha^2 k_{\mathrm{in}}}{\mathsf{b}} \right) \sum_{k=1}^{k_{\mathrm{in}}} \mathbb{E}\left[ \|H_{t,k}\|^2 | \mathcal{F}_{t,0} \right] .$$

Summing from $t = 1$ to $t = k_{\mathrm{out}}$, dividing by $k_{\mathrm{out}} k_{\mathrm{in}}$, and taking expectation we get:

$$\frac{1}{k_{\mathrm{out}} k_{\mathrm{in}}} \sum_{t=1}^{k_{\mathrm{out}}} \sum_{k=0}^{k_{\mathrm{in}}-1} \mathbb{E}[G_{t,k}] \leq \frac{2}{\alpha k_{\mathrm{out}} k_{\mathrm{in}}} \mathbb{E}[G_{1,0}]$$
$$+ \frac{4}{\alpha^2 k_{\mathrm{out}} k_{\mathrm{in}}} L^2 \gamma^2 \left( 1 + \frac{\alpha^2 k_{\mathrm{in}}}{\mathsf{b}} \right) \sum_{t=1}^{k_{\mathrm{out}}} \sum_{k=1}^{k_{\mathrm{in}}} \mathbb{E}[\|H_{t,k}\|^2] .$$

We used that $G_{t,k_{\mathrm{in}}} = G_{t+1,0}$. By denoting $(\tau, K)$ a uniform random variable on $[k_{\mathrm{out}}]^\star \times [k_{\mathrm{in}} - 1]$ – independent of the path $\{\widehat{S}_{t,k}, t \in [k_{\mathrm{out}}]^\star, k \in [k_{\mathrm{in}}]\}$, we have

$$\mathbb{E}[G_{\tau,K}] \leq \frac{2}{\alpha k_{\mathrm{out}} k_{\mathrm{in}}} \mathbb{E}[G_{1,0}] + \frac{4}{\alpha^2} L^2 \gamma^2 \left( 1 + \frac{\alpha^2 k_{\mathrm{in}}}{\mathsf{b}} \right) \mathbb{E}[\|H_{\tau,K+1}\|^2] .$$

From Theorem 3, this yields (note that $\alpha = (1+\omega)^{-1}$ and $b \geq k_{\text{in}}/(1+\omega)^2$)

$$\mathbb{E}[G_{\tau,K}] \leq \frac{2(1+\omega)}{k_{\text{out}} k_{\text{in}}} \mathbb{E}[G_{1,0}] + \gamma \frac{16(1+\omega)^2 L^2}{v_{\min} k_{\text{in}} k_{\text{out}}} \left( \mathrm{W}(\widehat{S}_{1,0}) - \min \mathrm{W} \right) \ .$$

□

# F  Supplement to the numerical section

This section gathers additional details concerning the models used in our numerical experiments. Namely, Appendix F.1 presents the full derivations for the `FedEM` algorithm for finite Gaussian Mixture Models, and Appendix F.2 provides the detailed pseudo-code for the `FedMissEM` algorithm for federated missing values imputation introduced in Section 4 and provides the necessary information to request access to the data we used on the eBird platform [1].

## F.1  Gaussian Mixture Model

Let $y_1, \ldots, y_N$ be $N$ $\mathbb{R}^p$-valued observations; they are modeled as the realization of a vector $(Y_1, \ldots, Y_N)$ with distribution defined as follows:

- conditionally to a $\{1, \ldots, L\}$-valued vector of random variables $(Z_1, \ldots, Z_N)$, $(Y_1, \ldots, Y_N)$ are independent; and the conditional distribution of $Y_i$ is $\mathcal{N}_p(\mu_{Z_i}, \Sigma)$.

- the r.v. $(Z_1, \ldots, Z_n)$ are i.i.d. with multinomial distribution of size $1$ and with probabilities $\pi_1, \ldots, \pi_L$.

Equivalently, the random variables $(Y_1, \ldots, Y_N)$ are independent with distribution $\sum_{\ell=1}^{L} \pi_\ell \mathcal{N}_p(\mu_\ell, \Sigma)$. For $1 \leq i \leq N$, the negative log-likelihood of the observation $Y_i$ is given up to an additive constant term by

$$\theta \mapsto \frac{1}{2} \ln \det \Sigma + \frac{1}{2} \langle Y_i Y_i^\top, \Sigma^{-1} \rangle - \ln \sum_{z=1}^{L} \exp \left( \langle s(Y_i, z), \phi(\theta) \rangle \right)$$

where, denoting $\mathbb{1}_{\{l\}}(z)$ the indicator function equal to $1$ if $z = l$ and $0$ otherwise:

$$s(y,z) := \begin{pmatrix} \mathbb{1}_{\{1\}}(z) \\ \vdots \\ \mathbb{1}_{\{L\}}(z) \\ y\mathbb{1}_{\{1\}}(z) \\ \vdots \\ y\mathbb{1}_{\{L\}}(z) \end{pmatrix}, \qquad \phi(\theta) := \begin{pmatrix} \log(\pi_1) - \frac{1}{2}\mu_1^\top \Sigma^{-1} \mu_1 \\ \vdots \\ \log(\pi_L) - \frac{1}{2}\mu_L^\top \Sigma^{-1} \mu_L \\ \Sigma^{-1}\mu_1 \\ \vdots \\ \Sigma^{-1}\mu_L \end{pmatrix}. \qquad (56)$$

The goal is to estimate the parameter $\theta := (\pi_1, \ldots, \pi_L, \mu_1, \ldots, \mu_L, \Sigma)$ by minimizing the normalized negative log-likelihood:

$$F(\theta) := \frac{1}{2} \ln \det \Sigma + \frac{1}{2} \left\langle \frac{1}{N} \sum_{i=1}^{N} Y_i Y_i^\top, \Sigma^{-1} \right\rangle - \frac{1}{N} \sum_{i=1}^{N} \ln \int \exp \left( \langle s(Y_i, z), \phi(\theta) \rangle \right) \nu(\mathrm{d}z) \quad (57)$$

where $\nu$ is the counting measure on $\{1, \ldots L\}$.

**Classical EM algorithm**  We use the EM algorithm: in the Expectation (E) step, using the current value of the iterate $\theta_{\text{curr}}$, we compute a majorizing function $\theta \mapsto Q(\theta, \theta_{\text{curr}})$ given up to an additive constant by

$$Q(\theta, \theta_{\text{curr}}) = -\langle \bar{s}(\theta_{\text{curr}}), \phi(\theta) \rangle + \psi(\theta),$$

where

$$\psi(\theta) := \frac{1}{2} \ln \det \Sigma + \frac{1}{2} \left\langle \frac{1}{N} \sum_{i=1}^{N} Y_i Y_i^\top, \Sigma^{-1} \right\rangle,$$

$\bar{s}(\theta_{\text{curr}}) := \frac{1}{N} \sum_{i=1}^N \bar{s}_i(\theta)$, and for any $i \in [N]^\star$, $\bar{s}_i(\theta)$ is the conditional expectation of the complete data sufficient statistics:

$$
\bar{s}_i(\theta) = \begin{pmatrix} \bar{\rho}_{i,1}(\theta) \\ \vdots \\ \bar{\rho}_{i,L}(\theta) \\ Y_i \bar{\rho}_{i,1}(\theta) \\ \vdots \\ Y_i \bar{\rho}_{i,L}(\theta) \end{pmatrix}, \text{ where for } \ell \in [L]^\star, \bar{\rho}_{i,l}(\theta) := \frac{\pi_\ell \, \mathcal{N}_p(\mu_\ell, \Sigma)[Y_i]}{\sum_{u=1}^L \pi_u \, \mathcal{N}_p(\mu_u, \Sigma)[Y_i]} . \tag{58}
$$

In (58), $\mathcal{N}_p(\mu, \Sigma)[y]$ is the density function of the distribution $\mathcal{N}_p(\mu, \Sigma)$ evaluated at $y$.

In the optimization step (M-step), a new value of $\theta_{\text{curr}}$ is computed as a minimizer of $\theta \mapsto Q(\theta, \theta_{\text{curr}})$. Let us now detail this step.

---

**Algorithm 5:** Classical EM algorithm for mixture of Gaussians

---

1: **Input:** $k_{\max} \in \mathbb{N}, X, \hat{S}_0, \hat{\theta}_0$
2: **Output:** The sequence of statistics: $\{\hat{S}_k, k \in [k_{\max}]\}$; the sequence of parameters $\{\hat{\theta}_k, k \in [k_{\max}]\}$
3: **for** $k = 0, \ldots, k_{\max} - 1$ **do**
4:   *Expectation step*: compute conditional expectations given current parameter $\hat{\theta}_k$: Set $\hat{S}_{k+1} = \frac{1}{N} \sum_{i=1}^N \bar{s}_i(\hat{\theta}^k)$
5:   *Maximization step*: update parameter $\hat{\theta}_{k+1}$ based on current statistics $\hat{S}_{k+1}$ according to update rule (60)
6: **end for**

---

**The M step: the map T.** Let

$$
s = (s^{(1)}, s^{(2)}) = (s^{(1),1}, \ldots, s^{(1),L}, s^{(2),1}, \ldots, s^{(2),L}) \in \mathbb{R}^L \times \mathbb{R}^{pL} ;
$$

we write $\langle s, \phi(\theta) \rangle = \sum_{j=1}^2 \langle s^{(j)}, \phi^{(j)}(\theta) \rangle$ where the functions $\phi^{(j)}$ are defined by

$$
\phi^{(1)}(\theta) := \begin{pmatrix} \log(\pi_1) - \frac{1}{2} \mu_1^\top \Sigma^{-1} \mu_1 \\ \vdots \\ \log(\pi_L) - \frac{1}{2} \mu_L^\top \Sigma^{-1} \mu_L \end{pmatrix}, \qquad \phi^{(2)}(\theta) := \begin{pmatrix} \Sigma^{-1} \mu_1 \\ \vdots \\ \Sigma^{-1} \mu_L \end{pmatrix} . \tag{59}
$$

By definition, $T(s) = \operatorname{argmin}_{\theta \in \Theta} - \langle s, \phi(\theta) \rangle + \psi(\theta)$. Here, this optimum is unique and defined by $T(s) = \{\pi_\ell(s), \mu_\ell(s), \ell = 1, \ldots, L; \Sigma\}$ with

$$
\pi_\ell(s) := \frac{s^{(1),\ell}}{\sum_{u=1}^L s^{(1),u}} , \tag{60}
$$

$$
\mu_\ell(s) := \frac{s^{(2),\ell}}{s^{(1),\ell}} , \tag{61}
$$

$$
\Sigma(s) := \frac{1}{N} \sum_{i=1}^N Y_i Y_i^\top - \sum_{\ell=1}^L s^{(1),\ell} \mu_\ell(s) \, \mu_\ell^\top(s) . \tag{62}
$$

The expressions of $\pi_\ell(s)$ and $\mu_\ell(s)$ are easily obtained. We provide details for the covariance matrix. We have for any symmetric matrix $H$

$$
\ln \frac{\det(\Gamma + H)}{\det(\Gamma)} = \ln \det(I + \Gamma^{-1} H) = \ln(1 + \operatorname{Tr}(\Gamma^{-1} H) + o(\|H\|))
$$

$$
= \operatorname{Tr}(\Gamma^{-1} H) + o(\|H\|) = \langle H, \Gamma^{-1} \rangle + o(\|H\|)
$$

thus showing that the derivative of $\Gamma \mapsto \ln \det \Gamma$ is $\Gamma^{-1}$. $T(s)$ depends on $\Sigma^{-1}$ through the function

$$
\Sigma^{-1} \mapsto -\frac{1}{2} \ln \det(\Sigma^{-1}) + \frac{1}{2} \left\langle \Sigma^{-1}, \frac{1}{N} \sum_{i=1}^N Y_i Y_i^\top \right\rangle + \left\langle \Sigma^{-1}, \frac{1}{2} \sum_{\ell=1}^L s^{(1),\ell} \mu_\ell \mu_\ell^\top - \sum_{\ell=1}^L \mu_\ell (s^{(2),\ell})^\top \right\rangle .
$$

The optimum solves

$$\Sigma = \frac{1}{N}\sum_{i=1}^{N} Y_i Y_i^\top + \sum_{\ell=1}^{L} s^{(1),\ell}\mu_\ell \mu_\ell^\top - 2\sum_{\ell=1}^{L}\mu_\ell \left(s^{(2),\ell}\right)^\top$$

Hence, $\Sigma(s)$ is this solution when $\mu_\ell \leftarrow \mu_\ell(s)$ which yields the expression since $s^{(2),\ell} = s^{(1),\ell}\mu_\ell(s)$.

**In the federated setting.** In the federated setting, the data is distributed across $n$ local servers. For all $c \in [n]^\star$, the $c$-th server possesses a local data set of size $N_c$; $N_c \geq 1$ and $\sum_{c=1}^{n} N_c = N$. We write

$$\bigcup_{i=1}^{N}\{Y_i\} = \bigcup_{c=1}^{n}\bigcup_{j=1}^{N_c}\{Y_{cj}\}\,,$$

thus meaning that each local worker $\#c$ processes the data set $\{Y_{c1},\ldots,Y_{cN_c}\}$.

The computation of the map $\mathsf{T}$ requires the knowledge of a statistic of the full data set, namely $N^{-1}\sum_{i=1}^{N} Y_i Y_i^\top$. For this reason, we want the map $\mathsf{T}$ to be available at the central server only. Since

$$\sum_{i=1}^{N} Y_i = \sum_{c=1}^{n}\sum_{j=1}^{N_c} Y_{cj}$$

this full sum can be computed during the initialization of the algorithm by the central server, by using the $n$ local summaries $\sum_{j=1}^{N_c} Y_{cj}$ sent by the local workers.

In the FL setting, we write the objective function as follows

$$\theta \mapsto \psi(\theta) - \frac{1}{N}\sum_{c=1}^{n}\sum_{j=1}^{N_c}\ln\int \exp\left(\langle s(Y_{cj},z),\phi(\theta)\rangle\right)\nu(\mathrm{d}z)$$

$$= -\frac{1}{N}\sum_{c=1}^{n}\ln\prod_{j=1}^{N_c}\int \exp\left(\langle s(Y_{cj},z),\phi(\theta)\rangle - \frac{N}{nN_c}\psi(\theta)\right)\nu(\mathrm{d}z)$$

$$\propto -\frac{1}{n}\sum_{c=1}^{n}\ln\prod_{j=1}^{N_c}\int \exp\left(\langle s(Y_{cj},z),\phi(\theta)\rangle - \frac{N}{nN_c}\psi(\theta)\right)\nu(\mathrm{d}z)\,.$$

It is of the form (1) with $\mathsf{R}(\theta) = 0$ and

$$\mathcal{L}_c(\theta) := -\ln\prod_{j=1}^{N_c}\int \exp\left(\langle s(Y_{cj},z),\phi(\theta)\rangle - \frac{N}{nN_c}\psi(\theta)\right)\nu(\mathrm{d}z)\,.$$

In the case $nN_c = N$ for any $c \in [n]^\star$, we have

$$\mathcal{L}_c(\theta) = -\sum_{j=1}^{N/n}\ln p(Y_{cj};\theta)\,,$$

with

$$p(y;\theta) := \int p(y,z;\theta)\,\nu(\mathrm{d}z) \qquad p(y,z;\theta) := \exp\left(\langle s(y,z),\phi(\theta)\rangle - \psi(\theta)\right)\nu(\mathrm{d}z)\,.$$

$p(y,z;\theta)$ is of the form (2); this yields

$$\bar{s}_{cj}(\theta) := \sum_{z=1}^{L} s(Y_{cj},z)\bar{\rho}_{cj,z}(\theta)\,, \qquad \bar{s}_c(\theta) := \frac{n}{N}\sum_{j=1}^{N/n}\bar{s}_{cj}\,,$$

where $\bar{\rho}_{cj,z}(\theta)$ is defined by (58).

The pseudo code for the FedEM algorithm is given in Algorithm 6.

**Algorithm 6:** Federated EM algorithm for distributed GMM without compression

1: **Input:** $k_{\max} \in \mathbb{N}$; for $c \in [n]^\star$, $V_{0,c} \in \mathbb{R}^{L+pL}$; $\widehat{S}_0 \in \mathbb{R}^{L+pL}$; $\hat{\theta}_0 \in \mathbb{R}^L \times (\mathbb{R}^p)^L \times \mathbb{R}^{p \times p}$; a positive sequence $\{\gamma_{k+1}, k \in [k_{\max} - 1]\}$; $\alpha$
2: **Output:** The `FedEMsequence`: $\{\widehat{S}_k, k \in [k_{\max}]\}$
3: **for** $k = 0, \ldots, k_{\max} - 1$ **do**
4:    **for** $c = 1, \ldots, n$ **do**
5:      *(agent #i, locally)*
6:      Sample a batch $\mathcal{I}_{k,c} \subset [N_c]$
7:      Set $\mathsf{S}_{k+1,c} = \frac{1}{|\mathcal{I}_{k,c}|} \sum_{i \in \mathcal{I}_{k,c}} \bar{s}_i(\hat{\theta}_k)$, where $\bar{s}_i$ is defined in (58)
8:      Set $\Delta_{k+1,c} = \mathsf{S}_{k+1,c} - \widehat{S}_k - V_{k,c}$
9:      Update $V_{k+1,c} = V_{k,c} + \alpha \, \mathrm{Quant}(\Delta_{k+1,c})$
10:      Send $\mathrm{Quant}(\Delta_{k+1,c})$ to the controller
11:    **end for**
12:    *(the controller)*
13:    Compute $H_{k+1} = V_k + \frac{1}{n} \sum_{c=1}^n \mathrm{Quant}(\Delta_{k+1,c})$
14:    Set $\widehat{S}_{k+1} = \widehat{S}_k + \gamma_{k+1} H_{k+1}$
15:    Set $V_{k+1} = V_k + \alpha n^{-1} \sum_{c=1}^n \mathrm{Quant}(\Delta_{k+1,c})$.
16:    Send $\widehat{S}_{k+1}$ and $\hat{\theta}_{k+1} = \mathsf{T}(\widehat{S}_{k+1})$ to the agents, where $\mathsf{T}(\hat{S}_{k+1})$ is given by the update rule (60)
17: **end for**

## F.2   Federated missing values imputation

• *Model and the `FedMissEM` algorithm.* $I$ observers participate in the programme, there are $J$ ecological sites and $L$ time stamps. Each observer $\#i$ provides a $J \times L$ matrix $X^i$ and a subset of indices $\Omega^i \subseteq [J]^\star \times [L]^\star$. For $j \in [J]^\star$ and $\ell \in [L]^\star$, the variable $X^i_{j\ell}$ encodes the observation that would be collected by observer $\#i$ if the site $\#j$ were visited at time stamp $\#\ell$; since there are unvisited sites, we denote by $Y^i := \{X^i_{j\ell}, (j,\ell) \in \Omega^i\}$ the set of observed values and $Z^i := \{X^i_{j\ell}, (j,\ell) \notin \Omega^i\}$ the set of unobserved values. The statistical model is parameterized by a matrix $\theta \in \mathbb{R}^{J \times L}$, where $\theta_{j\ell}$ is a scalar parameter characterizing the distribution of species individuals at site $j$ and time stamp $\ell$. For instance, $\theta_{j\ell}$ is the log-intensity of a Poisson distribution when the observations are count data or the log-odd of a binomial model when the observations are presence-absence data. This model could be extended to the case observers $\#i$ and $\#i'$ count different number of specimens on average at the same location and time stamp, because they do not have access to the same material or do not have the same level of expertise: heterogeneity between observers could be modeled by using different parameters for each individual $\#i$ say $\theta^i \in \mathbb{R}^{J \times L}$. Here, we consider the case when $\theta^i_{j\ell} = \theta_{j\ell}$ for all $(j,\ell) \in [J]^\star \times [L]^\star$ and $i \in [I]^\star$.

We further assume that the entries $\{X^i_{j\ell}, i \in [I]^\star, j \in [J]^\star, \ell \in [L]^\star\}$ are independent with a distribution from an exponential family with respect to some reference measure $\nu$ on $\mathbb{R}$ of the form: $x \mapsto \rho(x) \exp\{x\theta_{j\ell} - \psi(\theta_{j\ell})\}$. The function $\psi$ is for instance defined by $\psi(\tau) = -\frac{1}{2}\tau^2$ for a Gaussian model with expectation $\tau$ and variance 1, $\psi(\tau) = \log(1 + \mathrm{e}^\tau)$ for a Bernoulli model with success probability $\tau$, and $\psi(\tau) = \mathrm{e}^\tau$ for a Poisson model with intensity $\tau$. Therefore, the joint distribution of $(Y^i, Z^i)$ is given by $p_i(y^i, z^i; \theta) := \left( \prod_{(j,\ell) \in \Omega^i} \rho(y^i_{j\ell}) \right) \left( \prod_{(j,\ell) \notin \Omega^i} \rho(z^i_{j\ell}) \right) \exp\left( \langle s_i(y^i, z^i), \theta \rangle - \sum_{j\ell} \psi(\theta_{j\ell}) \right)$; where $s_i(Y^i, Z^i)$ is a $J \times L$ matrix with entry $\#(j, \ell)$ given by $Y^i_{j\ell}$ if $(j,\ell) \in \Omega^i$ and $Z^i_{j,\ell}$ otherwise.

In order to estimate the unknown matrix $\theta \in \mathbb{R}^{J \times L}$, we assume that $\theta$ is low-rank; we use the parameterization $\theta = UV^\top$, where $U \in \mathbb{R}^{J \times r}$ and $V \in \mathbb{R}^{L \times r}$ with $\mathrm{rank}(\theta) = r$ and $r < \min(J, L)$. The estimator is defined as a minimizer of the negative penalized log-likelihood: $\min_{U \in \mathbb{R}^{J \times r}, V \in \mathbb{R}^{L \times r}} F(U, V)$, with $F(U, V) := \frac{1}{n} \sum_{i=1}^n \mathcal{L}^i(UV^\top) + \frac{\lambda}{2}\left( \|U\|_F^2 + \|V\|_F^2 \right)$, where for $\theta \in \mathbb{R}^{J \times L}$, $\mathcal{L}^i(\theta) := -\log \int p_i(Y^i, z^i; \theta) \prod_{(j,\ell) \notin \Omega^i} \nu(\mathrm{d}z^i_{j\ell})$.

FedMissEM **algorithm.** Algorithm 7 provides the pseudo-code for the Federated EM algorithm for mising values imputation.

---

**Algorithm 7:** Federated EM algorithm for distributed missing data imputation

---

1: **Input:** $k_{\max} \in \mathbb{N}$; for $c \in [n]^\star$, $V_0^c \in \mathbb{R}^{I \times J}$; $\widehat{S}_0 \in \mathbb{R}^{I \times J}$; a positive sequence
   $\{\gamma_{k+1}, k \in [k_{\max} - 1]\}$; $\alpha$; the quantization function Quant
2: **Output:** The FedEM sequence: $\{\widehat{S}_k, k \in [k_{\max}]\}$
3: **for** $k = 0, \ldots, k_{\max} - 1$ **do**
4:     **for** $c = 1, \ldots, n$ **do**
5:        *(agent #i, locally)*
6:        Initialize $\mathsf{S}_{k+1,c} = 0$ and $\Delta_{k+1,c} = 0$ everywhere.
7:        Sample a minibatch $(\mathcal{I}_k^c, \mathcal{J}_k^c) \subset [I]^\star \times [J]^\star$
8:        **for** $i \in \mathcal{I}_k^c$ **do**
9:          **for** $j \in \mathcal{J}_k^c$ **do**
10:            Set $(\mathsf{S}_{k+1}^c)_{i,j} = \mathbb{1}_{i,j \in \Omega^c} Y_{i,j}^c + (1 - \mathbb{1}_{i,j \in \Omega^c})(\hat{\theta}_k)_{i,j}$
11:            Set $(\Delta_{k+1}^c)_{i,j} = (\mathsf{S}_{k+1}^c)_{i,j} - \widehat{S}_{i,j} - (V_k^c)_{i,j}$
12:          **end for**
13:        **end for**
14:        Update $V_{k+1}^c = V_k^c + \alpha \, \text{Quant}(\Delta_{k+1,c})$
15:        Send $\text{Quant}(\Delta_{k+1}^c)$ to the controller
16:     **end for**
17:     *(the controller)*
18:     Compute $H_{k+1} = V_k + n^{-1} \sum_{c=1}^{n} \text{Quant}(\Delta_{k+1}^c)$
19:     Set $\widehat{S}_{k+1} = \widehat{S}_k + \gamma_{k+1} H_{k+1}$
20:     Set $V_{k+1} = V_k + \alpha n^{-1} \sum_{c=1}^{n} \text{Quant}(\Delta_{k+1}^c)$.
21:     Send $\widehat{S}_{k+1}$ and $\hat{\theta}_{k+1} = \mathsf{T}(\widehat{S}_{k+1})$ to the agents
22:     *(Note: thresholded SVD for Gaussian model or computed iteratively for a general exponential family model)*
23: **end for**

---

**eBird data information.** In our experiments, we used a sample of the eBird data set [1], provided upon request by the Cornell Lab of Ornithology. We are not allowed to disclose the data itself, but we provide here the details to reproduce our experiments on the same data set, after requesting acess on the eBird platform (https://ebird.org/data/request). We selected the counts recorded anywhere in France, between January 2000 and September 2020, for two different species: the Mallard and the Common Buzzard. These two species were analyzed independently (see Section 4); the corresponding code is also available as supplementary material.