# OpenReview forum: "Federated-EM with heterogeneity mitigation and variance reduction"
_NeurIPS.cc/2021/Conference — NeurIPS 2021 Poster_

### Official Review · Reviewer_WGsy · 2021-07-01

**Rating:** 5
**Confidence:** 5

**Summary:**

The paper proposes methods to implement Expectation-Maximization (EM) algorithm under Federated Learning (FL) setting. The methods proposed incorporate gradient compression and variance reduction techniques, and also allow for partial participation during training process. Moreover, the paper gives finite-time complexity analysis of proposed methods. Finally, paper demonstrates their methods by both simulated and real data experiments.

**Limitations And Societal Impact:**

The paper discussed about its potential negative societal impact. The paper could perhaps think more about whether EM under FL can protect privacy.

**Main Review:**

While the problem solved by the paper is important and interesting, the way it is solved in the paper seems straightforward. More specifically, the EM algorithm is well-studied, and both of the gradient compression and variance reduction techniques used to implement EM for FL setting are borrowed from previous literatures. The theoretical analysis, while it is appreciated, looks like adaptation from existing papers. All in all, I do not see much novelty in this paper.

The paper is technically correct. The paper gives finite-time complexity analysis of proposed methods. The paper demonstrates their methods by both simulated and real data experiments. The work is overall complete.

The submission is well-organized. The methods are clearly motivated and explained. The proof is clear and complete as well.

The problem studied by the paper will be interesting for many practitioners. However, the paper is more like a combination of existing works in to guide engineering, rather than a novel research. There is limited unique results in the work. Thus, overall, it is of limited significance. I think at least paper should make it more clear about what is new in this work.

**Time Spent Reviewing:**

4

---

> ### Author Response · Authors · 2021-08-10
> **Response to reviewer WGSy**
>
> We thank the reviewer for his time and comments, and for considering that the problem solved by the paper is important and interesting for many practitioners, and for underlining that the paper is technically correct, the submission  well organized with clear and complete proofs, and with clearly and well explained methods.
>
> Nevertheless, the reviewer rates the paper poorly. As far as we can understand through a short review which does not explicitly raise questions, she/he considers the novelty is not significant enough since she/he considers that it is more a combination of existing works. Since the reviewer does not provide references to these existing works, we assume in our answer that he/she means existing works on Gradient Descent (GD) in FL and variance reduction in non-FL EM.
>
>
> Our paper obviously takes its roots in the analysis of first order algorithms. It builds on recent progress in the domain of Stochastic-GD and EM, relying on references to which we carefully referred. Yet, we believe the combination of these two  aspects results in numerous challenges, and opens novel directions of research for many people working on EM. If this paper were simply an "combination of existing works in to guide engineering", the same remarks could apply to most variance reduction techniques developed over the last 5 years, and to many papers in the FL area.
>
> Simply put, *does the reviewer consider FedAvg [Ref 1] a "combination of existing works in to guide engineering"?*
>
> In other words, we do not think it is a good approach to disregard a paper because the proofs "look like" the ones of the literature it builds on.
> Despite the lack of questions, we decided to answer this review by clarifying the positioning of the paper and highlighting the points that differ from the SGD context. In the following, we discuss some of the fundamental differences between our approach and previous work. Then, we detail the originality of our paper (which was also provided in the introduction of the paper). Our answer contains a technical part since the referee is expert with the related works and checks the maths and/or other details carefully.
>
>
> We sincerely hope this will help her/him to understand our contributions and that the reviewer will consider revising her/his conclusion, yielding a better rating of our work.
>
>
>
> **EM is not among the gradient descent algorithms.**  It is well commented in the literature that EM is not a gradient descent algorithm (see e.g. the reference [16]). It is among the class of Majorize-Minimization algorithms with a surrogate function which is not (except in toy examples) a quadratic function.
>
> **What Gradient Descent (GD) would perform for solving Eq. (1-2).** Addressing the optimization problem (1-2) with a GD-based algorithm would transfer to each node $\# i$ a local objective function given by  $\theta \mapsto (\nabla \phi(\theta))’  \bar{s}_i(\theta) - \nabla \psi(\theta)$ -- assuming the functions are smooth enough. Three key observations:
> - [A] In a Gradient setting, the problem is formulated and solved in the parameter space $\theta$ by both the nodes and the central server.  In the EM setting that we consider, each local node works in the expectation space and the central server, thanks to the knowledge of the operator T, ensures the jumps from the expectation space to the (original) parameter space and transmit $T(\hat{S}_k)$ to all the nodes.
> - [B] the local objective functions of the gradient setting differ from the ones in our paper, which are $u \mapsto \bar{s}_i \circ T(u) - u$. Consequently, our algorithms and their analysis are fundamentally different from the Gradient case. For example, on the algorithmic side: in Algo 1, each node communicates a compressed approximation of $\bar{s}_i\circ T(\hat{S}_k) -  \hat{S}_k$ while in a Gradient method, each node communicates a compressed approximation of $\bar{s}_i(\theta_k)$
>
> -  [C] there is no map T in the gradient approach; see the next comment.
>
> **Unshared map T.**
> In the exponential family framework considered in the paper, the optimization of the surrogate function is summarized by the map T (see A3).
> In the FL-EM framework we propose in the paper, this map T is available at the central server (for example, for privacy constraints -- see the answer to  Point 6; referee at3L).  Consequently, the local node does not have access to its local objective function $u \mapsto \bar{s}_i \circ T(u)-u$ : even if each local node were iterating a lot between two communications (or in the extreme case, were not communicating with the central server), the node will be unable to solve its local optimization.  This makes the EM framework really different from the Gradient-Descent one.
>
> **Our Variance Reduced methods result in a *biased* local approximation transmitted to the central server.**  Even when the quantization step is unbiased, the quantity sent by a local node to the central server is not necessarily an unbiased approximation of the 'ideal' quantity i.e. the local objective function $\bar{s}_i \circ T(u) - u$ evaluated at the current point.
>
>
> Hence, the approximated mean field $H_{t,k+1}$ is **biased** (see Proposition 19) and this requires a specific technique of proof.  It cannot be a trivial adaptation of existing proof; let us compare to Theorem 4 in [Ref 2]: in the gadient setting, [Ref 2] provides a variance reduction technique which guarantees the unbiasedness of the stochastic approximation (of the gradient).
> On the technical point of view, in our case:
> - $(i)$ the martingale-increment  property of the noisy mean-field is lost.
> - $(ii)$ the drift terms in the Lyapunov property is no more the quantity of interest $E[ \| h(\hat{S}_{t,k}) \|^2 ]$  (compare our Eq (46), and  Eq (36) in [Ref 2]).
> - $(iii)$ the control of the quadratic remainder terms  in the Lyapunov inequality, involve more intricate computations (see e.g. the term $\sum_\ell \gamma_{t,\ell}^2 \|H_{t,\ell}\|^2$ in Proposition 16 compared to Lemma 9 in [Ref 2]. In our case, we *have to* control the cumulated sum of the squared bias of $H_{t,k}$).
>
> **From the single node case to FL.** In the case of a single server (n=1, i.e. no "FL') and no compression,  [Ref 3] provides a variance-reduced EM algorithm which is based on the same control variate as we use in algorithm 2 (see Line 8).
> Here again, the proof for the case 'n>1, with compression' is not a trivial adaptation of the proof of the case 'n=1, no compression'.
> - First, the compression step implies an additional noisy term in the update mechanism of the quantity of interest performed by the central server (see line 14 Algo 2).
> - Second, the FL context necessitates the introduction of a memory (see the variables $V_{t,k,i}$, line 10 Algo 2). This quantity does not exist in the no-FL EM context. It requires a specific control (see Proposition 18). Here again, the biased approximation $H_{t,k+1,i}$ makes this control far more intricate than in [Ref 2] since it involves again the cumulated squared bias of $H_{t,k}$.
>
> **Unique results.**
> Our paper provides unique results, explicitly listed in the introduction. For example:
> - The paper is the first to propose an algorithm tackling communication and heterogeneity constraints for EM.
> - The analysis is fundamentally different as we do not rely on gradients (and more generally on unbiased approximation of the mean field)
> - EM has by nature specificities (linked to the majoration minorization structure), that make it difficult to tackle local iterations.
> - We provide a VR technique, and provide detailed comments on the suitable batch size, the regimes in which compression is the most useful, the dependency on $\omega$, etc.
>
>
> ---
> - [Ref 1] Communication-Efficient Learning of Deep Networks from Decentralized Data, McMahan et al., Aistats, 2017.
> - [Ref 2] Stochastic distributed learning with gradient quantization and variance reduction, S. Horvath, et al., 2019.
> - [Ref 3] A stochastic path integral differential estimator expectation maximization algorithm, Fort et al. Neurips, 2020.
>  ---

---

> > ### Comment · Reviewer_WGsy · 2021-08-29
> > **Thanks for response**
> >
> > Thanks for authors' response. Now I can appreciate the papers' contributions better. I guess the main problem about wirtting is that there are too many claimed contributions, and it is easily to give the impression that the contributions are simple sum-up of exisiting papers. I would suggest discussing what would be naive approach to solve the problem and what difficulities will appear before introducing your methods. I think the current writing brings much noise.
> >
> > Besides, I still feel that the discussion about data heterogeneity is vague. Since the paper is estimating a single $\theta$ in (3) and (4), and assumes the same parameteric form for all clients, I think it is equivalent to assume that the clients' data are i.i.d.. The authors should more carefully state their heterogeneity definition, and discuss why their approach is robust to it.
> >
> > In terms of scores, I will raise them accordingly.

---

> > > ### Author Response · Authors · 2021-09-01
> > > **Thank you for your response - some explanations on heterogeneity**
> > >
> > > We would like to thank you for taking the time to consider our rebuttal and for updating your score.
> > >
> > > Regarding the data-heterogeneity setting in the paper, this point has been clarified in the revised version of the paper, and we would like to  summarise some aspects of our approach.
> > > Our paper copes with data heterogeneity in the following way:
> > > -  **[A]** the model: independent observations are required  (implying that the likelihood of the observations has a product form), but we **do not require identical distributions**. The framework described by A1-A3 covers for example the framework of regression by setting $y_{ij}$ as the pair (response, covariates), and each user may have its own covariate vector, thus making users’ data distribution different. See e.g. Section 2 in [Ref 1] where the same model assumptions A1-A2-A3 are satisfied by the logistic regression. Especially, **we do not assume the same parametric form for all clients**.
> > > - **[B]** Regarding theoretical analysis and the robustness of the convergence results, **we do not make any assumption on the discrepancy between $\bar{s}_i$ and $\bar{s}_j$ for $i \neq j$ to guarantee convergence.** In other words, the convergence is robust to the heterogeneity.
> > >
> > > These points have been highlighted and the logistic regression example detailed in the revised version.
> > >
> > > We hope this can alleviate your concern regarding that aspect of the paper, and thank you again for your constructive comments and consideration.
> > >
> > > [Ref 1] Fort, G. and Moulines, E. The Perturbed Prox-Preconditioned SPIDER algorithm for EM-based large scale learning. In Proceedings IEEE Statistical Signal Processing Workshop 2021 (to appear).

---

### Official Review · Reviewer_XoK6 · 2021-07-06

**Rating:** 3
**Confidence:** 4

**Summary:**

Authors propose and study a distributed variant of Expectation-Maximization for exponential families. Theoretical analysis is conducted to characterize the convergence of the proposed algorithms towards stationary points of the underlying objective function. The empirical performance of proposed algorithms is studied via numerical experiments on synthetic and open data.


**Limitations And Societal Impact:**

this is theoretic work and therefore the societal impact can only be discussed on a rather high level.

**Main Review:**

* It is unclear how data heterogeneity is modeled and handled by the proposed algorithms. As i understood, the main model underlying the proposed algorithms are homogeneous i.i.d. draws from a single exponential famliy Eq. (2). How would your FedEM algorithm cope with completely heterogeneous data which e.g. conforms with a networked exponential family as studied in

A. Jung, "Networked Exponential Families for Big Data Over Networks," in IEEE Access, vol. 8, pp. 202897-202909, 2020, doi: 10.1109/ACCESS.2020.3033817.

* It would help the reader to have a dedicated section or paragraph about existing work and how the proposed methods related to it. Also, Section 2 could be dedicated to the precise problem statement including the communication link constraints.

* It would be good to have a dedidacted paragraph in Section 1 that summarizes the paper's contributions. This paragraph should also discuss the relation between FedEM and VR-FedEM. Is the latter a generalization of the former? Do both algorithms apply in different settings?

* Pls add more discussion on how to choose the input parameters of Algorithm 1 and Algorithm 2

* The numerical experiments need to be expanded and also compare with other methods, e.g., network Lasso methods such as studied in

SarcheshmehPour, Y., Tian, Y., Zhang, L., and Jung, A., “Networked Federated Multi-Task Learning”, <i>arXiv e-prints</i>, 2021. https://arxiv.org/abs/2105.12769

* Pls discuss if the Assumptions 1 - 9 are satisfied for the datasets and models considered in the numerical experiments

* There needs to be more discussion/analysis of the obtained numerical results from the perspective of the theoretical results. Do the plots in Figure 1 confirm the statements/insights provided by your Theorems ?

* I did not find any discussion/reference to Figure 2 in the main text.

* it is mentioned that "A major advantage of the EM algorithm stems from its invariance under homeomorphisms,..." However, it is not clear to me how this invariance is relevant for your methods?

* unclear what is meant by "..where heterogeneity is characterized by the non-cancellation of the local gradients at the optimum,.."

* unclear what is meant by "..conditions must therefore be derived to ensure convergence at the central server." which conditions are you referring to here?

* "...the complexity grows at the rate..."  which complexity do you mean? computational complexity or sample complexity?

* "...see [16] for an interesting discussion on the connection of EM and mirror descent." how is this connection relevant for the following developments?

* could you provide more motivation/intuition for introducing the concept of a mean-field in Eq. (6) ? does your convergence analysis make heavy use of this mean-field?

* "...that the conditional variance is (uniformly) bounded" the conditional variance of what quantity?

* "... of the new sufficient statistics." not clear what the "new sufficient statistic" is

* "Crucially, a naive implementation of compressed distributed EM without assumptions on the heterogeneity would not converge." maybe add a reference to a proof for this statement

* pls add more discussion on the consequences of Proposition 3. how strong does partial participation hurt the performance compared to having full participation?







**Time Spent Reviewing:**

4

---

> ### Author Response · Authors · 2021-08-10
> **Response to Reviewer XoK6**
>
> We are grateful to the referee for his/her many questions. We sincerely hope that the answers below will clarify the novelty and strength of our contributions, and will convince him/her to propose an upwarded rating of the paper.
>
>
>
> We tried to address all of them as precisely as possible, and thus apologize for the long answer. We hope these comments clarify your questions, especially regarding heterogeneity. The revised version will insist strongly on the robustness of our algorithms to heterogeneity, both through the models and numerically.
>
> **Sentences to be clarified.** We agree with the referee that many sentences could be clarified; and the examples pointed by the referee (and by the other referees) will be carefully addressed.  The description of Figure 2 will be also inserted (see the answer to referee nRvM).
>
> **Data Heterogeneity.**  See also the answer to referee At3L, point 1.  Our paper copes with data heterogeneity through many aspects: **[A]** the model: independent observations are required  (implying that the likelihood of the observations has a product form), but we **do not require identical distributions**. The framework described by A1-A3 covers for example the framework of regression by setting $y_{ij}$ as the pair (response,covariates),  each user may have its own covariate vector, thus making users’ data distribution different. See e.g. Section 2 in [Ref 1] where the same model assumptions A1-A2-A3 are satisfied by the logistic regression (this model is part of the examples in the paper in A.Jung (2020), example III-B - pointed by the referee). **[B]** the assumptions: we do not make any assumption on the discrepancy between $\bar{s}_i$ and $\bar{s}_j$ for $i \neq j$ to guarantee convergence. **[C]** the conclusions of our theorems, expressed as a function of the mean value of quantities specific to each node $\# i$ (the $L_i$’s and the $\sigma_i$’s for example).
> Note that the EM framework differs from the inference approach in A. Jung (2020) -- roughly speaking, our approach would correspond to adding a hierarchical parametric model on the unknown weights $w^{(i)}$, and learn the parameters of the $w^{(i)}$'s distribution.
>
> **Communication link constraint.** We consider a centralized framework, with partial availability of the users and limited bandwidth. We will clarify this point.
>
> **Paper’s contribution.** Contributions are listed in the introduction section of the paper.  In the comparison of FedEM and VR-FedEM, the essential points are : **[A]** VR-FedEM differs from FedEM by proposing a technique which reduces the variability of the oracles $S_{k+1,i}$  at each agent $\# i$; it is the first algorithm with this goal in the FL-EM literature. **[B]** FedEM covers the online-learning setting with partial participation. For ease of exposition (and ease of proof), VR-FedEM is derived and studied in the case (i) all the workers participate, (ii) each agent possesses a mini-batch of examples.
>
> To our best knowledge, it is the first algorithmic and theoretical contribution to EM in the FL setting; we will cite references to EM in the distributed setting and outline the essential differences. As explained in the answers to Referee WGsy (and also in the answer to Referee at3L where the existence of the T operator makes EM really different from other optimization methods) EM and Gradient Descent (GD) are two different settings and for this reason, it is not adequate to compare our contribution to the GD literature.
>
> **Validity of Assumptions.** See also the answer to  Point 5, referee at3L. A1, A2, A3 are easily verified (see the supplementary material); they are really classical in the EM literature. They ensure the E-step to consist in computing expectations of summarizing statistics (so called *sufficient statistics*) and the M-step to be well defined.  In many examples, the assumptions A4, A5, and the condition on the variance in A6 and A9 are satisfied if $\hat{S}_k$ remains in a compact subset. Preliminary results in the EM for non FL context are provided in [Ref 1] to ensure this compactness property. A7 is a classical assumption on compression operators, which is valid for a large class of compressors (including quantization, sparsification, randomization). A8 is a model on the users availability, that could be extended to user dependent probabilities of participation.
>
> **Discussion on input parameters.** A discussion on how the complexity (for example, in terms of number of optimization steps) depends on the input parameters in Algo 1 and Algo 2, will be added. It will show how to scale these input parameters to guarantee an accuracy level when the algorithms are stopped (even if stopped with a random termination rule, as in Theo 1 and Theo 4). For the parameter $\alpha$, we will highlight the fact that theory suggests the use of the largest possible $\alpha$ smaller than a value depending on $\omega$: this is why we choose the proposed values in the theorems.
>
> **Proposition 3.** Comments on the consequence of partial participation will be added. For example, it impacts the maximal learning rate except if the number of agents $n$ increase or the compression rate decreases.
>
> **Numerical results.** Figure 1 and Figure 2 illustrate the variance reduction on the path of the objective function evaluated along the stochastic sequence $\hat{S}_{t,k}$ produced by VR-FedEM when compared to the path $\hat{S}_k$ produced by FedEM. Since the objective function evaluated at the current point (see Eq. 6) is not explicit in the framework addressed in our paper, we show that its approximation inherits this property. We will add complementary experiments in the appendix to highlight the impact of heterogeneity, and other parameters (step sizes, number of iterations) -- See the answer to referee nRvM.
>
> **Mean Field.**  “*mean-field*” is classical in the Stochastic Approximation (SA) literature to name the function of which the method has to find the roots. As in SA, our proofs are based on the existence of a differentiable Lyapunov function (see e.g. section 7.4 Eq(15)) which expresses that the algorithm is attracted by the roots of the drift term (see Eqs (15-16)). When the drift term is unbiased (case of Theorem 1), the roots are the roots of $h$ under A5; when it is biased (case of Theorem 4), the proof is more complex but the limiting points of the algorithm remain the roots of $h$. As a conclusion, our work is part of the “FL-through-SA” literature, we find more correct to outline the link by using the adequate vocabulary.
>
> **Divergence of a naive implementation of compressed distributed EM.** A naive implementation of Distributed EM would consist in aggregating the (compressed) values for the participating devices, and not using control variates $V_{k,i}$. At line 13 in FedEM, one would use $H_{k+1} = \frac{1}{np} \sum_{i\in \mathcal{A}_{k+1}} Q(\hat{S}_{k+1})$. With *either* compression or partial participation, the convergence of such an algorithm is severely hindered by heterogeneity (see [Ref 2], for a similar observation in  the gradient case). We will numerically illustrate this phenomenon -- see answer to referee nRvM.
>
> **Heterogeneity in the Gradient Descent setting.** One classically used assumption on  heterogeneity  in gradient based FL frameworks is that the local gradient on worker $\# i$ at the optimal model $\nabla F_i(w_*)$ is bounded.
>
> **Invariance by homeomorphism, EM and mirror descent.** We agree with the referee that these comments are not directly relevant for the developments in the paper. They were also introduced to outline the differences between EM and Gradient Descent (based on the paper [16]); these sentences will be either reformulated in order to address the remarks of the referee  WGsy, or removed.
>
>
> We hope that these clarifications will alleviate the reviewer's concerns and that he/she will revise his judgment on the article. We will be happy to provide further clarification if necessary.
>
>  ---
>
> - [Ref 1] Fort, G. and Moulines, E. The Perturbed Prox-Preconditioned SPIDER algorithm for EM-based large scale learning. In Proceedings IEEE Statistical Signal Processing Workshop 2021 (to appear).
> - [Ref 2] Philippenko, C and Dieuleveut, A. "Artemis: tight convergence guarantees for bidirectional compression in federated learning", 2020

---

### Official Review · Reviewer_at3L · 2021-07-13

**Rating:** 7
**Confidence:** 3

**Summary:**

The authors study the problem of expectation-maximization in a federated setting. They propose two communication-efficient algorithms where the clients compress certain sufficient statistics to save communication. For each algorithm, convergence rates are presented that match their centralized counterparts in the absence of compression.

**Limitations And Societal Impact:**

The authors briefly discuss some limitations and societal impacts of their work in Section 5.

**Main Review:**

Originality and Pros: Both the problem studied, and the algorithms presented, are novel to the best of my knowledge. The compression strategies used are somewhat unique to the problem being studied which makes them interesting. Concrete analytical results are presented to support the proposed algorithms.

Clarity: The paper is mostly well-written, but certain parts could do with better explanations. In what follows, I explain these concerns in detail.

(1) In the standard FL setting, statistical heterogeneity causes the local loss functions of the clients to have potentially different minima. As a result, infrequent communication leads to "client-drift" which, in turn, can cause convergence to incorrect points. For the federated EM problem the authors' study, I found it hard to understand what heterogeneity means (despite the explanation in lines 58-59), and what the implications of heterogeneity are for convergence. A concrete mathematical description of heterogeneity would be very helpful in this context.

(2) One of the major concerns in FL is to reduce the number of communication rounds. To do so, almost all standard FL algorithms I am aware of (e.g., FedAvg, FedProx, SCAFFOLD, FedSplit, etc. ) involve periodic communication between the clients and the server. In between two communication rounds, clients perform multiple local computations on their private data. In contrast, if I understand correctly, both the algorithms proposed in the paper involve communication between the clients and the server *at every iteration*. This is quite undesirable since the number of communication rounds can be significantly higher relative to existing FL algorithms.

The authors need to discuss this potential limitation of their approach. How would periodic interactions impact the convergence guarantees of their proposed algorithms?

(3) Continuing with the above point, it is precisely due to infrequent communication that data heterogeneity creates a problem in standard FL settings. Indeed, even in the presence of heterogeneity, if the clients exchanged model updates with the server at every iteration, the iterates of all clients would evolve synchronously, i.e., this would essentially be equivalent to a centralized setting. Said differently, if the communication period is 1, then the "client-drift" phenomenon that arises due to data heterogeneity in common FL algorithms would cease to exist.

Since the authors do not consider periodic communication, how (if at all) does heterogeneity create a challenge? To make my question clearer, suppose there is no compression/quantization. Moreover, suppose all clients participate in every round of communication. Even in these scenarios, data heterogeneity leads to challenges in the analysis of common FL algorithms such as FedAvg and FedProx, precisely due to infrequent, periodic communication. In contrast, I would imagine that if there is no compression, and full client participation at all times, Fed-EM would essentially behave as a standard centralized EM algorithm. Isn't this correct?

The point I am trying to make here is that not considering infrequent communication takes away the typical challenges posed by data heterogeneity in standard FL settings. This point needs to be acknowledged and discussed.

(4) The authors do not cite or discuss any of the popular FL algorithms such as FedAvg, FedProx, or SCAFFOLD (to name a few). I would encourage the authors to compare their model of heterogeneity and the communication structure of their algorithm to the ones in the following references.

[R1] "Tighter theory for local SGD on identical and heterogeneous data", Khaled et al, AISTATS 20.

[R2] "On the convergence of fedavg on non-iid data", Li et al., arXiv 19.

[R3] "On the convergence of federated optimization in heterogeneous networks", Sahu et al., arXiv 18.

[R4] "Scaffold: Stochastic controlled averaging for federated learning", Karimireddy et al., ICML 20.

(5) Regarding Section 2, the authors list several assumptions without commenting on how strong these assumptions are. Are these assumptions standard in the analysis of EM algorithms? How do they compare to the typical assumptions of convexity and smoothness in standard FL optimization settings? (such as those in [R1]-[R4])

(6) A couple of questions regarding Algorithm 1: (i) Is the operator T known to all clients? (ii) Lines 6-8 appear to be inspired by the "error-feedback" mechanism employed when one considers compression/sparsification of gradients. If so, this should be stated clearly. (iii) In line 5, what does it mean to sample a sufficient statistic S_{k+1,i}?

(7) (i) Although the variance reduction method in Algorithm 2 appears to be theoretically sound, it is not adequately motivated as I explain below. Typically in FL, variance reduction or gradient tracking is used to reduce the "client-drift" phenomenon that arises due to data heterogeneity and infrequent communication (see [R4] above and [R6] below). Since the authors do not clearly explain what challenge heterogeneity poses for their specific problem, I found it hard to appreciate the need for Algorithm 2.

(ii) Unlike Algorithm 1, Algo. 2 involves both outer and inner loops. However, no explanation is provided for the rationale behind this.

(iii) How does variance reduction improve (if at all) the convergence guarantees relative to Algorithm 1? Does this come at the expense of more communication? It would be instructive if the authors clearly compared the complexity bounds of Theorem 4 with those of Theorem 1.

(iv) The authors should also discuss how their variance reduction approach compares with those used in standard FL settings. In this context, the most relevant works are [R4] above and [R5], [R6] below.

[R5] "Federated Learning with Compression: Unified Analysis and Sharp Guarantees", Haddadpour et al., AISTATS 21.

[R6] "Achieving Linear Convergence in Federated Learning under Objective and Systems Heterogeneity", Mitra et al., arXiv 21.

Reference [R6] in particular employs both gradient tracking and error-feedback to tackle statistical heterogeneity and compression/sparsification, just as in VR-FedEM. Appropriate comparisons are thus warranted.

========================== REVIEW UPDATE =======================================

I have read the response of the authors and also the comments of the other Reviewers. The rebuttal by the authors has improved my understanding of the paper. So I am increasing my score to a 7.





**Time Spent Reviewing:**

4.5 hours

---

> ### Author Response · Authors · 2021-08-10
> **Response to Reviewer at3L**
>
> We sincerely thank the referee for the insightful work on the paper. The revised version will include comments answering the questions raised by the referee; we provide here detailed answers. We tried to answer as precisely as possible all points raised by the reviewer, and apologize for the long answer (maybe the most crucial aspects are Points 1, 2 and 6).
>
> ##### Point 1. Heterogeneity.
> We will answer this point by clarifying what heterogeneity means here, and why heterogeneity is an important problem/challenge even without local iterations.
> **Heterogeneity covered in the paper.** [a] Non iid examples. The observations are assumed independent but **not** necessarily with the same distribution -- see the general form of A1. For example, in a regression framework, each observed example $y_{ij}$ could be the pair (answer, regressors) with a vector of regressors specific to each individual. [b] Non uniform-in-$i$ ssumptions. For example, A2 and A4 allow each node to have their (objective) statistic $\bar{s}_i \circ T$ with their own properties:  we have no assumptions inducing similarities. In our results, the upper bounds depend on the mean values of local properties (e.g. $\sigma_i^2$ and  $L_i$ in Theorem 1).
>
> **Client-drift and communications.** **[a]** Combining compression and heterogeneity imply non-trivial challenges: heterogeneity strongly hinders convergence in the presence of compression. This is known in the Gradient Descent setting, see e.g. [Refs 1,3]), the intuition being the followings: let us say convergence is reached and the current iterate of all the agents is equal to the limiting value $w^\star$, then they communicate a non-zero value since $w^\star$ is not the local optimum, which are amplified by the compression operator (see the quadratic variance in A7). Therefore, the central server synthetizes a term with high variability thus preventing convergence. **[b]** Consequently, not considering multiple local iterations does not make the problem easy. The interactions between compression and heterogeneity, even with a single local iteration, have been an active area of research [Refs 2,3,4,5]. **[c]** Finally, local iterations are questionable in the EM setting (see Point 2).
>
> ##### Point 2. Communication complexity and local iterations.
> **Local iterations in EM.** **[a]** let us mention that *performing local iterations may not be possible in the EM setting*: one iteration of EM is the combination of two steps E and M and the M step, which corresponds here to the use of the map T, is **only** performed by the central server; this remark is fundamental for the points 3 and 6 too and it is a fundamental specificity of the EM framework (which is not shared by the gradient framework).  In applications, we usually do not want T to be available at each local node (see Point 6).
> **[b]** Our work *allows us to perform multiple local iterations of the E step before communicating with the central server*. In Algorithm 1, the local statistics $S_{k+1,i}$ are general enough: they may be the result of a single new observation, or of a more complex local processing aggregating many observations. Assumption A7 requires $S_{k+1,i}$ to be an *unbiased* estimator of $\bar {s}_i \circ T (\hat{S}_k)$ and to have a uniformly bounded (conditional) variance. Many local E iterations may help reducing this variance. Algorithm 2 uses mini-batches (see line 8) and in that sense, includes many E-iterations.
> **[c]** In the non-FL EM setting, it is known that performing maximization steps in the first iterations improve the convergence rate (see e.g. Section 5 of  [Ref 6] where many calls to T before a full pass on the data explain why so-called “variance-reduced incremental EM” drastically improve on EM in the large scale learning setting). Such a property is expected in the FL setting and will be illustrated numerically.
>
> **Reduction of communication complexity without local iterations.** Four techniques can be used to reduce the amount of communication: (i) increasing the minibatch size and reducing the number of iterations, (ii) increasing the number of local steps between two communications, (iii) using compression, (iv) sampling clients at each step. Here, we provide a careful and tight analysis of strategies (i),  (iii) and (iv) (sampling client is part of Partial Participation).
> Moreover, from a theoretical standpoint, tradeoffs between larger minibatch and more local iterations are unclear [Ref 7].
> ##### Point 3
> This is a good remark; with no compression and without local iterations, we expect a naive distributed EM to perform similarly. However, the compression step makes the heterogeneity management difficult, even without local iterations (see Point 1).
> ##### Point 4.
> Many thanks for these references: the above discussion (point 2) will be added, explaining why comparing to FedAvg, FedProx, Scaffold is not directly possible.
> ##### Point 5.
> See also the answer to referee XoK6.  **[A]** A1 and A2 are standard in the EM literature: they define the so-called “exponential family setting” in which EM (and especially the M-step) is doable.  **[B]** A3 guarantees the existence of the map T: in EM, it is absolutely essential that the minimization of the surrogate function is simple. Even if the initial optimization problem (defined by Eq. (1-2)) is not necessarily convex, A3 cannot cover *general* non-convex objective functions. **[C]** A4, A5 and part of A6 (about the conditional variance) are the most challenging conditions; they are easily verified as soon as the boundedness (or at least, the growth) of the statistics $S_{k+1,i}$ is controlled. Preliminary results exist in the literature for incremental-type EM (close to Algorithm 2 in the non FL-case) but the theoretical analysis is far more intricate  [see Ref 8] and extensions to FL-EM are out of the scope of this paper.  **[D]** A7 is a classical assumption, valid for a large class of compressors (including quantization, sparsification, randomization). **[E]** A8 is a model on the users availability, that could be extended to user-dependent probabilities of participation.
> ##### Point 6.
> In the batch setting (i.e. when a *finite* number of examples is  processed by the system), the map T may depend on *all* the observations. See for example Eq. (48) where the function $\psi$ is a sum over the full data set (there is a typo in the paper, the sum is missing).  We do not require T to be known by the local servers. First, in order to reduce the computational cost, the optimization step (lines 16 in Algo 1 and Algo 2) can be factorized and performed by the central server. Second, for privacy considerations, T is available at the central server (even if T necessitates a computation of a sum over all the data, each local server can send its *local full sum*  to the central server prior the run of the algorithm and not all its examples !).
> **Error-feedback** (EF) is a *different approach* to the memory process we adopted (first introduced in [Ref 3]) with a different purpose: EF was introduced for *biased* contractive compression operators (which is not the case here, see A7), while memory serves on heterogeneity.
> “Sample a sufficient statistic” means “sample a random variable that approximates $\bar{s}_i \circ T(\hat{S}_k)$”; it has to satisfy A6.
> ##### Point 7.
> **(i, ii, iv)** The central server performs a (robust-to-compression and FL) Stochastic Approximation (SA) procedure, see line 14 of Algorithm 2, in order to find the roots of the non-explicit mean-field $h$ (see Eq.(6)). To that goal, SA methods compute an approximation of $h$ evaluated at the current iterate; “variance reduction” means here that the variance of this approximation is reduced. In our paper, the variance reduction relies on a “control variate technique” thus explaining the inner/outer loops. The inner loops define this control variate (see line 8), but unfortunately provide a *biased approximation* of the mean field (see Proposition 19). Therefore, at each outer loop, the control variate  is refreshed (see line 20)  thus removing the bias (see Proposition 14).
> **(iii)** We will comment on this point by a careful comparison of the complexities of the methods. Numerically, the variance reduction and its benefit can be observed in Figures 1 and 2.
>
>
> ---
>
> - [Ref 1] Philippenko, C and Dieuleveut, A. "Artemis: tight convergence guarantees for bidirectional compression in federated learning", 2020
> - [Ref 2] Horvath and Richtarik, "A Better Alternative to Error Feedback for Communication-Efficient Distributed Learning", ICLR 2021.
> - [Ref 3] Mishchenko et al., 2019, "Distributed Learning with Compressed Gradient Differences"
> - [Ref 4] Gorbunov et al., "MARINA: Faster Non-Convex Distributed Learning with Compression", ICML 2021.
> - [Ref 5] Alistarh et al, "QSGD: Communication-Efficient SGD via Gradient Quantization and Encoding", NeurIPS 2017.
> - [Ref 6] Gach, P. and Fort, G. and Moulines, E. Fast Incremental Expectation Maximization for finite-sum optimization: non asymptotic convergence.  Statistics and Computing, 2021 (to appear).
> - [Ref 7] Woodworth et al., 2020, “Is local SGD better than minibatch SGD?
> - [Ref 8] Fort, G. and Moulines, E. The Perturbed Prox-Preconditioned SPIDER algorithm for EM-based large scale learning. In Proceedings IEEE Statistical Signal Processing Workshop 2021 (to appear).

---

> > ### Comment · Reviewer_at3L · 2021-08-24
> > **Response to Rebuttal**
> >
> > I thank the authors for their rebuttal. I have a better understanding of the paper now. I would sincerely recommend the authors to make revisions to their paper that incorporate the above discussions, and also the comments by the other Reviewers. Specifically, (i) the notion of heterogeneity needs to be better explained; (ii) the challenges of considering heterogeneity+compression for EM need to be highlighted; (iii) the references on FL algorithms (in particular, the algorithms that use variance-reduction+compression) need to be discussed; and (iv) the benefits of variance reduction should be clearly stated. With the hope that the authors will make these revisions, I am improving my score to a 7.

---

### Official Review · Reviewer_nRvM · 2021-07-30

**Rating:** 8
**Confidence:** 1

**Summary:**

This paper investigates how to federate the Expectation Maximization (EM) algorithm in the case of a curved exponential family.
The authors propose a novel algorithm, called FedEM, which translates the EM algorithm in the FL setting, which consists in iteratively sending local sufficient statistics to the server and updating the value of the model. This algorithm takes into account the distributed aspects, communication constraints (using an unbiased compressor with an error feedback mechanism for the local statistics to send), partial participation, and data heterogeneity of the FL setting.
The authors prove the convergence of FedEM (Corollary 2).
A variance-reduced version of FedEM is also proposed (VR-FedEM), also with convergence guarantees (Th. 4).
Experiments are performed on toy synthetic data, MNIST and real data (missing value imputation from a tabular dataset, bird).

**Limitations And Societal Impact:**

See above

**Main Review:**

## Originality

To the best of my knowledge, this is the first work tackling the federation of the EM algorithm, which makes it novel.

## Clarity

Overall, the paper is very well-written and easy to follow even for non-experts. However, in the experimental section (page 8), it is not stated what figure 2 refers to - I assume it is the MNIST experiments given the number of components estimated.

## Quality

Regarding the theoretical results, I did not check in detail the proofs, so I cannot judge their correctness.

If any, the experimental section could be further expanded.
- For instance, the author state that a naïve implementation of the EM algorithm in FL would not be resilient to heterogeneity, but there is no experimental demonstration of this assertion.
- Further, there is no study of the effect of compression
- Last, there is no comparison with the results obtained in a single node setting: does one retrieve results which are close to them?

## Significance

The proposed algorithms and the convergence proofs are the main contributions of the paper. While the experimental section is a bit weak, I do think this is a minor problem given the other contributions.


========
Update following rebuttal
========

I thank the authors for clearly addressing all my criticisms in their answer, and for the time and precision they devoted to answer the other reviewers. I will raise my score to 8.

**Time Spent Reviewing:**

2

---

> ### Author Response · Authors · 2021-08-10
> **Response to Reviewer nRvM**
>
> First, we sincerely thank the referee for his/her time, comments and his/her positive appreciation of our contributions.
>
> **Comments on Figure 2**. We agree that the comments on Figure 2 are missing; it is indeed related to the MNIST example.  The revised version will include a description “ Figure 2, shows the sequence of parameter estimates for the weights and the squared norm of the approximated mean field $\| H_k \|^2$ for FedEM (resp. $\|H_{t,k}\|^2$ for VR-FedEM) vs the number of epochs”; (an “epoch” is defined above in the text; it is a full pass on the data set).
>
> **Numerical section** In the revised version, we added numerical experiments both in the main text and the supplementary material, to provide more insight on the effect of heterogeneity and compression:
> - In the simulated Gaussian Mixture Model (GMM) application,  we considered increasing heterogeneity settings, where examples from the two components of the mixture are (i) assigned to the local servers completely at random, (ii) assigned to the local servers at random with unbalanced distributions and (iii) assigned to different local servers. In all cases we compare FedEM and VR-FedEM to a naive distributed EM implementation; this experiment demonstrates that the latter diverges in heterogeneous settings.
>
> - In the same example, we study the effect of the compression operator on the empirical convergence of EM and VR-FedEM. To do so, we use the vector quantization compression operator with different compression levels, and highlight the impact of the compression level on the two algorithms. In this experiment we observe that, compression by a factor between 4 and 8 (from 32 bits per coordinate to 8 or 4 bits per coordinate) does not impact significantly the convergence of the two algorithms.
>
> - We provide numerical comparisons of the results of FedEM and VR-FedEM to a single node setting with a classical EM. On the GMM example with two mixture components, the average relative estimation error for the mixture weights is of order 1e-2.
>
> - We study empirically the impact of partial participation of the local servers on the obtained results. To do so, we perform experiments where each worker participates with probability $0<p<1$ at each iteration. We performed experiments for $p\in\{0.25, 0.5, 0.75, 1\}$. Our experiments confirm our theoretical findings; for instance, for the FedEM algorithm, we observed that the saturation is inversely proportional to $p$ ($p=1$: $2\times 1e-3$, $p=0.5$: $4\times 1e-3$).

---

> > ### Comment · Reviewer_nRvM · 2021-08-31
> > **Thanks**
> >
> > Thank you for your precise answer. In light of the additional experiments you propose and the very clear and precise comments you gave in other answers to reviewers, I raised my score accordingly.

---

### Decision · Program_Chairs · 2021-09-27

**Decision:**

Accept (Poster)

**Comment:**

The reviewers appreciate the incorporation of EM algorithm into federated learning for dealing with data heterogeneity. The paper makes good theoretical contributions but also has weaknesses in experiments. Overall, I am in favor of acceptance. Please incorporate the new results in the final version. Please revise the paper based on the reviews and rebuttals.